# A NPAS4–NuA4 complex couples synaptic activity to DNA repair

Elizabeth A. Pollina[1,3,4], Daniel T. Gilliam[1,4], Andrew T. Landau[2], Cindy Lin[1], Naomi Pajarillo[1], Christopher P. Davis[1], David A. Harmin[1], Ee-Lynn Yap[1], Ian R. Vogel[1], Eric C. Griffith[1], M. Aurel Nagy[1], Emi Ling[1], Erin E. Duffy[1], Bernardo L. Sabatini[2], Charles J. Weitz[1] & Michael E. Greenberg[1✉]

Neuronal activity is crucial for adaptive circuit remodelling but poses an inherent risk to the stability of the genome across the long lifespan of postmitotic neurons[1–5]. Whether neurons have acquired specialized genome protection mechanisms that enable them to withstand decades of potentially damaging stimuli during periods of heightened activity is unknown. Here we identify an activity-dependent DNA repair mechanism in which a new form of the NuA4–TIP60 chromatin modifier assembles in activated neurons around the inducible, neuronal-specific transcription factor NPAS4. We purify this complex from the brain and demonstrate its functions in eliciting activity-dependent changes to neuronal transcriptomes and circuitry. By characterizing the landscape of activity-induced DNA double-strand breaks in the brain, we show that NPAS4–NuA4 binds to recurrently damaged regulatory elements and recruits additional DNA repair machinery to stimulate their repair. Gene regulatory elements bound by NPAS4–NuA4 are partially protected against age-dependent accumulation of somatic mutations. Impaired NPAS4–NuA4 signalling leads to a cascade of cellular defects, including dysregulated activity-dependent transcriptional responses, loss of control over neuronal inhibition and genome instability, which all culminate to reduce organismal lifespan. In addition, mutations in several components of the NuA4 complex are reported to lead to neurodevelopmental and autism spectrum disorders. Together, these findings identify a neuronal-specific complex that couples neuronal activity directly to genome preservation, the disruption of which may contribute to developmental disorders, neurodegeneration and ageing.

Sensory experience is essential for proper neuronal maturation and circuit plasticity[1]. The signalling cascades initiated by experience-driven neuronal activity culminate in the induction of gene programmes that control diverse processes such as dendrite and synapse growth, synapse elimination, recruitment of inhibitory neurotransmission, and adaptive myelination[6,7]. However, neuronal activity also threatens the genomic integrity of postmitotic neurons that must survive the lifetime of an organism. For example, heightened metabolic demands during periods of elevated activity may increase oxidative damage to actively transcribed regions of the genome[8]. Activity-induced transcription itself poses a further threat to genome stability, as it has been linked to the induction of repeated DNA double-strand breaks (DSBs) at regulatory elements, such as the promoters of stimulus-inducible genes[2–5,9,10]. Although the coupling of transcription to DNA breaks is observed across cell types, this process poses a specific challenge to long-lived neurons, which cannot use replication-dependent DNA repair pathways and possess limited regenerative mechanisms to replace damaged cells[11]. Accumulating DNA damage to neuronal genomes is a cardinal feature of

neurodegenerative disorders and organismal ageing[12,13]. Thus, understanding the strategies that neurons use to prevent and repair damage may have direct translation to human longevity and ageing therapies. So far, there are no examples of neuronal-specific repair machinery that mitigate the risks of genome instability during heightened activity. By investigating features of the activity-dependent transcriptional programme specific to neurons, we discover a biochemical coupling of neuronal activity to DNA repair through a previously unknown form of the NuA4 chromatin remodeller–DNA repair complex that assembles around the inducible, neuronal-specific transcription factor NPAS4.

## Identification of the NPAS4–NuA4 complex

Unlike most activity-inducible transcription factors, which are broadly expressed and induced by various stimuli, NPAS4 is selectively expressed in neurons following membrane depolarization-induced calcium signalling[14]. To understand the functions of this factor, which is specifically attuned to neuronal activity, we sought to purify

[1]Department of Neurobiology, Harvard Medical School, Boston, MA, USA. [2]Department of Neurobiology, Howard Hughes Medical Institute, Harvard Medical School, Boston, MA, USA. [3]Present address: Department of Developmental Biology, Washington University School of Medicine, St Louis, MO, USA. [4]These authors contributed equally: Elizabeth A. Pollina, Daniel T. Gilliam. ✉e-mail: meg@hms.harvard.edu

NPAS4-containing protein complexes from the adult mouse brain. We reasoned that NPAS4 might assemble into a multisubunit complex that expands its biochemical activities in activated neurons. Using size-exclusion chromatography and non-denaturing gel electrophoresis, we observed that NPAS4 resides in a high molecular weight complex of around 1 MDa. As the predicted size of NPAS4 with either of its heterodimer partners (ARNT1 and ARNT2) is around 175 kDa, this finding suggests that NPAS4 interacts with multiple unknown protein partners (Extended Data Fig. 1a).

To facilitate the purification of this putative NPAS4 complex, we generated *Npas4–Flag-HA* and *Arnt2–Flag-HA* knock-in mouse lines in which the epitope tags Flag and haemagglutinin (HA) are appended to the carboxy termini of NPAS4 and ARNT2. In homozygous knock-in mice, we validated the correct genomic insertion of the tags and verified overlapping immunostaining of endogenous NPAS4 or ARNT2 and the HA epitope (Fig. 1a,b and Extended Data Fig. 1b,c). We also demonstrated that wild-type and tagged NPAS4–Flag-HA (NPAS4–FH) exhibit similar expression levels and induction kinetics (Extended Data Fig. 1d). To generate high levels of NPAS4 required for biochemical purification, we stimulated neurons in the hippocampus of *Npas4–FH* mice through the injection of low-dose kainic acid (KA), a glutamate receptor agonist that synchronously depolarizes hippocampal neurons. We then immunopurified NPAS4 using anti-Flag antibodies and performed mass spectrometry (Fig. 1c and Extended Data Fig. 1e). The mass spectrometry data revealed interactions between NPAS4 and all reported subunits of a single chromatin modifier, the NuA4 complex, the estimated size of which is approximately 1.0–1.3 MDa (Fig. 1c and Supplementary Table 1)[15–17].

We first demonstrated co-immunoprecipitation between NPAS4 and several NuA4 subunits (TRRAP, EP400 and DMAP1) in the visual cortex using light exposure as a physiological stimulus to induce neuronal activity (Extended Data Fig. 1f). We further characterized this complex by immunoprecipitating either NPAS4 or a component of the NuA4 complex, TIP60 (also known as KAT5), followed by immunoblotting and mass spectrometry analyses. These experiments confirmed that the interaction between NPAS4 and the NuA4 complex is reciprocal. Moreover, NPAS4–ARNT2 dimers were among the most abundant transcription factors associated with NuA4 in the brain. In addition, a new subunit of the complex, the poorly characterized protein ETL4, was identified (Extended Data Figs. 1g–j and 2a,b and Supplementary Table 1). We observed that several subunits of the NuA4 complex interacted before stimulation, which suggests that following NPAS4 induction, NPAS4–ARNT dimers join a pre-existing complex (Extended Data Fig. 2b). However, we cannot exclude the possibility that other subunits associate with, or post-translational modifications are added to, the NuA4 complex following activity. NPAS4 is the major inducible component of the complex at the RNA level (Extended Data Fig. 2c). Neither the related protein NPAS3 nor another activity-inducible factor, FOS, co-immunoprecipitated with NuA4 components in the brain. This result highlights the specificity of the NPAS4–NuA4 interaction (Extended Data Fig. 2b). Moreover, NPAS4, but not other activity-inducible transcription factors such as FOS and EGR1, interacted with NuA4 components in a heterologous expression system (HEK293T cells) (Extended Data Fig. 2d,e). In both human and mouse cells, expression of these 19 new interactors of NPAS4 were enriched in neurons compared with other brain cell types[18] (Fig. 1d and Extended Data Fig. 2f,g). This result suggests that within the brain, the NPAS4–NuA4 complex functions specifically within neurons.

To determine whether NPAS4 and NuA4 co-localize on chromatin, we performed CUT&RUN[19] to obtain a map of NPAS4–NuA4 genomic binding in stimulated hippocampal nuclei isolated from mice injected with low doses of KA (Fig. 1e and Supplementary Table 2). We confirmed the specificity of the NPAS4 CUT&RUN signal by performing NPAS4 CUT&RUN on nuclei isolated from unstimulated brains (in which little to no NPAS4 is expressed) and on nuclei isolated from brains of stimulated *Npas4* knockout mice. This experiment generated a list of 10,225 high-confidence NPAS4-binding sites (Fig. 1e and Extended Data Fig. 3a–c). These binding sites were highly correlated with NPAS4 signal from chromatin immunoprecipitation assays with sequencing (ChIP–seq) and showed enrichment of the E-box and bHLH–PAS binding motifs (Extended Data Fig. 3d,e). NPAS4, its partner ARNT2, the NuA4 component EP400 and the newly identified NuA4 subunit ETL4 also co-localized across the genome in stimulated neurons (Fig. 1e–g and Extended Data Fig. 3f,g). Moreover, binding of both NPAS4 and ETL4 to the genome was highly inducible at NPAS4 sites (Fig. 1f). By contrast, EP400 was present at NPAS4-binding sites before stimulation, which suggests that it may be retained at these sites in the absence of NPAS4 owing to the ability of NuA4 to bind to acetylated histones[20] or by NuA4-independent binding of EP400 (Fig. 1f,g). However, significantly more EP400 CUT&RUN signal was observed at NPAS4 sites that lack FOS than the converse (that is, sites with FOS but no NPAS4). This result demonstrates the specificity of NPAS4 and EP400 co-binding rather than a general recruitment of EP400 by activity-inducible transcription factors (Extended Data Fig. 3h,i). In summary, our biochemical evidence and genomic binding assays identified a neuronal-specific form of the NuA4 complex that assembles with NPAS4 in multiple brain regions in an activity-dependent manner.

## NPAS4–NuA4-driven inducible gene programmes

We next investigated the functions of the NPAS4–NuA4 complex in the brain. Studies using yeast, flies and non-neuronal cells have ascribed two activities to the NuA4 complex: (1) controlling transcription through chromatin regulation[15–17] and (2) coordinating repair of DNA DSBs[21–26]. We therefore considered the possibility that NPAS4, by recruiting NuA4, might serve a dual purpose. Previous studies of NPAS4 suggest that the NPAS4–NuA4 complex probably has a central role in activity-dependent gene regulation[14,27]. In addition, the role of NuA4 in DSB repair suggests that NPAS4 might have a previously undescribed function in neuronal activity-dependent DNA repair at promoters and enhancers. We examined these two possible functions of NPAS4–NuA4 by disrupting either NPAS4 or TIP60, the histone acetyltransferase component of NuA4 (refs. [15,16]).

To determine whether NPAS4 and NuA4 coordinate activity-dependent transcription, we injected *Npas4[fl/fl]* or *Tip60[fl/fl]* mice with a virus expressing either Cre-mCherry or a recombination-deficient version of Cre (ΔCre-GFP) into contralateral sides of the hippocampus to unilaterally remove either *Npas4* or *Tip60* (Fig. 2a and Extended Data Fig. 4a–c). Note that depletion of *Tip60* does not impair the induction of NPAS4 (Extended Data Fig. 4d,e). We administered a low dose of KA to activate all classes of neurons present in the hippocampus, collected nuclei 6 h later to capture maximal induction of NPAS4 target genes[14] and performed single-nucleus RNA-sequencing (snRNA-seq) (Fig. 2a and Extended Data Fig. 4f–i). In both *Tip60[fl/fl]* and *Npas4[fl/fl]* datasets, we identified ten principal cell types, including five neuronal subtypes (dentate gyrus, CA1, CA3, subiculum and inhibitory), and multiple non-neuronal cell types. We used a nested PCR strategy as an additional step in library preparation to amplify viral transcripts. This step enabled us to assign infection status to each nucleus as Cre-infected (cKO), ΔCre-infected (wild-type) or uninfected (Fig. 2a,b). In both datasets, within each neuronal cluster, the cells subclustered according to infection status (Cre compared with ΔCre). This effect was not driven by the expression of viral transcripts (Methods), but rather reflects the inability of Cre-infected cells to fully induce activity-dependent genes following the depletion of either NPAS4 or TIP60.

We identified differentially expressed genes (Methods; negative fold change (Cre/ΔCre); adjusted $P < 0.01$) within each principal neuronal subtype and focused our analysis on downregulated genes that are more likely to be direct targets of the complex. Across each of the neuronal

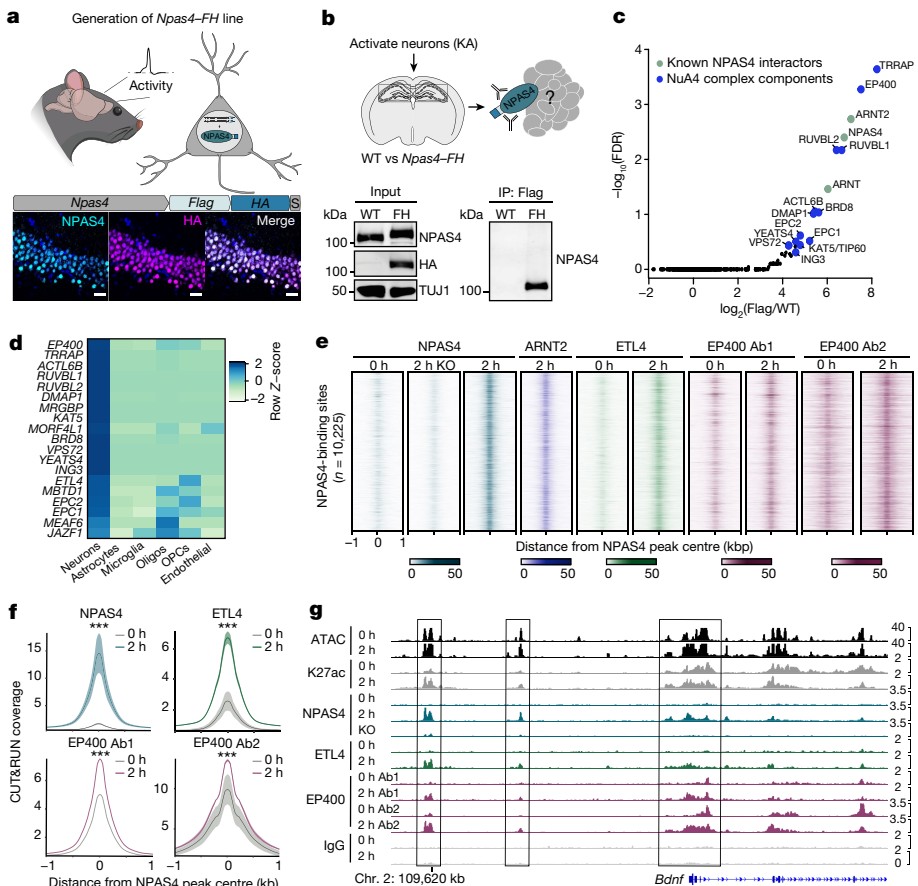

**Fig. 1 | Neuronal activity assembles the NPAS4–NuA4 complex on chromatin. a**, Top, schematic of generation of the *Npas4–FH* mouse model. Bottom, representative immunohistochemistry (performed in triplicate) for NPAS4 and HA in hippocampus samples from *Npas4–FH* mice following 2 h of KA stimulation. Scale bars, 25 μm. **b**, NPAS4–FH and associated protein complexes isolated by anti-Flag immunoprecipitation (IP). Representative western blot (performed in triplicate) with anti-NPAS4 antibody confirms immunoprecipitation only in *Npas4–FH* mice. For gel source data, see Supplementary Fig. 1. TUJ1 processing control was run on a separate gel. **c**, NPAS4-interacting proteins. −log$_{10}$(false discovery rate (FDR)) compared with log$_2$-transformed fold change in peptides obtained by anti-Flag immunoprecipitation in stimulated *Npas4–FH* tissue compared with wild-type tissue (*n* = 3 pools of 8–10 mice), followed by mass spectrometry. Green points indicate known NPAS4 interactors. Blue points indicate NuA4 components. **d**, Expression levels of NuA4 subunits across cell types in human primary motor cortex, displayed as row *Z*-score. snRNA-seq data published by the Allen Brain Institute[18]. Oligos, oligodendrocytes; OPCs,

oligodendrocyte precursor cells. **e**, CUT&RUN signals for NPAS4–NuA4 components in either unstimulated (0 h) or stimulated (2 h) hippocampal tissue. Each NPAS4-binding site is represented as a horizontal line centred at the peak summit and extended ±1 kb. The colour intensity represents the read depth (normalized to 10 million) as indicated by the scale bar for each factor. Data plotted show the aggregate signal from all replicates per factor. Ab1, antibody 1 (Bethyl A300-541-A); Ab2, antibody 2 (Abcam Ab5201). For **e**–**g**, *n* = 3–5 mice pooled per replicate. Replicate numbers provided in Supplementary Table 2. **f**, Aggregate CUT&RUN coverage (fragment depth per bp per peak) at NPAS4-binding sites. Signals displayed as mean ± s.e.m. ***P < 2.2 × 10$^{-16}$. *P* values were calculated from the average signal extracted in a 2 kb window centred on NPAS4 peaks using unpaired, two-tailed Wilcoxon rank-sum tests. **g**, Integrative Genomics Viewer (IGV) tracks of NPAS4–NuA4 CUT&RUN signal at the activity-inducible gene *Bdnf*. Data plotted show the aggregate signal from all replicates per factor. The *y* axis displays normalized coverage scaled for each factor at the chosen locus.

subtypes, genes downregulated owing to *Npas4* deletion were also significantly downregulated following *Tip60* loss in the corresponding cell cluster of the *Tip60$^{fl/fl}$* dataset, with the reciprocal comparison producing the same result (Fig. 2c and Extended Data Fig. 5). In addition to capturing known NPAS4 target genes such as *Nptx2*, *Plk2* and *Bdnf*, we identified 1,766 new targets of NPAS4 in the hippocampus and defined their cell-type-specific expression patterns (Supplementary Table 3). In primary cultures of mouse neurons, we independently confirmed that components of the NuA4 complex, NPAS4, TIP60 and EP400, each regulate the same inducible transcriptional programme (Extended Data Fig. 6a–c). As further corroboration of the importance of the NPAS4–NuA4 interaction for gene activation, expression of truncated forms of NPAS4 that do not strongly interact with NuA4 (Extended Data Fig. 2d) significantly impaired the ability of NPAS4 to activate its target binding sites in luciferase reporter assays (Extended Data Fig. 6d,e). Together, these findings provide evidence that NPAS4 and NuA4 are not

only key regulators of neuronal activity-dependent gene transcription but also activate gene expression as a single functional unit.

As a final test confirming that NPAS4 and NuA4 function together as a complex in neurons, we asked whether TIP60 has the same role in activated neuronal circuits as observed for NPAS4. A key function of NPAS4 in excitatory CA1 pyramidal neurons is to mediate recruitment of somatic inhibition in hippocampal pyramidal neurons in response to neuronal activity[7,28]. We therefore sparsely injected the CA1 region of *Npas4$^{fl/fl}$* and *Tip60$^{fl/fl}$* mice with an adeno-associated virus (AAV) encoding Cre-mCherry and performed simultaneous recordings of evoked inhibitory postsynaptic currents (IPSCs) from neighbouring wild-type (Cre-mCherry⁻) and cKO (Cre-mCherry⁺) pyramidal neurons in acute hippocampal slices (Fig. 2d,e). Axons within the stratum pyramidale were stimulated with the minimum strength required to generate IPSCs from wild-type and cKO neurons. Deletion of NPAS4 or TIP60 significantly reduced activity-dependent somatic inhibition induced

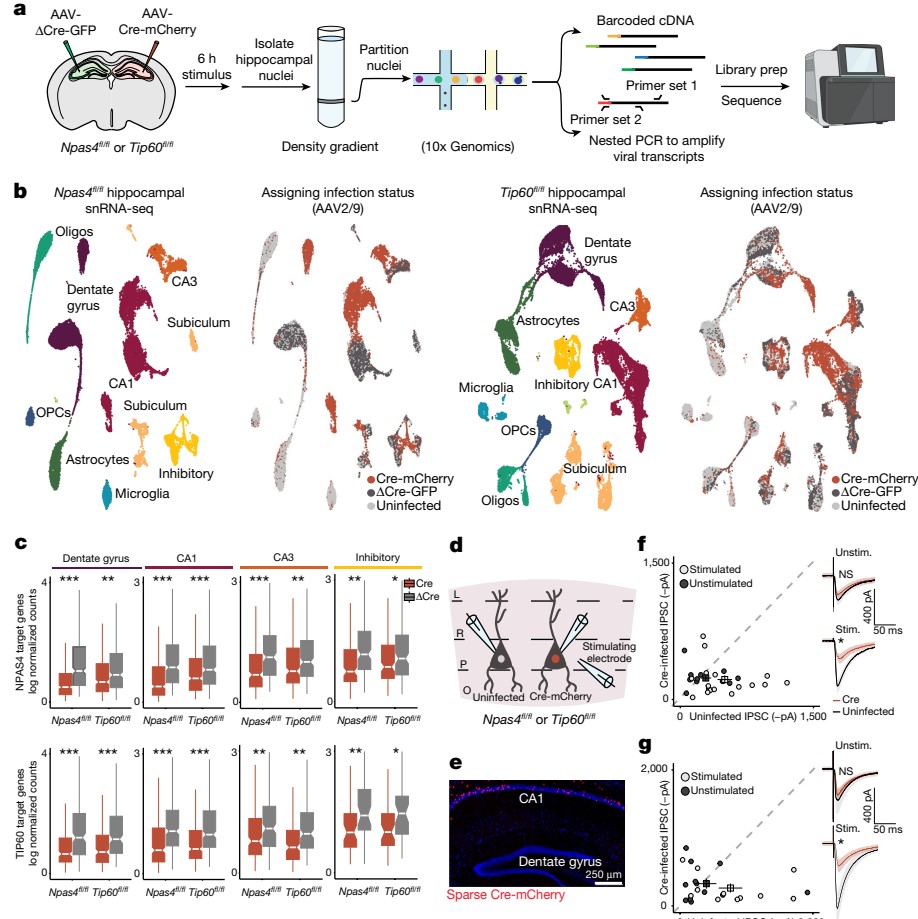

**Fig. 2 | NPAS4–NuA4 regulates activity-dependent transcription and recruitment of somatic inhibition. a**, Scheme of the experimental design. Nuclei were isolated from hippocampi of *Npas4*[fl/fl] and *Tip60*[fl/fl] mice expressing either Cre-mCherry or ΔCre-GFP, followed by snRNA-seq. An additional PCR step to amplify viral transcripts within the cDNA library was used to assign infection status to each nucleus. **b**, Left, uniform manifold approximation and projection (UMAP) visualizations of *Npas4*[fl/fl] and *Tip60*[fl/fl] snRNA-seq datasets. Right, UMAP visualizations of infection status. *Npas4*[fl/fl]: 32,418 nuclei from 2 mice, 12,963 Cre-infected, 8,845 ΔCre-infected. *Tip60*[fl/fl]: 44,511 nuclei from 3 mice, 13,536 Cre-infected, 13,461 ΔCre-infected. **c**, Boxplots showing the average expression (Seurat log(e) normalized counts) of NPAS4 or TIP60 target genes (Methods) comparing Cre-infected and ΔCre-infected nuclei within each neuronal celltype. Boxplots show the median (line), inter-quartile range (IQR; box) and 1.5× IQR (whiskers), and notches indicate the median ± 1.58× IQR/sqrt($n$). \*\*\*$P < 5 × 10^{-10}$, \*\*$P < 5 × 10^{-4}$, \*$P < 0.01$. $P$ values were calculated using unpaired, two-tailed Wilcoxon rank-sum tests. Exact $P$ values and cell

numbers per cluster provided in the source data. **d**, Schematic of the recording configuration, showing the stimulating electrode in the centre of the stratum pyramidale to measure IPSCs onto neighbouring Cre-infected and uninfected CA1 pyramidal neurons in either *Npas4*[fl/fl] or *Tip60*[fl/fl] acute hippocampal slices. L, lacunosum; O, oriens; P, pyramidale; R, radiatum. **e**, Representative image (performed in triplicate) showing sparse Cre-mCherry expression in the CA1. **f,g**, Scatterplots of IPSC amplitudes recorded from pairs of neighbouring uninfected and Cre-infected cells in unstimulated (Unstim.; saline) or stimulated (Stim.; low-dose KA) *Npas4*[fl/fl] (**f**) or *Tip60*[fl/fl] (**g**) mice. Points represent an uninfected–Cre-infected pair. Squares represent the average ± s.e.m. Right insets, traces of uninfected or Cre-infected cells. *Npas4*[fl/fl]: stimulated: $n = 16$ pairs from 4 mice; unstimulated: $n = 12$ pairs from 5 mice. *Tip60*[fl/fl]: stimulated: $n = 11$ pairs from 2 mice; unstimulated: $n = 12$ pairs from 2 mice. \*$P = 3.73 × 10^{-3}$ for *Npas4*[fl/fl]; \*$P = 0.0209$ for *Tip60*[fl/fl]. Data are mean ± s.e.m. $P$ values were calculated using unpaired, two-tailed $t$-tests. The sequencer image in **a** is from BioRender (https://biorender.com).

by low-dose KA stimulation. By contrast, no differences between Cre-infected and uninfected cells were observed in saline-injected control animals (Fig. 2f,g and Extended Data Fig. 6f–i). Together, these experiments indicate that the NPAS4–NuA4 complex assembles in activated neurons to coordinate inducible gene transcription and to dynamically reorganize stimulated neuronal circuits in the brain.

## Inducible DNA breaks at NPAS4–NuA4 sites

The conserved function of NuA4 in DSB repair[21–26] prompted us to next investigate the possibility that NPAS4, through integration into the NuA4 complex, might play a previously unrecognized role in DNA damage control in neurons. Notably, stimulus-dependent gene induction in neurons has been suggested to result in DSBs, which are probably mediated by topoisomerase enzymes such as TOP2B, as paused RNA

Pol II is released into productive elongation[3,4,9,10,29]. However, the extent to which DSBs occur in response to neuronal activity in vivo and the mechanisms that neurons might use to repair these breaks and mitigate accumulating damage remain unclear. We proposed that the formation of a complex between NPAS4 and NuA4 may represent a mechanism by which neurons efficiently drive activity-induced transcriptional responses while simultaneously preserving genome stability downstream of neuronal activity (Fig. 3a). We sought to identify genomic loci that undergo both damage and repair in response to activity. We also examined whether these regions are targeted by NPAS4–NuA4 and tested whether perturbing NPAS4–NuA4 impairs DNA repair at these sites.

We used assay for transposase-accessible chromatin with sequencing (ATAC-seq) and CUT&RUN for the histone modification H3K27ac to identify the landscape of constitutive and activity-responsive genomic

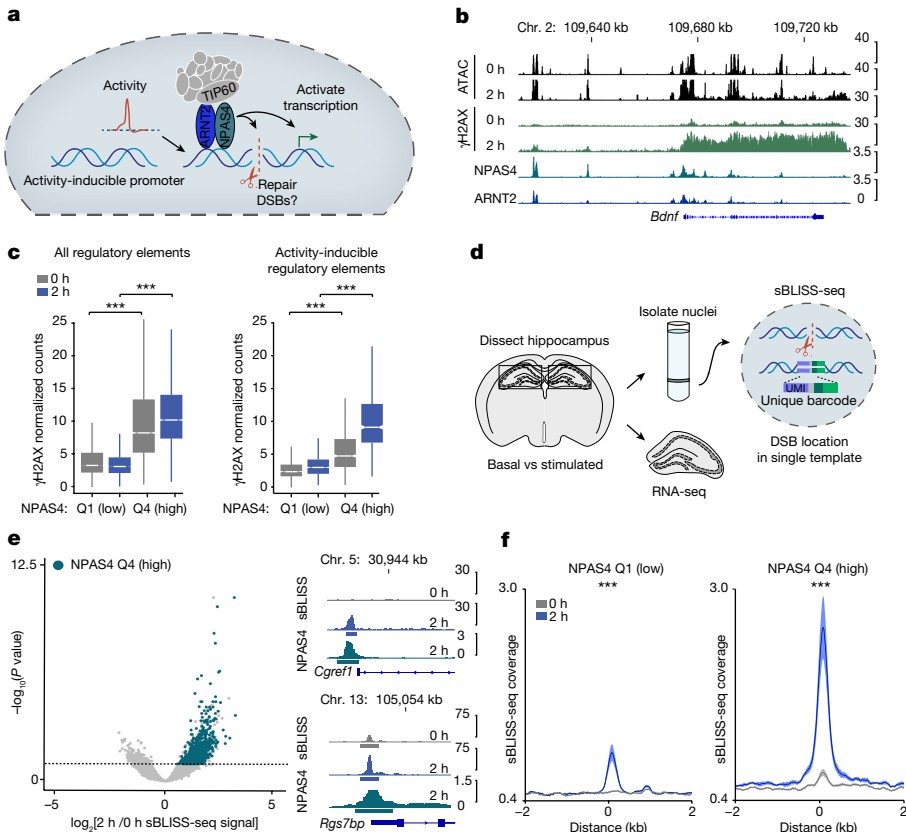

**Fig. 3 | NPAS4–NuA4-bound sites undergo recurrent DNA breaks in vivo.**
**a**, A model for the dual function of NPAS4–NuA4 in stimulating transcription and DNA repair in active neurons. **b**, IGV tracks displaying aggregate signals for ATAC-seq ($n = 3$), NPAS4 CUT&RUN ($n = 5$), ARNT2 CUT&RUN ($n = 2$) and γH2AX ChIP–seq ($n = 3$) at the activity-inducible gene *Bdnf*. $n = 3$–5 mice pooled per replicate. **c**, Boxplots of average γH2AX ChIP–seq normalized counts ($n = 3$ pools of 3–5 mice) at all regulatory elements (left) and activity-inducible regulatory elements (right), subset by quartiles of NPAS4 CUT&RUN signal (Methods). All regulatory elements: Q1 = 44,864 sites, Q4 = 7,378 sites. Activity-inducible elements: Q1 = 1,017 sites, Q4 = 764 sites. Boxplots show the median (line), IQR (box) and 1.5× IQR (whiskers), and notches indicate the median ± 1.58× IQR/sqrt($n$). ***$P < 2.2 \times 10^{-16}$. $P$ values were calculated using unpaired, two-tailed Wilcoxon rank-sum tests. **d**, Schematic of sBLISS-seq to map DSBs in brain nuclei. A tissue aliquot from each sBLISS-seq sample was kept for paired

RNA-seq analysis. **e**, Volcano plot depicting the DeSeq2 $\log_2$-transformed fold change versus $-\log_{10}$(Benjamini–Hochberg adjusted $P$ value) of sBLISS-seq signals between 0 and 2 h KA stimulation. Dotted line indicates adjusted $P < 0.1$. Elements with $\log_2$-transformed fold change > 0, adjusted $P < 0.1$ and that are within Q4 of NPAS4 IgG-normalized CUT&RUN signals are indicated in blue. Right inset, IGV tracks of aggregate sBLISS-seq ($n = 8$ individual mice) and NPAS4 CUT&RUN ($n = 5$ pools of 3–5 mice) at *Cgref1* and *Rgs7bp* promoters. Coloured bars represent statistically defined peaks. **f**, Aggregate plots showing sBLISS-seq coverage (fragment depth per bp per peak) at activity-inducible regulatory elements, subset by quartiles of NPAS4 binding. Signals are mean ± s.e.m. ($n = 8$ individual mice). ***$P < 2.2 \times 10^{-16}$. $P$ values were calculated using the average signal extracted in a 500 bp window around the element centre using unpaired, two-tailed Wilcoxon rank-sum tests.

regulatory elements in the hippocampus. We defined regulatory elements as regions of the genome with either an ATAC-seq or H3K27ac CUT&RUN peak, hereafter referred to as 'all regulatory elements'. We further defined 'activity-inducible regulatory elements' as sites that exhibited dynamic increases in the ATAC-seq signal (twofold increase; adjusted $P < 0.05$) and/or an increase in the H3K27ac CUT&RUN signal (1.5-fold increase; adjusted $P < 0.05$) after 2 hours of low dose KA stimulation (Extended Data Fig. 7a–c and Supplementary Table 4). Within this dataset, we classified NPAS4-bound elements as those that overlapped with our list of reproducible NPAS4 peaks. We first asked whether neuronal stimulation in vivo leads to chromatin signatures of DNA damage, especially at elements bound by NPAS4–NuA4. We performed ChIP–seq for the DNA damage-associated histone modification γH2AX (phosphorylated Ser139 on histone H2AX) in both unstimulated and stimulated neurons (Fig. 3b and Extended Data Fig. 7d,e). After stimulation, we observed increased γH2AX levels at NPAS4-bound sites, with the maximal signal and inducibility at sites in the highest quartile of NPAS4 binding (Fig. 3c). Notably, we observed a larger increase in γH2AX levels following stimulation at NPAS4-bound elements that fell within our landscape of activity-inducible elements (Fig. 3c, right).

Consequently, we focused primarily on these activity-inducible regulatory elements as they appeared to best capture the relationship between activity-induced transcription, NPAS4–NuA4 binding and DNA damage.

Next we directly mapped DSBs in vivo using suspension breaks in situ ligation and sequencing (sBLISS-seq), a sequencing-based method that identifies DSBs through the ligation of DNA sequencing adapters onto free DNA ends[30]. After validating the ability of sBLISS-seq to capture CRISPR–Cas9-induced DSBs (Extended Data Fig. 7f–h), we profiled the landscape of DSBs that occur in the adult hippocampus under basal conditions or following stimulation. In parallel, we performed RNA-seq on the same samples to examine how DSBs correlate with transcriptional dynamics (Fig. 3d). We first examined basic features of these DNA breaks across the genome at all time points. As previously reported[30], we observed maximal sBLISS-seq signals at the promoters of the most highly expressed genes (Extended Data Fig. 7i–k). There was also a significant correlation between the sBLISS-seq signal and the γH2AX ChIP–seq signal at 2 h after stimulation. This result provides an independent demonstration of damage to these sites (Extended Data Fig. 7l). Motifs enriched in statistically defined peaks of reproducible

sBLISS-seq signal included several activity-dependent transcription factors such as ATF1, EGR and NPAS4–ARNT, which indicates that a subset of these breaks is driven by transcriptional induction downstream of increased neuronal activity (Extended Data Fig. 7m).

We next examined how neuronal stimulation influences the landscape of DSBs. Although neuronal activity did not alter the overall distribution of sBLISS-seq signal (DSBs) across the genome (Extended Data Fig. 7i), we identified 1,581 regulatory elements (adjusted $P < 0.1$) that significantly increased in DSB signal at 2 h after stimulation (Fig. 3e). The activity-inducible elements displayed increased DSB signals after stimulation, which was in contrast to the non-inducible elements (Extended Data Fig. 7n). Moreover, 69% of the elements that had significantly increased DSB signal at 2 h after stimulation were in the top quartile of NPAS4 CUT&RUN signal, and the NPAS4-bound activity-inducible regulatory elements displayed the most significant increases in DSB signal after stimulation (Fig. 3e,f). We corroborated these findings using a complementary assay (END-seq)[31] to map DSBs in primary mouse cortical neurons at either 0 h or 2 h after stimulation with 55 mM KCl. To enhance sensitivity, we performed these assays in the presence of etoposide, which blocks the re-ligation of DSBs generated by topoisomerase enzymes. As reported previously[31], we observed enrichment of END-seq signal at CTCF-bound sites (Extended Data Fig. 8a). In line with our sBLISS-seq data from the hippocampus, the END-seq results revealed a stimulus-dependent increase in DNA breaks at NPAS4-bound sites in primary neurons (Extended Data Fig. 8b). Notably, the overall level and inducibility of both sBLISS-seq and END-seq signals were higher at sites that have a CUT&RUN peak for NPAS4 but not for the activity-dependent factor FOS than the converse (Extended Data Fig. 8c–i). Together, our independent measures of DNA damage by γH2AX ChIP–seq, sBLISS-seq and END-seq demonstrate that NPAS4 preferentially binds to sites that undergo activity-inducible DNA breaks in neurons.

## NPAS4–NuA4 sites undergo repair

We next asked whether these damaged elements, particularly those bound by NPAS4–NuA4, also undergo repair. To this end, we examined the levels of DSBs at a third time point, 10 h after KA stimulation, reasoning that a subset of sites may return towards the baseline DSB signal as activity-induced transcription subsides. We therefore used RNA-seq datasets collected from the same tissue as the sBLISS-seq datasets to identify samples in which activity-dependent transcription was returning to baseline. Principal component analysis (PCA) and hierarchical clustering of the RNA-seq data revealed two clusters of samples at the 10 h time point. One cluster displayed a significantly lower level of activity-inducible gene expression than at the 2 h time point (termed 'less active'), whereas the other 10 h cluster maintained higher levels of inducible genes (termed 'still active') (Fig. 4a and Extended Data Fig. 9a,b). PCA of the sBLISS-seq data also demonstrated that these less-active samples clustered closer to unstimulated samples (Extended Data Fig. 9c). Notably, we observed a separation between unstimulated and the less-active 10-h samples, which suggests that the less-active samples were returning to the baseline state rather than failing to initially stimulate. In the less-active 10-h samples, DSB signal at activity-inducible regulatory elements was significantly reduced relative to the 2 h time point. This result suggests that there was ongoing repair that resolves DSBs alongside declining transcriptional activity. This increase in the DNA break signal at 2 h after stimulation, coupled with a decrease at 10 h, was most pronounced at activity-inducible regulatory elements with the highest levels of NPAS4 binding (Fig. 4b and Extended Data Fig. 9d,e). We confirmed that NPAS4-bound sites undergo active DNA repair by examining published maps of DNA synthesis-dependent repair in human neurons (SAR-seq)[32]. Although this method does not exclusively detect DSB repair, it does capture the incorporation of new nucleotides into the repaired

DNA strand, which can occur with nonhomologous end joining[32]. We observed enrichment of SAR-seq signal at NPAS4-bound sites and higher levels of repair (SAR-seq signal)[32] at NPAS4-bound sites relative to FOS-bound sites (Extended Data Fig. 9f). To further investigate repair mechanisms at NPAS4-bound sites, we performed CUT&RUN for components of the MRE11–RAD50–NBS1 (MRN) complex. This complex is an early responder to sites of DSBs and plays an important part in processing broken DNA ends and initiating multiple repair pathways[33]. We examined both the MRE11 subunit, which mediates the removal of lethal topoisomerase cleavage products[34] and serves as a marker for DSB repair across the genome[35], and RAD50, which facilitates assembly of the complex and potentiates the endonuclease activity of MRE11 (ref. [33]). After validating the specificity of the MRE11 CUT&RUN signal using *Mre11* cKO mice (Extended Data Fig. 10a,b), we observed a strong activity-dependent co-localization of MRE11 and RAD50 at NPAS4-bound elements that was not driven by nonspecific IgG binding, histone acetylation or chromatin accessibility (Fig. 4c,d and Extended Data Figs. 10c–f and 11a,b). Notably, MRE11 was also present at NPAS4-bound elements in the absence of stimulation, which suggests that it may be retained following stimulation or that these sites are primed with other repair factors. These findings from multiple independent assays indicate that NPAS4–NuA4-bound sites are hotspots of DNA damage and repair in activated neurons.

## NPAS4–NuA4 disruption impairs DSB repair

Given its targeting to sites of damage, we next asked whether NPAS4–NuA4 stimulates repair at these sites in part by recruiting additional DSB repair machinery to the genome. By performing CUT&RUN for the NuA4 components EP400 and MRE11, we observed that depletion of NPAS4 resulted in a significant reduction in the binding of both proteins to NPAS4–NuA4 sites (Fig. 5a and Extended Data Fig. 11c–h). Although MRE11 was significantly reduced, it was not completely abolished, which suggests that MRE11 may have additional targeting mechanisms, such as direct interaction with free DSB ends or association with transcriptional machinery[33,36]. If NPAS4–NuA4 stimulates repair, the depletion of NPAS4–NuA4 subunits should result in increased DSBs in neurons. To test this prediction, we injected *Npas4*[fl/fl] mice with either Cre-expressing or ΔCre-expressing AAVs, isolated nuclei at 0, 2 or 10 h after stimulation and performed sBLISS-seq. After short-term depletion of NPAS4, the number of DSBs at activity-inducible regulatory elements increased at 2 and 10 h after stimulation. This result suggests that loss of NPAS4 renders neurons unable to efficiently repair these transcription-coupled breaks (Fig. 5b). We also observed an increase in the number of breaks before stimulation, which may reflect accumulated effects of dysregulated repair during previous stimulation (Fig. 5b and Extended Data Fig. 12a–d). The DSB increase in NPAS4-depleted nuclei at activity-inducible sites was not observed following Cre expression in wild-type neurons (Extended Data Fig. 12b). Of note, deletion of NPAS4 or the NuA4 component TIP60 also resulted in a significant increase in genome-wide DSBs that was not observed in wild-type mice and was not a secondary consequence of increased cell apoptosis in *Npas4* or *Tip60* cKO cells (Fig. 5c and Extended Data Fig. 12d–g). This increased level of DSBs across the genome may be due in part to dysregulated neuronal inhibition resulting from NPAS4–NuA4 disruption. However, we cannot rule out the possibility that expression of Cre, which has previously been reported to cause off-target DNA breaks on the genome[37], has a more pronounced effect in NPAS4 and TIP60 mutants owing to broadly dysregulated repair signalling.

## Age-dependent mutations at NPAS4 sites

A distinctive feature of neurons is their long postmitotic lifespans, which provide ample time for the accumulation of unresolved DNA breaks and mutations. We wondered whether the repeated activation of

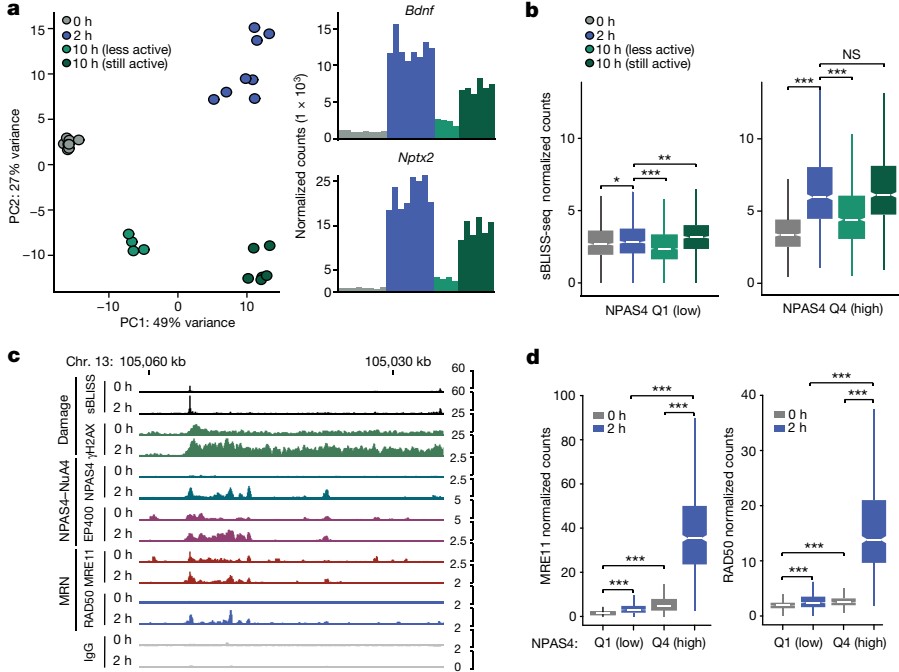

**Fig. 4 | NPAS4–NuA4-bound sites undergo DNA repair as inducible transcription subsides. a**, PCA of paired RNA-seq prepared from the same tissue used to generate the sBLISS-seq time course in hippocampus. Samples clustered according to the stimulation state, with 10 h stimulation samples separating into groups with either dampening or sustained transcriptional induction. DeSeq2-normalized counts of inducible genes *Bdnf* and *Nptx2* shown for each sample (0 h, *n* = 8; 2 h, *n* = 8; 10 h, *n* = 10). One mouse per replicate. **b**, Boxplots of average sBLISS-seq-normalized counts in 0 h (*n* = 8), 2 h (*n* = 8), 10 h less active (*n* = 4), and 10 h still active (*n* = 6) samples at activity-inducible regulatory elements, subset by quartiles of NPAS4 binding. One mouse per replicate. Boxplots show the median (line), IQR (box) and 1.5× IQR (whiskers), and notches indicate the median ± 1.58× IQR/sqrt(*n*). Q1 = 1,017 sites, Q4 = 764 sites. Q1, *\*P* = 0.03435, \*\**P* = 2.049 × 10$^{-8}$, \*\*\**P* = 3.057 × 10$^{-13}$; Q4, \*\*\**P* < 2.2 × 10$^{-16}$, not significant (NS) = 0.1742. *P* values were calculated using unpaired, two-tailed

Wilcoxon rank-sum tests. **c**, IGV tracks of sBLISS-seq, γH2AX ChIP–seq, NPAS4, EP400, MRE11 and RAD50 CUT&RUN signals. Data plotted show the aggregate signals from all replicates per factor. The *y* axis displays normalized coverage scaled for each factor at the chosen locus. CUT&RUN and γH2AX ChIP–seq, *n* = 3–5 mice pooled per replicate. sBLISS-seq, *n* = 1 mouse per replicate. Replicate numbers provided in Supplementary Table 2. **d**, Boxplots of average MRE11 and RAD50 normalized counts at activity-inducible regulatory elements, subset by quartiles of NPAS4 CUT&RUN signals. Q1 = 1,017 sites, Q4 = 764 sites. Boxplots show the median (line), IQR (box) and 1.5× IQR (whiskers), and notches indicate the median ± 1.58× IQR/sqrt(*n*). *n* = 3–5 mice pooled per replicate. Replicate numbers provided in Supplementary Table 2. \*\*\**P* < 2.2 × 10$^{-16}$. *P* values were calculated using unpaired, two-tailed Wilcoxon rank-sum tests.

NPAS4-bound elements could predispose these sites to increased mutational load during ageing. To assess the mutational load at NPAS4-bound elements, we used fluorescence-activated cell sorting (FACS) to isolate NeuN-expressing nuclei from the hippocampus of young (3 months old), middle-aged (12 months old) and old (23–27 months old) mice and extracted neuronal DNA (Fig. 5d and Extended Data Fig. 13a,b). We then performed targeted amplicon sequencing for sensitive mutation detection[38] at NPAS4-bound elements compared with negative control elements (that is, sites with little to no NPAS4 binding, termed NPAS4-unbound). Incorporation of unique molecular identifiers (UMIs) during PCR amplification enabled the detection of mutations attributable to a single allele template while excluding base changes arising from sequencing errors[38] (Extended Data Fig. 13c). We first validated the ability of this technique to detect mutations introduced at CRISPR–Cas9-induced cut sites in neurons (Extended Data Fig. 13d).

We next defined a panel of fragile sites that undergo routine breaks or repair (assessed by levels of γH2AX and/or MRE11 binding) and are bound by NPAS4–NuA4. We compared this dataset with a set of elements that do not share these signatures of damage and are not bound by NPAS4-NuA4, but are matched for levels of inducible H3K27ac, inducible chromatin accessibility and AT/GC content (Extended Data Fig. 13e,f). To account for differences between age groups in the representation of target sites amplified in our assay, each of which may display a different baseline rate of mutational frequency, we normalized the mutation rate of each site to the median mutation frequency of

that same site in the young animals. Notably, sites not bound by NPAS4 accumulated mutations with age, showing an approximately twofold increase by 12 months. By contrast, our panel of NPAS4-bound sites did not appear to increase in mutational frequency with age (Fig. 5e). These data suggest that NPAS4–NuA4-bound sites are relatively protected against the additional mutations that accrue over the course of organismal ageing. These differences were not attributable to an unusually high rate of mutations in unbound sites, as we observed a higher rate of both single nucleotide variant and insertion and deletion events at NPAS4-bound sites relative to non-bound sites in young animals (Extended Data Fig. 13g). These results further corroborate that NPAS4–NuA4 is targeting fragile sites and raise the possibility that the NPAS4–NuA4 complex may be required as an additional layer of protection for these recurrently broken sites.

## NPAS4–NuA4 disruption decreases lifespan

Our data showed that disruption of NPAS4–NuA4 function leads to dysregulation of activity-dependent gene expression, increased DNA breaks at activity-regulated promoters and enhancers, impaired localization of protective repair machinery and defects in pyramidal neuron somatic inhibition. Therefore, we reasoned that disruption of the NPAS4–NuA4 complex would ultimately have widespread, long-term consequences as animals age. These changes could include deleterious effects on genome integrity, excitatory/inhibitory balance and

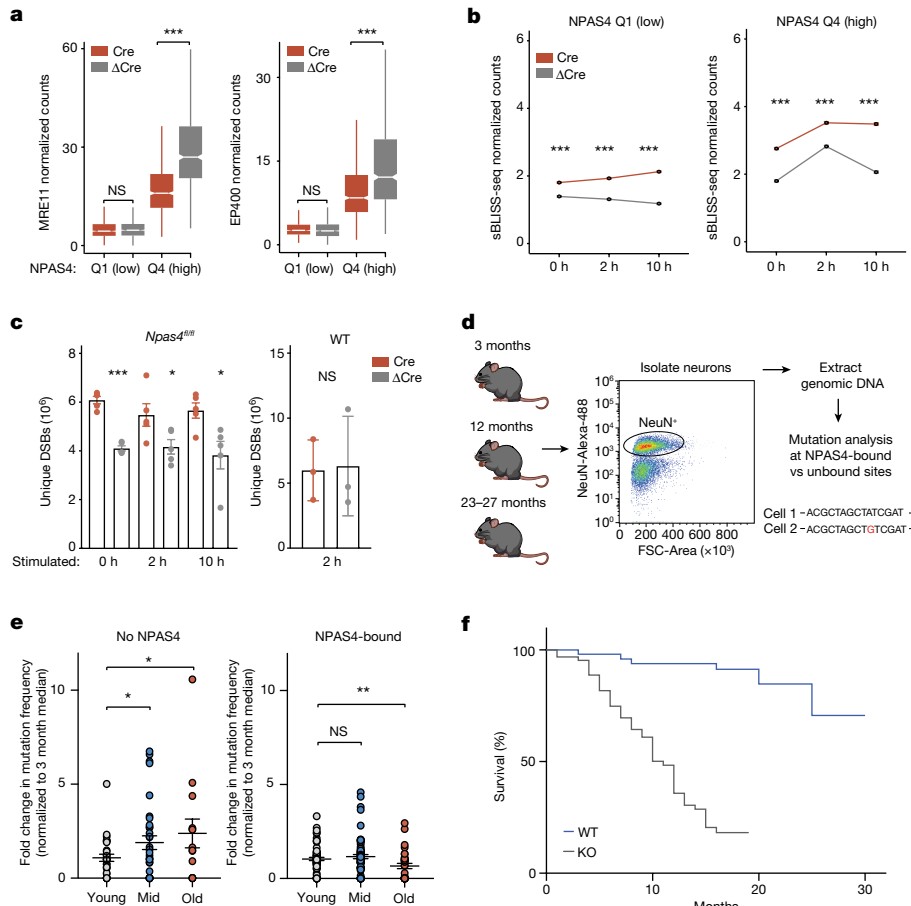

**Fig. 5 | NPAS4–NuA4 disruption impairs genome stability and reduces lifespan. a**, Boxplots of average MRE11 or EP400 CUT&RUN normalized counts in Cre-infected or ΔCre-infected hippocampi of *Npas4^fl/fl* mice at activity-inducible sites, subset by quartiles of NPAS4 binding. Q1 = 1,017 sites, Q4 = 764 sites. Boxplots show the median (line), IQR (box) and 1.5× IQR (whiskers), and notches indicate the median ± 1.58× IQR/sqrt(*n*). *n* = 2–3 mice pooled per replicate. Replicate numbers provided in Supplementary Table 2. MRE11 Q1, *P* = 0.2453; Q4, ****P* < 2.2 × 10^−16. EP400 Q1, *P* = 0.9962; Q4, ****P* < 2.2 × 10^−16. *P* values were calculated using unpaired, one-tailed Wilcoxon rank-sum tests (ΔCre > Cre). **b**, Line plot depicting average ± s.e.m. of sBLISS-seq normalized counts in Cre-infected or ΔCre-infected hippocampi of *Npas4^fl/fl* mice, subset by quartiles of NPAS4 binding. Q1 = 1,017 sites, Q4 = 764 sites. Data from *n* = 5 each for Cre-infected and ΔCre-infected mice. ****P* < 2.2 × 10^−16. *P* values were calculated using unpaired, one-tailed Wilcoxon rank-sum tests (Cre > ΔCre). See Extended Data Fig. 12a for boxplot distribution of data. **c**, Genome-wide breaks in hippocampal nuclei isolated from *Npas4^fl/fl* (*n* = 5) or wild-type (*n* = 3)

mice infected with Cre or ΔCre virus. Data are mean ± s.e.m. *Npas4^fl/fl*: 0 h, *P* = 5.24 × 10^−6; 2 h, *P* = 0.044; 10 h, *P* = 0.022. Wild-type: 2 h, *P* = 0.906. *P* values were calculated using two-tailed, unpaired *t*-tests. **d**, Collection of hippocampal neuronal nuclei across lifespan. **e**, Normalized mutation frequency at NPAS4-bound and unbound sites in wild-type mice. Mutation frequency (base changes and insertions and deletions) with age was calculated per site and normalized to the median frequency for that site in young animals. Points represent the normalized rate for one site sampled from one mouse. Data are mean ± s.e.m. Data are from *n* = 4 (young), 4 (middle aged) and 3 (old) mice. No NPAS4 sites: mid–young, **P* = 0.03; old–young, **P* = 0.016. NPAS4-bound sites: mid–young, NS = 0.18; old–young, ***P* = 0.0095. *P* values calculated using one-tailed, unpaired *t*-tests. **f**, Survival of *Npas4* wild-type (*n* = 53; 28 females, 25 males) and *Npas4* knockout (*n* = 64; 37 females, 27 males) mice, showing abbreviated lifespan of *Npas4* knockout mice. *P* = 4.09 × 10^−13 by a two-tailed Mantel–Cox test, *P* = 3.63 × 10^−11 by a two-tailed Gehan–Breslow–Wilcoxon test.

organismal lifespan. Notably, we observed that loss of this neuronal factor substantially shortened the lifespan of both male and female mice, leading to a median lifespan of 12 and 11 months, respectively (Fig. 5f). This result corroborates results from an independent *Npas4* knockout line[39] and demonstrates a clear effect on longevity for mice of both sexes (Extended Data Fig. 13h,i). That the reduced lifespan of the germline *Npas4* knockout mice is due specifically to the loss of NPAS4 in the brain is buttressed by our snRNA-seq data demonstrating that NPAS4 is highly specific to neurons (Extended Data Fig. 2f). Moreover, mice with *Npas4* deleted in forebrain *Camk2a*-expressing excitatory neurons also had reduced lifespan (Extended Data Fig. 13j). However, we cannot exclude the possibility that transient expression of *Npas4* in non-neuronal cells contributes to this longevity phenotype. Together, these data raise the possibility that the protective role of NPAS4–NuA4 in facilitating DSB repair helps to ensure the long-term

fidelity of transcriptional responses to stimulation and proper inhibitory control in the brain that may be crucial for normal lifespan.

## Discussion

The extent to which different cell types in the body specialize their DNA repair mechanisms is poorly understood. Emerging evidence suggests that neurons continuously repair DNA at select locations within the genome[32,40], yet the mechanisms for preferential targeting of repair remained obscure. This may be due in part to the heterogeneity of recurrently damaged sites across the vast number of neuronal cell types resulting from their diverse activity patterns and cell-type-specific transcriptional programmes. Using a combination of new mouse models, biochemistry, single-cell genomics and electrophysiology, we identified a specialized neuronal form of the NuA4 complex that assembles

around NPAS4 in activated neurons to regulate cell-type-specific inducible transcription and suppress DNA damage. Our findings suggest that neurons have evolved a specific chromatin regulatory mechanism that couples synaptic activity to genome preservation. This mechanism may reduce accumulating damage at pivotal regulatory elements in each neuronal cell type and preserve the ability to mount appropriate responses to environmental cues.

The mechanisms that lead to both the formation and repair of DSBs at NPAS4-bound sites remain to be fully elucidated. As previously suggested[4], these breaks may arise from pre-bound topoisomerase enzymes that are post-translationally modified downstream of neuronal activity. In addition, DSBs may form in the process of resolving DNA–RNA hybrids (R-loops) or releasing stalled transcription complexes that occur with a rapid induction of previously quiescent regulatory elements. Notably, NuA4 has been reported to bind R-loops[41], which suggests that R-loop formation may contribute to both DNA damage and repair at activity-dependent regulatory elements. Studies outside the nervous system have suggested that RNA itself could facilitate repair of these transcribed regions by serving as a template in place of a sister chromatid in postmitotic cells[42,43]. Although it is probable that canonical nonhomologous end joining pathways mediate much of the DSB repair in activated neurons, it is possible that NPAS4–NuA4 engages multiple repair pathways. Future studies that probe the precise mechanisms that neurons use to repair activity-induced damage, including those mediated by NPAS4–NuA4, will be important areas of investigation.

This work provides an example of specialized chromatin machinery in the brain and adds to the repertoire of neuronal epigenomic features that are frequently dysregulated in both neurodevelopmental and neurodegenerative disorders. Several components of NPAS4–NuA4 (*Ep400*, *Trrap*, *Actl6b* and *Tip60*) are mutated in neurodevelopmental and autism spectrum disorders[44–46]. Our discovery of a link between neuronal activity and DNA repair mediated by NPAS4–NuA4 suggests that damage at activity-dependent regulatory elements may be a source of neuronal dysfunction in these disorders.

Loss of genome integrity is a hallmark of ageing across organisms[12,13], and the ability to efficiently repair DSBs has been linked to the evolution of longer lifespans in mammals[47]. However, much remains unknown about how neuronal activity influences genome stability with age and contributes to cellular and organismal longevity. Sustaining neuronal vitality over time appears to require careful balancing of the proper ratio of excitation and inhibition[48]. In addition to the role of NPAS4 in DNA repair, the compounding loss of inhibition in *Npas4* knockout mice might contribute to their shortened lifespan. Over time, impaired NPAS4–NuA4 signalling may disrupt a key regulatory feedback loop in which deletion of NPAS4 impairs the expression of genes that mediate recruitment of somatic inhibition, which in turn leads to excessive excitation that further threatens genome stability. Future experiments that decouple the roles of NPAS4 in transcription and repair are needed to understand the relative contribution of these two processes in neuronal and organismal longevity. Given that the neuronal-specific expression pattern of NPAS4–NuA4 is conserved in the human brain, in which neurons are subject to recurrent activity-induced DNA breaks over many decades, the NPAS4–NuA4 signalling axis may also serve as an important entry point to understanding the breakdown of cognitive and sensory processing in ageing and neurodegenerative diseases in humans.

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

## Methods

### Experimental models

**Mouse models.** Animal use was approved and overseen by the Harvard University Institutional Animal Care and Use Committee and the Harvard Center for Comparative Medicine. The following mouse lines were used: wild-type C57/BL6 (Jackson Laboratory, 000664); *Npas4*[fl/fl14]; *Npas4*-knockout[14]; *Tip60*[fl/fl 49]; *Npas4–Flag-HA* (this manuscript); *Arnt2–Flag-HA* (this manuscript), *Tip60-H3F*[50]; *Mre11*[fl/fl51,52]; *B6;129-Gt(ROSA)26Sor<tm5(CAG-Sun1/sfGFP)Nat>/J* (Jackson Laboratory, 021039)[53]; and *B6.Cg-Tg(Camk2a-cre)T29-1Stl/J* (Jackson Laboratory, 005329)[54]. Mice were housed in a temperature and humidity-controlled environment using ventilated microisolator cages. Mice were kept under a standard 12 h light–dark cycle, with food and water provided ad libitum. Male and female littermate mice were used in similar proportions and divided between control and experimental groups for all experiments conducted. No statistical methods were used to predetermine sample sizes. For biochemistry and genomic experiments, animals were collected at 4–6 weeks of age throughout the manuscript. For physiology experiments, animals were dissected and patched at postnatal day 24 (P24) to P28. For ageing experiments, animals were collected at 3-4 months, 12 months and 23–27 months of age. Details of animal age and sex are detailed within each protocol.

**Generation of *Npas4–Flag-HA* and *Arnt2–Flag-HA* knock-in mouse lines.** Zygote injections were performed at the Harvard Genome Modification Facility in accordance with their practices and guidelines. Guide RNA sequences in proximity to the 3′ end of *Npas4* and *Arnt2* loci were chosen based on predicted cutting efficiency and low off-target effects. Guide RNA sequences for NPAS4 (5′-cacagacttattcaaa acgt-3′) and ARNT2 (5′-gagtagcttcaggcaaagcc-3′) were cloned into a FUGW backbone containing the guide sequence upstream of a tracrRNA (5′-gttttagagctagaaatagcaagttaaaataaggctagtccgttatca acttgaaaaagtggcaccgagtcggtggtgcttttttt-3′). To generate templates for in vitro transcription, PCR was performed using these FUGW plasmids to incorporate a T7 promoter using the following primers: *Npas4* forward (F): 5′-taatacgactcactataggcacagacttattcaaaa cgtgttttagag-3′; *Npas4* reverse (R): 5′-aaaaaaagcaccaccgactcgg-3′; *Arnt2* F: 5′-taatacgactcactataggagtagcttcaggcaaagcc-3′; and *Arnt2* R: 5′-aaaaaaagcaccaccgactcgg-3′.

PCR products were verified on DNA gels to ensure a single product and were purified using Qiagen's PCR clean-up kit. In vitro transcription (IVT) was performed using 400 ng of purified RNA template for 16 h at 37 °C using a Megascript Short Transcript IVT kit (Thermo Fisher, AM1354). RNA clean-up was performed using a MEGAclear Transcription Clean-Up kit (Ambion, AM190). Finally, RNA size and quality were assessed by running 200 ng of purified product on 10% TBE gels. The HDR template was ordered as a 200 bp, single-stranded Ultramer from IDT, with 75 bp of homology on either side of the cut site. CRISPR guide RNA, Cas9 RNA and Ultramer donor template were injected into zygotes of mixed background DBA/C57BL6 mice according to the guidelines and procedures of the Harvard Genome Modification facility. DNA from founders and F[1] progeny were screened using Sanger sequencing to ensure proper insertion of the tag and that no mutations occurred in the flanking DNA regions. *Npas4–Flag-HA* and *Arnt2–Flag-HA* mice have been subsequently backcrossed to C57BL6/J mice and are available upon request.

**Mouse neuron culture.** Embryonic cortices (embryonic day 16.5–17) were dissected from 5–6 embryos of mixed sex and pooled for a single replicate culture. Papain (Sigma-Aldrich, 10108014001) was used to dissociate tissue through a 10 min incubation at 37 °C. Digestion was terminated with the addition of ovomucoid (trypsin inhibitor, Worthington). Digested tissue was gently triturated through a P1000 pipette to release cells and passed through a 40 μm filter. Neurons were plated onto cell culture dishes pre-coated overnight with poly-D-lysine (20 μg ml$^{-1}$) and laminin (4 μg ml$^{-1}$). Neurons were grown in neurobasal medium (Gibco) containing B27 supplement (2%), penicillin–streptomycin (50 U ml$^{-1}$ penicillin and 50 U ml$^{-1}$ streptomycin) and glutaMAX (1 mM). Neurons were grown in incubators maintained at 37 °C with a $CO_2$ concentration of 5%. Neurons were collected for all experiments at 7 days in vitro (DIV), and fresh medium was added at 3–4 DIV (50% total volume) unless otherwise indicated. Independent replicates were generated by preparing primary cultures from pools of embryos dissected on different days and maintained in culture for 7 DIV. For RNA-seq experiments, cells were plated at a density of 156,250 cells per cm$^2$ (1.5 million neurons per well) in 6-well culture dishes. For luciferase assays, cells were plated at a density of 210,526 cm$^{-2}$ (400,000 cells per well) in 24-well culture dishes. To depolarize neurons, neurons were treated with 55 mM KCl. For treatment with different stimuli (Extended Data Fig. 1d), 7 DIV neurons were treated with 50 μM NMDA or 50 ng ml$^{-1}$ BDNF. To chelate calcium, neurons were pretreated with 5 mM EGTA 15 min before stimulation with 55 mM KCl solution.

### Experimental procedures

**Collection time points.** For experiments in which seizures were induced, KA (5–8 mg kg$^{-1}$ (electrophysiology) or 12–15 mg kg$^{-1}$ (gene expression, CUT&RUN, sBLISS-seq and immunoprecipitations) (Sigma-Aldrich, K0250) was intraperitoneally injected. For genomic-level analyses, mice were euthanized 2, 6 or 10 h after injection, as indicated, based on previous work showing that immediate-early gene transcription factors and late-response gene programmes are induced at 2 and 6 h after stimulus onset, respectively[1,14,28]. The unstimulated or basal time point indicates mice injected with saline control and were dissected and processed in parallel with the stimulated tissue. For electrophysiological experiments in CA1 pyramidal neurons, mice were killed 18–24 h after injection with low levels of KA to allow sufficient time for the expression of NPAS4, its associated activity-dependent target genes and the execution of potential synaptic regulation. For proteomics analysis, mice were killed 3 h after injection to allow sufficient time for NPAS4 to be produced and for the assembly of its associated protein complex.

**Stereotaxically guided surgery.** For genomic assays, including snRNA-seq, CUT&RUN and sBLISS-seq, P21–P24 *Tip60*[fl/fl] and *Npas4*[fl/fl] mice were anaesthetized by isoflurane inhalation (3–5% for induction, 1–3% maintenance) and positioned within a stereotaxic frame (Kopf) in which the temperature of the mouse was maintained at around 37 °C with a heat pad. All surgeries were performed according to protocols approved by the Harvard University Standing Committee on Animal Care and were in accordance with federal guidelines. Fur around the scalp was removed using a shaver and sterilized with three alternating washes of betadine and 70% ethanol. For genomic assays requiring broad infection of the hippocampus, a burr hole was drilled through the skull above the hippocampus, and a glass pipette filled with AAV was lowered to a central region of the hippocampus to enable broad infection of the various subregions (medial–lateral: ±2.5; anterior–posterior: −2.5; dorsal–ventral: −2.5). AAV (1,000 nl, diluted to $1.0 \times 10^{12}$ genome copies per ml) was injected at 150 nl min$^{-1}$, and the pipette was left in place for 5 min after completion of virus infusion to allow for viral spreading. All animals were given postoperative analgesic (buprenophine slow-release formulation, 1 mg kg$^{-1}$) in accordance with Institutional Animal Care and Use Committee protocols. For electrophysiological assays requiring sparse infection of the CA1 region of the hippocampus, P14 mice were injected as described above, but the glass needle tip was positioned at different coordinates to limit infection to the CA1 (medial–lateral: ±3.4; anterior–posterior: −2.9; dorsal–ventral: −2.8), and a lower titre of virus was injected ($1.0 \times 10^{10}$ genome copies per ml).

**Acute slice preparation.** Transverse hippocampal slices were prepared from $Npas4^{fl/fl}$ and $Tip60^{fl/fl}$ C57BL/6 mice aged P21–P28. Isoflurane was used to anaesthetize the animals, and the brain was quickly removed, bisected along the midline and placed in ice-cold choline-containing artificial cerebrospinal fluid (choline-ACSF) containing (in mM): 110 choline-Cl, 25 NaHCO$_3$, 1.25 NaH$_2$PO$_4$, 2.5 KCl, 7 MgCl$_2$, 25 glucose, 0.5 CaCl$_2$, 11.6 ascorbic acid and 3.1 pyruvic acid. Choline-ACSF was equilibrated with 95% O$_2$ and 5% CO$_2$. Both cerebral hemispheres were transferred to a slicing chamber containing ice-cold choline-ACSF, and 300-µm-thick slices were cut using a Leica VT1200S vibratome. Slices were transferred to a holding chamber filled with ACSF saturated with 95% O$_2$ and 5% CO$_2$ and containing (in mM): 127 NaCl, 25 NaHCO$_3$, 1.25 NaH$_2$PO$_4$, 2.5 KCl, 2 CaCl$_2$, 1 MgCl$_2$ and 25 glucose. Slices were incubated at 34 °C for 30 min, then equilibrated to room temperature for 20 min before recordings. All recordings were performed at room temperature and were completed within 6 h of slice preparation. Epifluorescence was used to identify slices with sparse infection of the CA1 pyramidal layer with Cre-mCherry, and slices with >30% infected CA1 pyramidal neurons were not used for recordings. Slices were also discarded if Cre-mCherry expression was observed in the CA3 or dentate gyrus regions.

**Electrophysiology.** CA1 pyramidal neurons were visualized with infrared differential interference contrast microscopy to perform whole-cell voltage-clamp recordings. Neighbouring uninfected and Cre-mCherry-infected neurons were identified using epifluorescence driven by a light-emitting diode. Neurons were held at −70 mV for all experiments. Recording pipettes made from borosilicate glass (open resistance between 2 and 4 MΩ) were filled with an internal solution containing (in mM): 147 CsCl, 5 Na$_2$-phosphocreatine, 10 HEPES, 2 MgATP, 0.3 Na$_2$GTP, 2 EGTA and 5 QX-314 (Sigma-Aldrich). Internal solution was prepared in a single batch, with osmolarity adjusted to 290 mOsm with double-distilled water and pH adjusted to 7.3 with CsOH, and was stored at −20 °C. To record IPSCs, inhibitory currents were pharmacologically isolated through bath application of 10 µM (R)-CPP (Tocris Bioscience) and 10 µM NBQX disodium salt (Tocris Bioscience). Extracellular stimulation of perisomatic inhibitory axons was achieved using a concentric bipolar electrode (FHC) placed within the centre of the stratum pyramidale and within 100–200 µm laterally of the pair of voltage-clamped cells. The stimulus strength used was the minimum stimulation required to generate a reliable IPSC in both neurons.

**Electrophysiology data acquisition and analysis.** Recordings were made using a Multiclamp 700B amplifier (Axon Instruments), filtered at 3 kHz and sampled at 10 kHz. Data analysis was performed with custom software written in MatLab. Experiments were discarded if the holding current was less than −600 pA or if the series resistance was greater than 25 MΩ. Series resistance across simultaneously recorded cells was within 25% for each pair. Recordings were performed at room temperature (19–21 °C). The amplitude of IPSCs was calculated by averaging the amplitude 0.5 ms before to 2 ms after the peak of the current. The unsigned magnitude of synaptic currents are shown for clarity.

**Histology.** Males and females were used in equal proportions. Mice were anaesthetized by intraperitoneal injection of 10 mg ml$^{-1}$ ketamine and 1 mg ml$^{-1}$ xylazine. Once anaesthetized, mice were transcardially perfused with at least 10 ml of ice-cold PBS, followed by at least 20 ml of 4% paraformaldehyde (PFA) in PBS. Brains were removed and placed in 4% PFA at 4 °C for 24 h, followed by three successive washes in cold PBS. Brains were then transferred to 30% sucrose for cryoprotection, and were incubated at 4 °C until the tissue equilibrated and sank. The brains were then embedded in NEG-50 frozen section medium and

stored at −80 °C until sectioned (30 µm thick) on a cryostat (Leica CM1950). Sections were stored in PBS at 4 °C until further use. For immunohistochemistry, sections were permeabilized and blocked by incubating in PBS containing 0.1% Triton X-100 and 5% normal goat serum (blocking solution) for 1 h at room temperature. Staining in primary antibody (dilutions listed below) was performed overnight (16 h) at 4 °C with gentle shaking. Sections were then washed three times for 5 min each in PBS containing 0.1% Triton X-100. Secondary antibody staining was performed by diluting an Alexa Fluor dye-conjugated secondary antibody (Life Technologies; rat Alexa Fluor 555 (A21434), rabbit Alexa Fluor 647 (A31573); 1:250) and incubating slices for 2 h at room temperature. Sections were then washed again three times for 5 min each in PBS containing 0.1% Triton X-100, and then mounted in DAPI Fluoromount-G (Southern Biotech) and imaged on a slide-scanning microscope (Olympus VS120, VS ASW-FL). Antibodies were diluted in blocking solution as follows: rat anti-HA (Sigma-Aldrich, ROAHAHA; 1:250); rabbit anti-NPAS4 (in house; 1:1,000)[14]; rabbit anti-ARNT2 (in house; 1:1,000)[28]; rabbit anti-KAT5 (TIP60) (Proteintech, 10827-1-AP; 1:250); rabbit anti-cleaved caspase-3 (Cell Signaling Technology, 9664S; 1:1,000).

**Immunoprecipitation of NPAS4–NuA4 complexes.** To isolate protein complexes associated with NPAS4, 8–12 hippocampi were isolated from 6-week-old $Npas4–Flag-HA$ and $Arnt2–Flag-HA$ mice injected with 15 mg kg$^{-1}$ KA (Sigma-Aldrich, K0250) to induce high levels of NPAS4 expression. Males and females were used in equal proportions in pooled replicate samples. Wild-type mice of the same age and sex distributions were processed in parallel to serve as controls. Hippocampal regions were dissected 3 h after KA injection to allow sufficient time for NPAS4 expression and assembly into potential complexes. Hippocampi were collected and dounced 20× in 10 ml of NE1 buffer (20 mM HEPES pH 7.9, 10 mM KCl, 3 mM MgCl$_2$, 0.1% Triton and 0.1 mM EDTA) containing protease inhibitor cocktail (Roche) and phosphatase inhibitor cocktails 2 and 3 (Sigma-Aldrich, P5726 and P0044). Nuclei were released through 10 min of incubation in NE1 buffer and then pelleted by gentle centrifugation (1,000–1,500$g$). Nuclear pellets were resuspended in 1 packed nuclear volume of NE1 buffer. To facilitate release of chromatin-associated proteins, nuclei were incubated for 30 min at 4 °C with benzonase endonuclease (Sigma-Aldrich, E1014, >25 KU (1 µl per 10 million nuclei)) followed by the addition of NaCl to a final concentration of 300 mM. Following high-speed centrifugation to remove insoluble material, lysates were diluted to achieve a final salt concentration of 200 mM. To preclear nonspecific interactions, lysates were incubated for 30 min at 4 °C with 50 µl of ms-IgG-coated agarose beads (Sigma-Aldrich, A0919). Following preclear, lysates were incubated for 1.5 h with 60 µl of anti-M2-Flag resin (Sigma-Aldrich, A2220) per 1 ml of diluted lysate. Samples were washed 4× in NE1 buffer containing 250 mM NaCl for 5 min with rotation at 4 °C. NPAS4-interacting or ARNT2-interacting proteins were competitively eluted off M2 resin by incubation in 50–100 µl of 500 µg ml$^{-1}$ 3× Flag peptide (Sigma-Aldrich, F4799) diluted in NE1 buffer for 30 min at room temperature. For mass spectrometry analysis, eluted proteins were precipitated with trichloroacetic acid. Replicates shown in Fig. 1c consisted of 3 independent pools of 8–12 mice collected from wild-type or $Npas4–FH$ lines and processed on separate days. Validation immunoprecipitation assays using either $Npas4–Flag-HA$ mice or $Tip60–H3F$ mice followed by immunoblotting analysis were conducted under the same conditions as described above. For validation experiments in mouse visual cortex following light stimulation, 20 hippocampi were isolated from the V1 cortices of $Npas4–Flag-HA$ or wild-type controls. Mice were housed in the dark for 1 week followed by 2 h of light stimulation. Data are shown in Extended Data Fig. 1f.

To isolate high molecular weight complexes containing intact NPAS4–NuA4 for mass spectrometry, 8–10 6-week-old $Npas4–Flag-HA$, $Tip60–H3F$ and wild-type control mice were injected with KA and

dissected 2 h after injection. Males and females were used in equal proportions in pooled replicate samples. Replicates consisted of independent pools of 8–10 mice run on independent gradients and days. Two replicates were performed. Whole forebrain, minus hippocampus and striatum, were minced and transferred NE1 buffer, and nuclear lysates were prepared as described above. Approximately 6 ml gradients were prepared using a BIOCOMP Gradient Master 108 (Science Services) using a 10–40% long glycerol gradient for the SW41 Beckman rotor. Sample concentrations were normalized to around 2 mg ml$^{-1}$, and 1 ml was loaded on top of the 10–40% glycerol gradient. Samples were centrifuged for 16 h at 37,000 r.p.m. at 4 °C. One millilitre fractions were collected at 4 °C. Fractions were run on SDS–PAGE gels to assess which fractions contained components of the NuA4 complex. Immunoprecipitation assays were then performed from fractions 1–3, which contained the majority of TRRAP and EP400 in the lysates. To preclear nonspecific interactions from fractions, lysates were incubated for 30 min at 4 °C with 100 µl of ms-IgG-coated agarose beads (Sigma-Aldrich, A0919). Following preclear, lysates were incubated for 1.5 h with 100 µl of anti-M2-Flag resin (Sigma-Aldrich, A2220) per 1 ml fraction. Samples were washed 4× in NE1 containing 250 mM NaCl for 5 min with rotation at 4 °C. Interacting proteins were eluted off M2 resin by incubation in 500 µg ml$^{-1}$ 3× Flag peptide (Sigma-Aldrich, F4799). Eluates from immunoprecipitation assays performed on fractions 1–3 were pooled to have sufficient material for mass spectrometry, and eluted proteins were precipitated with trichloroacetic acid.

**Immunoprecipitation of NPAS4 truncation mutants in HEK293T cells.** Plasmids (FUW with UbC promoter) containing the sequence for Flag-HA-tagged FOS, JUN, EGR1 and NPAS4 or various truncated forms of NPAS4 (Supplementary Table 1) were expressed in HEK293T cells by CaCl$_2$–BBS transfection (2 µg DNA, 6 × 10$^6$ cells). HEK293T cells were obtained from Thermo Fisher Scientific (50188404FP). HEK293T cells were not authenticated or tested for mycoplasma. With the exception of nuclear GFP-FH control samples, the plasmids expressed GFP–IRES-NPAS4 to visualize transfection efficiency. Flag-HA tags were appended to the amino terminus of NPAS4 to minimize differences in tag accessibility across NPAS4 variants truncated at the C-terminal end. Cells were collected by gently scraping into ice-cold PBS containing protease inhibitor cocktail (Roche) and were pelleted by gentle centrifugation (2,000$g$). Nuclei were isolated, and Flag immunoprecipitation was performed as described above. A second immunoprecipitation step using HA was performed by collecting the Flag peptide eluate, increasing the volume to 1 ml in NE1 buffer, adding 50 µl anti-HA resin (Santa Cruz Biotechnology, SC-7392 AC) and gently rotating at 4 °C for 1.5 h. Samples were washed 4× in NE1 buffer containing 250 mM NaCl for 5 min with rotation at 4 °C. Proteins were eluted by incubating resin with HA peptide (Thermo Fisher, 26184) diluted in NE1 buffer for 30 min at room temperature. Eluted proteins were precipitated with trichloroacetic acid for mass spectrometry. Replicates consisted of independently transfected cultures and immunoprecipitation assays performed on separate days.

**Size-exclusion chromatography and blue native gels.** Whole-cell lysates (2% Triton X-100, 50 mM Tris, 150 mM NaCl and 1 mM EDTA) from hippocampal tissue or from hippocampal and striatal tissue were fractionated using a 400HR 1 fractionator set to 0.5 ml min$^{-1}$. Buffers for the column and fractionator consisted of 20 mM Tris-HCl pH 7.4, 100 mM NaCl, 1 mM EDTA, 3 mM MgCl$_2$ and 0.02% Triton. Seventeen fractions consisting of 6 ml each were collected, and fractions 8–16 were loaded onto Native Novex NuPAGE 4–16% gels (Thermo Fisher, BN1002BOX) to estimate the approximate size of NPAS4-containing protein complexes. Native page marker (1 µl; Invitrogen, LC0725) was used to estimate sizes of the complexes. Proteins were transferred onto PVDF membranes and immunoblotted for NPAS4 (1:1,000) using in-house antibodies.

**Immunoblotting.** Whole-cell or nuclear extracts from primary neurons or brain tissue were resolved on 3–8% Tris-acetate gels (Thermo Fisher, EA0375BOX) or 10% Tris-glycine gels and transferred to nitrocellulose membranes. Membranes were incubated overnight in the following primary antibodies: EP400 (Bethyl Labs, A300-541-A; Abcam, Ab5201; Abcam, 70301; 1:1,000); DMAP1 (Cell Signaling Technology, 13326; Santa Cruz, sc-373949; 1:500 or 1:1,000); TRRAP (Bethyl, A301-132A; 1:500 or 1:1,000); ARNT2 (in house; 1:1,000)[28]; NPAS4 (in house, 1:1,000)[14]; FOS (in house; 1:1,000); β-tubulin3 (Covance, MMS-435P; 1:5,000); GAPDH (Sigma-Aldrich, G9545; 1:5,000); histone H3 (Abcam, 1791; 1:10,000); NPAS3 (gift from S. McKnight; 1:1,000)[55]; and HA (Cell Signaling Technology, C29F4; 1:1,000). Following washing, membranes were incubated with secondary antibodies conjugated to IRdye 700 or 800 and imaged using a LiCOR Odyssey instrument.

**Mass spectrometry.** Samples were processed according to standard procedures of the Taplin Mass Spectrometry Facility (Harvard University). Rehydrated proteins were incubated with 50 mM ammonium bicarbonate solution containing 12.5 ng µl$^{-1}$ modified sequencing-grade trypsin (Promega) at 4 °C for 45 min. Following removal of excess trypsin solution, samples were placed at 37 °C in 50 mM ammonium bicarbonate solution. Peptides were recovered through the removal of ammonium bicarbonate solution and were subsequently washed in 50% acetonitrile and 1% formic acid before dehydration.

**Lentivirus production.** To generate lentiviral shRNA expression constructs, 21 bp targeted sequences from the TRC (Sigma-Aldrich) were subcloned into the FUW vector downstream of the U6 promoter. The following sequences were used: control/non-targeting shRNA (5′-gcg cgatagcgctaataattc-3′); *Npas4* shRNA-1 (5′-ggttgaccctgataanttta-3′); *Tip60* shRNA-1 TRCN0000039299 (5′-cctcctatcctaccgaagtta-3′); *Tip60* shRNA-2 TRCN0000039300 (5′-cggagtatgactgcaaaggtt-3′); *Ep400* shRNA-1 TRCN0000109315 (5′-ccgtgaacattagctttgatt-3′); and *Ep400* shRNA-2 TRCN0000305480 (5′-gtcgtcagaaggccttatatg-3′).

To generate LentiCRISPR constructs used for testing sBLISS-seq in cultured neurons, the following guide sequences were cloned into LentiCRISPRv2GFP (Addgene, 82416) using the listed primer sequences: (1) *Scg2* 5′-cggcccgagccctcactca-3′: primer F: 5′-caccgcggcccgagccct cactcag-3′; primer R: 5′-aaacctgagtgagggctcgggccgc-3′; (2) *Inhba* enhancer 5′-gagcagccactagcgaaccc-3′: primer F: 5′-caccggagcagcc actagcgaaccc-3′; primer R: 5′-aaacgggttcgctagtggctgctcc-3′; (3) *Bdnf* 5′-tgatagtggaaattgcatg-3′: primer F: 5′-caccgtgatagtggaaattgcatgg-3′; primer R: 5′-aaacccatgcaatttccactatcac-3′; (4) *Fos* 5′-gcgcggtcac tgctcgttc-3′: primer F: 5′-caccggcgcggtcactgctcgttcg-3′; primer R: 5′-aaaccgaacgagcagtgaccgcgc-3′.

To produce lentivirus for shRNA-mediated depletion of NPAS4, EP400 and TIP60 in primary neuronal cultures or to express LentiCRISPR constructs, 10 µg of lentiviral plasmid was transfected into HEK293T cells along with third-generation packaging plasmids pMDL (5 µg), RSV (2.5 µg) and VSVG (2.5 µg). For the LentiCRISPR pool, 2.5 µg each of LentiCRISPR *Fos* gRNA, LentiCRISPR *Scg2* gRNA, LentiCRISPR *Bdnf* gRNA and LentiCRISPR *Nptx2* gRNA were transfected, for a total of 10 µg of plasmid. At 12–16 h following transfection, transfected medium was exchanged for fresh medium, and supernatant containing virus was collected at 48 h after transfection. For shRNA constructs, supernatant containing virus particles from 10–15 plates of transfected HEK293T cells were pooled, and virus particles were isolated by high-speed centrifugation (25,000 r.p.m. for 90 min). Pelleted virus was resuspended overnight at 4 °C in 100–150 µl of 1× PBS. For LentiCRISPR constructs, supernatant containing virus particles from 10 plates of transfected HEK293T cells were collected. Particles were precipitated by the addition of 1 volume of 4× PEG solution (40% PEG-8000, 1.2 M NaCl in 1× PBS pH 7.4) to 3 volumes of viral supernatant and stored for 1 h at 4 °C. Following incubation of the PEG and viral mixture, particles

were precipitated by centrifugation at 1,500g for 45 min. Pelleted virus was resuspended overnight at 4 °C in 500 μl of 1× PBS. Individual viruses were titred, and the minimum amount of virus required to achieve approximately 85–100% infection, as assessed by GFP florescence, was determined for each lentivirus (1–10 μl per 1.5 million neurons for shRNA constructs, and 60 μl per 1.5 million neurons for LentiCRISPR constructs). Neurons were infected on 3 DIV with shRNA viruses and collected at 7 DIV for RNA-seq analysis. Neurons were infected on 2 DIV with LentiCRISPR viruses and collected at 7 DIV for sBLISS-seq analysis.

All AAV backbones were generated by using standard cloning and molecular biology techniques. AAV2/9 was prepared at the Boston Children's Hospital Viral Core.

**Neuronal transfection and luciferase reporter assays.** Luciferase induction was regulated by positioning NPAS4 target enhancers upstream of the luciferase gene. Sequences to test in luciferase reporter assays were chosen on the basis of the strength of NPAS4 binding in cultured neurons as previously described[28]. High-affinity NPAS4 sites include regions selected from the top 100 high-confidence NPAS4-binding peaks. Regions were PCR-amplified from mouse genomic DNA and subcloned into the pGL4.11 vector using standard Gibson assembly. See below for primer sequences.

To conduct the luciferase assays, 400,000 mouse hippocampal neurons were plated onto 24-well culture dishes and transfected with plasmids using Lipofectamine 2000 (Invitrogen) according to the manufacturer's protocol. At 5 DIV, neurons were transfected with 1 μg of total DNA consisting of 450 ng of firefly luciferase reporter DNA in pGL4.11, 50 ng pGL4.74 renilla luciferase reporter DNA (Promega) and 500 ng of NPAS4 overexpression construct. Lipofectamine (2 μl, Invitrogen) was used for each 1 μg of DNA. DNA–lipofectamine complexes were added dropwise to neurons and incubated for 2 h, after which the transfection medium was replaced with conditioned neuronal medium. Neurons were silenced on 6 DIV overnight through the addition of 1 μM TTX (Abcam, ab120055) and 100 μM AP5 (Thermo Fisher, 01-061-0). At 7 DIV, neurons were collected. In brief, neurons were washed 2× with PBS and lysed through the addition of 500 μl of passive lysis buffer (Dual-Luciferase Reporter Assay System, Promega, E1910). Next, 20 μl of each lysate was added to one well of a Costar white polystyrene 96-well assay plate (Corning). Luciferase Assay reagent II (LARII) and Stop & Glo reagent (Dual-luciferase assay system, Promega) were added to neuronal lysates, and luciferase/renilla measurements were made with a Synergy 4 Hybrid Microplate Reader (BioTek).

To control for variations in transfection efficiency and cell lysate generation, luciferase/renilla ratios were first calculated for each independently transfected well. Luciferase activity (luciferase/renilla) was then normalized to the average value of luciferase activity in nuclear GFP-expressing control samples collected on the same day from the same culture. Data were collected from at least three independent primary neuronal cultures generated on separate days, except for peak 1 Npas4_1–699 (collected from two independent experiments). For each experimental culture, two to four independent wells were transfected. Data from each independently transfected well across all experiments are displayed. Error bars are ±s.e.m. P values were calculated using two-tailed, unpaired t-tests using in Prism (v.8.4.2). Benjamini–Hochberg correction for multiple hypothesis testing was performed in R (v.3.6.1).

Primers used the clone the listed genomic locations into pGL4.11 are listed below:

Peak 1: chromosome 5: 103,753,620–103,754,119
F: 5′-ggaaacagtgtctcacacaacctgcagggtttctgggcactttcaaaact-3′
R: 5′-caatgccaactgttactagctggcgcgcccgggcaccggggggcagccgggcgag-3′
Peak 2: chromosome 7: 112,679,812–112,680,311
F: 5′-ggaaacagtgtctcacacaacctgcaggagcggccccgccgccct-3′
R: 5′-caatgccaactgttactagctggcgcgcctttttggggtcctcctggg-3′
Peak 3: chromosome 8: 84,197,485–84,197,984
F: 5′-ggaaacagtgtctcacacaacctgcaggtgcctgctcgcttgtttacctcc-3′
R: 5′-caatgccaactgttactagctggcgcgcctaccagggcgctgttggggtgggtg-3′
Peak 4: chromosome 1:118,825,347–118,825,846
F: 5′-ggaaacagtgtctcacacaacctgcagggcaggttgatttggtaggaa-3′
R: 5′-caatgccaactgttactagctggcgcgccgtgcaaggagagccgcgggcg-3′
Peak 5: chromosome 2: 94,243,619–94,244,238
F: 5′-ggaaacagtgtctcacacaacctgcagggcgggtgtgcaacgggcagagag-3′
R: 5′-caatgccaactgttactagctggcgcgcctcatgtcccacccctaggcttcctc-3′
Peak 6: chromosome 12: 104,415,550–104,416,049
F: 5′-agtgtctcacacaacctgcaggcactgcagccagcgtccaccatgctt-3′
R: 5′-aactgttactagctggcgcgccccgactccatctttgacaaaag-3′

**Validation by quantitative PCR with reverse transcription.** RNA was extracted using a Qiagen RNeasy Micro kit (Qiagen, 74004), and equivalent amounts of RNA (100–200 ng) across all samples were converted to cDNA using a High Capacity cDNA kit (Thermo Fisher, 4368813) according to manufacturer's instructions. cDNA was diluted by at least threefold before running standard quantitative PCR (qPCR) with reverse transcription methods using Sybr Green master mix. qPCR was performed with technical triplicates using a QuantStudio 3 qPCR machine (Applied Biosystems). Expression for each qPCR target gene was normalized to the housekeeping genes *Gapdh* or *Tubb3* as indicated. The following primer sets were used to amplify genes of interest.

Primer sets:
*Gapdh* F: 5′-AGGTCGGTGTGAACGGATTTG-3′
*Gapdh* R: 5′-GGGGTCGTTGATGGCAACA-3′
*Tubb3* F: 5′-TAGACCCCAGCGGCAACTAT-3′
*Tubb3* R: 5′-GTTCCAGGTTCCAAGTCCACC-3′
*S100b* F: 5′-TGGTTGCCCTCATTGATGTCT-3′
*S100b* R: 5′-CCCATCCCCATCTTCGTCC-3′
*Mog* F: 5′-ACCTCTACCGAAATGGCAAGG-3′
*Mog* R: 5′-TCACGTTCTGAATCCTAAGGGT-3′
*Aldh1l1* F: 5′-CAGGAGGTTTACTGCCAGCTA-3′
*Aldh1l1* R: 5′-CACGTTGAGTTCTGCACCCA-3′
*Pdgfra* F: 5′-AGAGTTACACGTTTGAGCTGTC-3′
*Pdgfra* R: 5′-GTCCCTCCACGGTACTCCT-3′
*Gfap* F: 5′-CGGAGACGCATCACCTCTG-3′
*Gfap* R: 5′-AGGGAGTGGAGGAGTCATTCG-3′
*Grin1* F: 5′-AGAGCCCGACCCTAAAAAGAA-3′
*Grin1* R: 5′-CCCTCCTCCCTCTCAATAGC-3′
*Rbfox3* F: 5′-ATCGTAGAGGGACGGAAAATTGA-3′
*Rbfox3* R: 5′-GTTCCCAGGCTTCTTATTGGTC-3′
*Grin2b* F: 5′-GCCATGAACGAGACTGACCC-3′
*Grin2b* R: 5′-GCTTCCTGGTCCGTGTCATC-3′
*Synapsin1* F: 5′-AGCTCAACAAATCCCAGTCTCT-3′
*Synapsin1* R: 5′-CGGATGGTCTCAGCTTTCAC-3′
*Npas4* F: 5′-ACCTAGCCCTACTGGACGTT-3′
*Npas4* R: 5′-CGGGGTGTAGCAGTCCATAC-3′
*Tip60* F: 5′-GTCACCCGGATGAAGAACAT-3′
*Tip60* R: 5′-GGAAACACTTGGCCAGAAGA-3′
*Ep400* F: 5′-CAGCTCCTCCTAAGCCACAG-3′
*Ep400* R: 5′-CCTCTTGAAGCTTTGGCAAC-3′

**FACS staining and sorting neuronal nuclei for DNA isolation.** To sort neuronal nuclei for DNA isolation, dissected hippocampal tissue was placed in 1 ml of buffer HB (0.25 M sucrose, 25 mM KCl, 5 mM MgCl$_2$, 20 mM Tricine-KOH, pH 7.8, 1 mM DTT, 0.15 mM spermine and 0.5 mM spermidine) and dounced 5× with a loose pestle and 10× with a tight pestle. IGEPAL CA-630 (5%, 32 μl) was added before douncing again with a tight pestle 5–8 times. The nuclei suspension in HB was filtered through a 40-μm strainer. Nuclei were then pelleted by centrifugation at 500g for 5 min and resuspended in 400 μl of FACS block/stain buffer (1% BSA, 0.05% IGEPAL CA-630, 3 mM MgCl$_2$ in 1× PBS). Nuclei were incubated for 15 min with gentle rotation at 4 °C. Following this blocking step, nuclei were pelleted and resuspended in FACS block/stain buffer containing 1:1,000 dilution of mouse anti-NeuN-Alexa488 (Millipore, MAB377X,

cloneA60). To set sort gates, an isotype control mouse IgG-Alexa488 (Life Technologies, MA518167) was included as a negative control along with an unstained sample. Samples were incubated in antibody mix for 1 h with gentle rotation at 4 °C. Nuclei were washed 1× with FACS block/ stain buffer, and DRAQ5 nuclear dye (Abcam, ab108410) was added (1:500) before sorting. NeuN high-expressing nuclei were separated from NeuN low-expressing nuclei using a SONY SH800. Nuclei were sorted directly into ATL lysis buffer provided in a QIAamp DNA Micro kit. DNA was extracted using a QIAamp DNA Micro kit (Qiagen, 56304) according to the manufacturer's protocol and stored at −80 °C until amplicon library preparation. FACS analyses in gating figures shown throughout the manuscript were performed using FlowJo (10.0.8rl).

**FACS sorting AAV-infected neuronal nuclei for CUT&RUN.** Dissected hippocampal tissue was examined under a fluorescent scope to detect GFP (ΔCre) or mCherry (Cre). Tissue that was uninfected, or in rare cases showed infection of both fluorophores in a single hemisphere, was discarded. Hippocampi were placed in 0.5 ml of buffer HB (0.25 M sucrose, 25 mM KCl, 5 mM MgCl$_2$, 20 mM Tricine-KOH, pH 7.8, 1 mM DTT, 0.15 mM spermine and 0.5 mM spermidine) and dounced 5× with a loose pestle and 10× with a tight pestle. IGEPAL CA-630 (5%, 32 μl) was added before douncing with a tight pestle 5–8 more times and filtering through a 40-μm strainer. DRAQ5 nuclear dye (Abcam, ab108410) was added (1:500), and nuclei expressing either mCherry or GFP were sorted on a SONY SH800. Negative gates were determined using uninfected tissue. Nuclei were collected in 1 ml of CUT&RUN wash buffer containing 2 mM EDTA. FACS analyses in gating figures shown throughout the manuscript were performed using FlowJo (10.0.8rl).

**RNA-seq library preparation.** Neurons infected with either control shRNA virus or viruses targeting *Npas4*, *EP400* or *Tip60* were washed twice with PBS to remove dead cells and scraped immediately into TRIzol. For wild-type time course in hippocampus paired with sBLISS-seq, microdissected tissue was flash frozen and then thawed in TRIzol. RNA was extracted using a RNAeasy kit (Qiagen) according to the manufacturer's instructions. Total RNA (1,000 ng) was used to generate libraries following ribosomal RNA depletion (NEBNext, E6310X) according to the manufacturer's instructions (NEBNext, E7420). For cultured neurons, 85 bp reads were generated on an Illumina NextSeq 500 and subsequently analysed with our standardized RNA-seq data analysis pipeline (below). For wild-type time course samples, 40 bp paired-end reads were obtained on an Illumina NextSeq 500.

**ATAC-seq library preparation.** For ATAC-seq libraries generated from hippocampal neuronal subtypes in vivo, *CamkIIa*-expressing CA1 pyramidal neurons were isolated from *Camk2a*$^{Cre}$;*Sun1*$^{fl/+}$ mice using the INTACT method[53]. In brief, hippocampi were dounced 15× in buffer HB (0.25 M sucrose, 25 mM KCl, 5 mM MgCl$_2$, 20 mM Tricine-KOH, pH 7.8, 1 mM DTT, 0.15 mM spermine and 0.5 mM spermidine) to release nuclei. Nuclei were purified by spinning through an iodixanol gradient at 10,000$g$ (see description for nuclear isolation in CUT&RUN experiments). Nuclei expressing SUN1–GFP on the nuclear membrane in a Cre-dependent manner were isolated by incubating the nuclear suspension with 10 μg of anti-GFP antibody (Invitrogen, G10362) for 30 min at 4 °C. Antibody-coated nuclei were subsequently captured by incubation with magnetic Protein G Dynabeads (Thermo Fisher). Following nuclear isolation and counting, approximately 30,000–40,000 nuclei per condition were resuspended in L1 lysis buffer (100 mM HEPES-NaOH pH 7.5, 280 mM NaCl, 2 mM EGTA, 2 mM EDTA, 0.5% Triton X-100, 1% NP-40 and 20% glycerol), followed by 5 min incubation in L2 lysis buffer (10 mM Tris-HCl pH 8.0, and 200 mM NaCl) and 5 min incubation in ATAC lysis buffer (10 mM Tris-HCl pH 7.4, 10 mM NaCl, 3 mM MgCl$_2$ and 0.1% IGEPAL CA-630).

Nuclei were transposed using a Nextera DNA Library Prep kit (Illumina, FC-121-1030) as previously described[56]. Transposition was carried out for 30 min at 37 °C. Transposed DNA fragments from individual samples were purified, independently barcoded and amplified for 8–11 cycles. ATAC-seq libraries were selected for fragments ranging from 200 to 1,000 bp by gel electrophoresis and sequenced on an Illumina NextSeq 500 with 75 bp single-end reads.

## CUT&RUN

CUT&RUN experiments were performed as previously described[6]. For mapping of NuA4 components in 4–6-week-old wild-type mice (Fig. 1), replicates consisted of pools of 4–5 mice processed separately. See Supplementary Table 2 for replicate numbers. Males and females were used in equal proportions in pooled replicate samples. In brief, fresh hippocampal tissue pooled from 4–5 mice was placed into 5 ml of buffer HB (0.25 M sucrose, 25 mM KCl, 5 mM MgCl$_2$, 20 mM Tricine-KOH, pH 7.8, 1 mM DTT, 0.15 mM spermine and 0.5 mM spermidine) and dounced 15× with a tight pestle. IGEPAL CA-630 (5%, 320 μl) was added before douncing with a tight pestle 5 more times, and the sample was filtered through a 40-μm strainer into a 15 ml conical collection tube. Five millilitres of working solution (50% iodixanol, 25 mM KCl, 5 mM MgCl$_2$, 20 mM Tricine-KOH, pH 7.8, supplemented with protease inhibitors, DTT, spermine and spermidine) was added to the sample. Homogenized tissue was gently added on top of 2 ml of 30/40% iodixanol layers. Samples were centrifuged at 10,000$g$ for 18 min, and 1 ml of nuclei was collected from the 30/40% iodixanol interface. Nuclei were counted and evenly distributed across unstimulated and stimulated conditions in the following amounts: FOS (in house), H3K27ac (Abcam, ab4729): 75,000 nuclei; NPAS4 (in house), EP400 (Bethyl Laboratories, A300-541A; Abcam, Ab5201), ARNT2 (in house), RAD50 (Novus Biologicals, NB100-154): 1,000,000 nuclei; MRE11 (Novus Biologicals, NB100-142), CTCF (Active Motif, 61311), ETL4 (Bethyl Laboratories, A304-928A): 500,000 nuclei. For samples FACS-isolated from *Npas4* cKO mice, MRE11 and EP400 CUT&RUN was performed with 100,000 nuclei. Replicates of infected, FACS-sorted samples consisted of independent pools of two to three mice.

Equal numbers of nuclei between unstimulated (PBS-injected mice) and stimulated conditions (2 h KA-injected mice) were aliquoted into 1 ml of CUT&RUN wash buffer (20 mM HEPES pH 7.5, 150 mM NaCl, 0.2% Tween-20, 1 mg ml$^{-1}$ BSA, 10 mM sodium butyrate and 0.5 mM spermidine supplemented with protease inhibitors). Magnetic concanavalin-A (ConA) beads (Bangs Laboratories) that had been washed with CUT&RUN binding buffer (20 mM HEPES-KOH pH 7.9, 10 mM KCl, 1 mM CaCl$_2$ and 1 mM MnCl$_2$) were added to each sample to bind nuclei. ConA-bead-bound nuclei were incubated overnight at 4 °C in wash buffer (20 mM HEPES pH 7.5, 150 mM NaCl, 0.2% Tween-20, 1 mg ml$^{-1}$ BSA, 0.1% Triton X-100, 2 mM EDTA, 10 mM sodium butyrate and 0.5 mM spermidine supplemented with protease inhibitors) containing 1:50 dilution of the aforementioned antibodies.

After overnight incubation with antibodies, ConA-bead-bound nuclei were washed with CUT&RUN antibody buffer and resuspended in CUT&RUN Triton-wash buffer (CUT&RUN wash buffer supplemented with 0.1% Triton X-100). Protein-A-MNase was added at a final concentration of 700 ng ml$^{-1}$, and samples were incubated at 4 °C for 1 h. ConA-bead-bound nuclei were next washed twice with CUT&RUN Triton-wash buffer and resuspended in 100 μl of CUT&RUN Triton-wash buffer. After the addition of 3 μl of 100 mM CaCl$_2$, samples were incubated on ice for 30 min. To stop the MNase reaction, 100 μl of 2× STOP buffer (340 mM NaCl, 20 mM EDTA, 4 mM EGTA, 0.04% Triton X-100, 20 pg ml$^{-1}$ yeast spike-in DNA and 0.1 μg ml$^{-1}$ RNase A) was added to each sample before incubation at 37 °C for 20 min. Following incubation, a magnet was used to capture Con-A-beads, and supernatants containing DNA fragments released by protein-A-MNase were collected. Two microlitres of 10% SDS and 2 μl of 20 mg ml$^{-1}$ proteinase K were added to supernatants followed by incubation at 65 °C with gentle shaking for 1 h. Standard phenol–chloroform extraction with ethanol precipitation was used to precipitate DNA.

Sequencing libraries from precipitated DNA suspend in TE buffer were generated as previously described[57], with the following changes: Rapid T4 DNA ligase (Enzymatics) was used to perform adapter ligation onto end-repaired and A-tailed DNA. Adaptor dimers were removed from PCR-amplified libraries using a 1.1× ratio of AMPure XP beads. CUT&RUN libraries were sequenced on Illumina NextSeq 500 using 40-bp paired-end reads.

**ChIP–seq.** For ChIP–seq, replicates consisted of pools of four to five mice performed on independent days. For ChIP–seq of NPAS4 from hippocampal tissue, hippocampi from 15 mice were dounced in 5 ml of 1× PBS containing protease inhibitor cocktail (Roche, 11836153001). Formaldehyde (1%) was added to tissue homogenate and incubated for 10 min at room temperature, followed by the addition of 0.125 M glycine for 5 min at room temperature. For ChIP of γH2AX from $Camk2a$-expressing CA1 pyramidal neurons, nuclei were isolated from $Camk2a^{cre};Sun1^{fl/+}$ mice using the INTACT method[53] (see ATAC-seq library preparation section for details of nuclear isolation). Following isolation, nuclei with attached beads were crosslinked with 1% formaldehyde in 1 ml of 1× PBS for 10 min at room temperature. Crosslinking was quenched by the addition of 0.125 M glycine for 5 min at room temperature. Crosslinked nuclei were frozen and stored at −80 °C before proceeding with the protocol outlined below.

Nuclei were release from tissue by 10 min incubation in lysis buffer 1 (LB1) (100 mM HEPES-NaOH pH 7.5, 280 mM NaCl, 2 mM EDTA, 2 mM EGTA, 0.5% Triton X-100, 1% NP-40 and 20% glycerol) followed by washing in buffer containing 10 mM Tris-HCl pH 8.0, and 200 mM NaCl. Chromatin was sheared using a Bioruptor (Diagenode) on high power mode for 40–42 cycles with 30-s pulses in sonication buffer (10 mM Tris-HCl pH 8.0, 100 mM NaCl, 1 mM EDTA, 0.5 mM EGTA, 0.1% sodium deoxycholate and 0.5% $N$-lauroylsarcosine).

Following sonication, 1.5 ml of chromatin from hippocampal tissue (about 60 μg) was supplemented with 1% Triton and incubated overnight at 4 °C with 4 μg NPAS4 (in house) coupled to 15 μl Protein A Dynabeads (Invitrogen, 10001D). For γH2AX ChIP, 1.5 ml of chromatin released from around 100,000 purified nuclei was incubated with 2 μl of anti-γH2AX (Abcam, ab2893) coupled to 15 μl Protein A Dynabeads. Beads were washed twice sequentially in 0.5 ml of the following buffers: low-salt buffer (20 mM Tris pH 8, 150 mM NaCl, 2 mM EDTA, 1% Triton X-100 and 0.1% SDS), high-salt buffer (20 mM Tris pH 8, 500 mM NaCl, 2 mM EDTA, 1% Triton X-100 and 0.1% SDS), LiCl wash buffer (10 mM Tris pH 8, 1 mM EDTA, 1% NP-40, 250 mM LiCl and 1% sodium deoxycholate), followed by a wash in 1 ml of TE buffer. Protein–DNA complexes were eluted off the beads by incubation with 200 μl of TE plus 1% SDS for 30 min at 65 °C. Crosslinked DNA was reversed by incubation overnight at 65 °C. The following day, RNA and protein were digested away by the addition of 10 μg of RNase A and 5–7 μl of proteinase K (New England Biolabs, P8107S). DNA was purified by standard phenol–chloroform extraction. ChIP–seq libraries were generated using an Ovation Ultralow V2 kit (Nugen, 0344-32) according to the manufacturer's instructions and PCR amplified for 13–16 cycles, depending on the antibody. Library quality was assessed using an Agilent 2100 Bioanalyzer (Agilent Technologies). Reads (75 bp) were generated using an Illumina NextSeq 500.

**snRNA-seq.** To isolate hippocampal nuclei, we placed flash-frozen hippocampal tissue in 0.5 ml of buffer HB (0.25 M sucrose, 25 mM KCl, 5 mM $MgCl_2$, 20 mM Tricine-KOH, pH 7.8, 1 mM DTT, 0.15 mM spermine and 0.5 mM spermidine) and dounced 5× with a loose pestle and 10× with a tight pestle. IGEPAL CA-630 (5%, 32 μl) was added before douncing with a tight pestle 5–8 more times and filtering through a 40-μm strainer into a 15 ml conical collection tube. Buffer HB (3.5 ml) and 5 ml working solution (50% iodixanol, 25 mM KCl, 5 mM $MgCl_2$, 20 mM Tricine-KOH, pH 7.8, supplemented with protease inhibitors, DTT, spermine and spermidine) were added. Homogenized tissue was gently layered on

top of 1 ml of 30% iodixanol on top of a layer of 1 ml of 40% iodixanol (diluted from a working solution). Samples were centrifuged at 10,000$g$ for 18 min, and 70 μl of nuclei was collected from the 30/40% iodixanol interface. An aliquot of each sample was incubated with trypan blue, and nuclei were counted using a standard haemocytometer.

snRNA-seq was performed using a 10x Genomics Chromium Single Cell kit (v.3). Each reaction lane was loaded with up to 10,000 nuclei from one hippocampal hemisphere (infected with either Cre-mCherry or ΔCre-GFP) from one mouse. Subsequent steps for cDNA amplification and library preparation were conducted according to the manufacturer's protocol (10x Genomics). Samples were sequenced using an Illumina NextSeq 500 with 28 bp (R1), 56 bp (R2) and 8 bp (index) reads.

To facilitate detection of viral transcripts, 2 μl of remaining UMI-barcoded cDNA was amplified in a separate set of PCRs to increase the abundance of UMI-labelled viral transcripts. Custom primers were used to amplify mCherry or GFP transcripts present in the barcoded cDNA library. A nested PCR strategy using Q5 high-fidelity polymerase was used to reduce nonspecific products, with the forward primer in the second reaction also adding the sequence for Illumina Read2 (R2) to the amplified product (primers: GFP amplification 1: 5′-CGCCGACCACTACCAGCAGAACACC-3′; GFP amplification 2: 5′-GTGACTGGAGTTCAGACGTGTGCTCTTCCGATCTgctggganttcgtgaccgccgcc-3′; mCherry amplification 1: 5′-CACTACGACGCTGAGGTCAAGACCACC-3′; mCherry amplification 2: 5′-GTGACTGGAGTTCAGACGTGTGCTCTTCCGATCTgcgccgagggccgccactcc-3′). PCR products were isolated following the first and second steps of the nested PCR using a 0.6× size selection with SPRIselect reagent. Final products from the nested PCR were then diluted 1:10 and put into a sample indexing PCR using the sample indexing primer and Chromium i7 sample index plate, as described in the 10x Genomics library preparation protocol. These sample-indexed viral transcripts were spiked into the sequencing runs used for the corresponding full cDNA libraries. Our final datasets include 32,418 nuclei collected from two independent $Npas4^{fl/fl}$ mice and 44,511 nuclei collected from three independent $Tip60^{fl/fl}$ mice. Ultimately, we identified 12,963 Cre-infected and 8,845 ΔCre-infected nuclei in the $Npas4^{fl/fl}$ snRNA-seq dataset, as well as 13,536 Cre-infected and 13,461 ΔCre-infected nuclei in the $Tip60^{fl/fl}$ dataset (Extended Data Fig. 4).

**sBLISS-seq in cultured neurons and in isolated hippocampal nuclei. Nuclear isolation, fixation and adapter ligation.** sBLISS-seq was carried out on cultured neuronal nuclei or on nuclei isolated from hippocampal tissue according to a previously described protocol[30]. Replicates consisted of individual mice for the wild-type time course. For the Cre and ΔCre datasets, a replicate consisted of one infected hemisphere from one mouse. Samples were paired such that 2 h KA_CRE1 and 2 h KA_ΔCre1 came from the same mouse. For sBLISS-seq performed on $Npas4$ or $Tip60$ wild-type or cKO nuclei infected with AAVs, hippocampal tissue was dissected and quickly examined under a fluorescent scope to detect GFP (ΔCre) or mCherry (Cre). Tissue that was uninfected, or in rare cases showed infection of both fluorophores in a single hemisphere, was discarded. For AAV-infected hippocampi, the tissue was not sorted to minimize stress on the nuclei that could induce ectopic DNA breaks. For uninfected hippocampal nuclei used in the wild-type sBLISS-seq time course, the hippocampi were dissected and processed as described below.

To process hippocampal tissue for sBLISS-seq, hippocampi were placed in 1 ml of buffer HB (0.25 M sucrose, 25 mM KCl, 5 mM $MgCl_2$, 20 mM Tricine-KOH, pH 7.8, 1 mM DTT, 0.15 mM spermine and 0.5 mM spermidine) and dounced 5× with a loose pestle and 10× with a tight pestle. IGEPAL CA-630 (5%, 32 μl) was added before douncing with a tight pestle 5–8 more times and filtering through a 40-μm strainer. To preserve DNA breaks for the remainder of the protocol, nuclei were fixed with 2% PFA (Electron Microscopy Sciences, 15710) followed by

quenching with 125 mM glycine. Fixed nuclei were gently pelleted by centrifuging at 500g. To remove excess debris and myelin, fixed nuclei were run through a modified iodixanol gradient. Pelleted nuclei were resuspended in 1 ml of 22% iodixanol, laid on top of 100 µl of 43% iodixanol. Nuclei were then centrifuged at 1,500g for 15 min in an Eppendorf Centrifuge 5804R in a swinging bucket rotor. Nuclei (100 µl), which settle at the 43/22% interface, were collected and transferred to a Protein Low-Bind tube. Nuclei were washed once with ice-cold PBS plus 1% BSA followed by 2 washes with ice-cold 1× PBS. Fixed nuclei were counted (715,000 nuclei for the wild-type time course, around 250,000 nuclei for *Npas4*[fl/fl] and wild-type Cre and ΔCre datasets, and about 100,000 nuclei for *Tip60*[fl/fl] Cre and ΔCre datasets) and were transferred to a fresh tube to begin adapter ligation.

Note that the wild-type time course samples were performed in two batches with stimulation time points split across these two batches. We did observe a batch effect from processing the samples, which was computationally removed (see analysis of sBLISS-seq samples). The batch for the associated samples is listed in Supplementary Table 2. Note also that the wild-type time course samples included a 2% spike-in of human HEK293T cells expressing a guide-RNA-induced cut site. Spike-in normalization was not performed in the final data processing owing to low expression of the guide and low coverage of reads across the cut site per sample.

To isolate nuclei from cultured neurons plated in 6-well dishes for Cas9 control experiments, neurons were washed with ice-cold PBS to remove debris and 1 ml of HB, and 32 µl of 5% IGEPAL CA-630 was added to each well. Cells were incubated for 10 min in HB with gentle rotation at 4 °C before removal by gentle scraping and transfer to Eppendorf tubes. Nuclei from cultured neurons were fixed with 2% PFA (Electron Microscopy Sciences, 15710) followed by quenching with 125 mM glycine and gently pelleted by centrifuging at 500g. The aforementioned gradient was omitted owing to a lack of debris and myelin in neuronal cultures. Nuclei were washed once with ice-cold PBS plus 1% BSA followed by 2 washes with ice-cold 1× PBS. Roughly 1 × 10^6 cultured nuclei were used for in vitro CRISPR–Cas9 datasets.

Fixed nuclei were incubated for 1 h at 37 °C in LB2 (10 mM Tris-HCl, 150 mM NaCl and 0.3% SDS). Nuclei were washed twice in CutSmart Wash Buffer (1× CutSmart (New England Biolabs, B7204) and 0.1% Triton X-100). At room temperature, nuclei were blunted using a Quick Blunting kit (New England Biolabs, E1201) for 1 h. Following two washes with CutSmart wash buffer, adapters were ligated onto free DNA ends in intact nuclei using T4 DNA ligase (5 U ml⁻¹; Thermo Fisher Scientific, EL0011). Ligation reactions were carried out for about 18 h at 16 °C in a Thermomixer. The following day, nuclei were washed twice with CutSmart wash buffer and incubated overnight in 100 µl of DNA extraction buffer (10 mM Tris-HCl, 100 mM NaCl, 10 mM EDTA and 0.5% SDS) containing 10 µl of proteinase K (20 mg ml⁻¹; New England Biolabs). DNA was extracted using standard phenol–chloroform extraction and resuspended in 18 µl of ultrapure DNase/RNase-free water for subsequent fragmentation.

**DNA fragmentation, IVT and library production.** In vitro and wild-type time course samples were fragmented using a Bioruptor, 30 s on, 60 s off, high intensity, for 35–40 cycles. Owing to limited input from the Cre and ΔCre samples, we found that using enzymatic fragmentation preserved fragmented DNA better than sonication. As a result, all samples from hippocampal tissue were fragmented using NEB Fragmentase for 40–50 min at 37 °C. This treatment resulted in fragments 400–800 bp in size, with the average library size approximately 650 bp. Following fragmentation and clean-up, 100 ng (for cultured neuron samples), 200 ng (for wild-type time course samples) or 35 ng (Cre and ΔCre samples) was reverse transcribed using a MegaClear IVT kit (Thermo Fisher, AMB13345). Libraries from IVT products were produced as previously described[30] without deviation from the protocol[30]. In brief, template DNA was removed by incubating the IVT products with 2 µl of DNaseI (Thermo Fisher, AM2222) for

15 min, and RNA was purified using RNAXP clean beads. RA3 adapter (/5rApp/TGGAATTCTCGGGTGCCAAGG/3SpC3/) was ligated onto RNA using T4 RNA Ligase 2, truncated (New England Biolabs, M0242L). RNA was then reverse transcribed using the reverse primers RTP (5′-GCCTTGGCACCCGAGAATTCCA-3′) and a SuperScript IV Reverse Transcriptase kit (Thermo Fisher, 18090200). Finally, DNA was amplified using NEBNext Ultra II Q5 master mix (New England Biolabs, M0544L) in 8 reactions of 50 µl each. Libraries were sequenced using single-end 75 bp reads on an Illumina NextSeq 500.

**END-seq.** Culture mouse cortical neurons for END-seq were infected with a non-targeting shRNA lentivirus on 3 DIV before collection on 7 DIV. Replicates consisted of cultures generated on independent days. To dissociate cultured mouse cortical neurons for END-seq, papain (Worthington Biochemical, LK003178) was dissolved in TrypLE Express enzyme solution at 37 °C for 20 min before sample collection. Culture medium was gently aspirated and the cells were gently washed once with PBS at 37 °C. Papain solution (500 µl) was added to each well of neurons in 6-well plates and incubated at 37 °C for 1 min. Papain solution was removed using a pipette, and 500 µl trituration solution (culture medium supplemented with freshly dissolved DNase) was added. Cells were gently triturated 5–10 times with a wide bore pipette tip, transferred to conical tubes and pelleted. Cells were gently resuspended in ice-cold PBS containing 0.1% BSA and 0.5 mM EDTA for counting. Cells were stored on ice while an aliquot was quickly counted using a haemocytometer. Next, 1.5 million neurons for each experimental condition were added to a new conical tube and pelleted. Cells were then embedded in agarose plugs using a CHEF Mammalian Genomic DNA Plug kit (Bio-Rad, 1703591) and processed for END-seq as previously described[32,58]. Experimental groups were processed back-to-back, collecting cells in one treatment group and embedding them in agarose plugs before collecting cells in the next treatment group to reduce overdigestion of the samples and to minimize time between cell dissociation and embedding in agarose. We observed better signal-to-noise in cells treated with etoposide (50 µM). Etoposide was added to the cells 6 h before collection and was included at 50 µM in all solutions applied to the cells until embedding in agarose.

**Amplicon PCR sequencing (SiMSen-seq). Selection of sites for mutation analysis.** To select sites to examine for mutational load during ageing, a universe of possible sites to examine was created by taking the union of all accessible regions in *CamkIIa*-expressing neurons (the predominant cell class in CA1 hippocampal regions; defined in our INTACT *CamkIIa* datasets) and all regions marked by H3K27ac in hippocampal nuclei. This generated a set of 179,841 possible regions, and we considered windows of 500 bp from the centre of these peaks. We calculated the normalized read intensity for ATAC-seq, H3K27ac CUT&RUN and γH2AX ChIP–seq using the function in DeSeq2 counts (dds, normalized=TRUE)[59] for each of these possible regions across all conditions. We also extracted the normalized signal intensity for NPAS4 and MRE11 CUT&RUN using Homer(homer/4.9) annotatePeaks.pl function with default normalization to 10 million total sequencing reads. Finally, we determined regions that overlapped a NPAS4 and/ or MRE11 peak by at least 1 bp. These sites were considered 'bound'.

Using the previously mentioned set of regions, we chose sites to interrogate based on the following criteria: (1) sites bound by NPAS4 and MRE11 with the highest normalized signal for NPAS4; (2) sites bound by NPAS4 and MRE11 that show an increase in γH2AX ChIP–seq signals following neuronal activation. We also included a set of sites that did not overlap with a NPAS4 or MRE11 peak and as a group did not differ significantly in terms of their GC content, chromatin accessibility or levels of H3K27ac. However, we note that in our final site selection, some sites that matched these criteria did not amplify efficiently in our assay and were therefore excluded owing to technical considerations. The final set of sites assayed are found in Supplementary Table 5.

Primers were designed using custom R scripts calling Primer3, with Primer3 formatting derived from http://bioinfo.ut.ee/primer3-0.4.0/input-help.html. The following sequence was appended onto the 5′ end of the forward primer GGACACTCTTTCCCTACACGACGCTCTTCCGATCTNNNNNNNNNNNNATGGGAAAGAGTGTCC, where the 12 Ns represent random nucleotides constituting a UMI. The following sequence was appended onto the 5′ end of the reverse primer GTGACTGGAGTTCAGACGTGTGCTCTTCCGATCT. Primer sequences for the amplicons targeted can be found in Supplementary Table 5. Primers were ordered as 200 pM or 4 nM Ultramers from IDT.

**Production of targeted amplicon libraries.** Amplicon libraries were generated using a previously described protocol[38] with minor modifications. For all samples, sites of interest were divided into pools of 15 primer sets (see Supplementary Table 5 for primer pooling). For each primer pool, 20–30 ng of DNA isolated from the FACS-isolated NeuN+ nuclei were first amplified with Phusion Hot Start II high-fidelity DNA polymerase (Thermo Fisher, F-549L) to append UMI sequences to DNA templates. The final concentration of each primer in the reaction was 40 nM. PCR parameters for these 4 cycles were as follows: 98 °C, 30 s; 4 cycles of 98 °C, 10 s, 62 °C, 6 min 72 °C, 30 s; followed by 65 °C 15 min; 95 °C, 15 min. During the 65 °C incubation, the PCR reaction was terminated by incubation with 10 μl of protease (Sigma-Aldrich, P5147-100MG; used at a final concentration of 0.06 μg μl⁻¹). For each primer pool, 10 μl of initial product were next amplified using Phusion Hot Start II high-fidelity DNA polymerase with the following parameters: 98 °C, 3 min; 2–35 cycles of 98 °C, 10 s 80 °C, 1 s; 72 °C, 30 s; 76 °C, 30 s. All with ramping at 0.2 °C s⁻¹. Duplicate reactions were performed for each primer pool to increase the diversity of UMI sampling in the final libraries. Samples were size-selected to include products approximately 275–650 bp in length. Analysis was conducted using Debarcer v.0.3.1 (https://github.com/oicr-gsi/debarcer/releases/tag/v0.3.1)[38].

**Lifespan analysis and tissue collection of NPAS4 knockout cohort.** *Npas4* wild-type or knockout littermates were housed in accordance with protocols approved by the Harvard University Standing Committee on Animal Care. Littermates were housed together and monitored by trained technicians for overall health. Any distressed animals or those that showed poor health were euthanized and censored during data collection. Animals that were used for tissue collection purposes before the final collection date were censored from the final lifespan analysis. Owing to limitations imposed by the COVID-19 shutdown, the final study ended in March 2020, when all remaining animals were dissected. For this reason, the lifespans of *Npas4* wild-type littermates do not go to completion. Significance of survival curves was determined using both a log-rank (Mantel–Cox) and a Gehan–Wilcoxon–Breslow test using Prism (v.8.4.2) software.

### Quantification and statistical analysis

**Statistical analysis and sample size determination.** The statistical analysis for each experiment is detailed in the figure legends. Electrophysiological assays were performed in a blinded manner such that the condition (saline-injected versus KA-injected) was not revealed until after the analysis was complete. Statistical methods were not used to predetermine sample sizes, but replicate numbers generally adhered to guidelines of the ENCODE consortium[60]. Sample randomization was not performed. All statistical analysis was performed in either Prism (v.8.4.2) or R (v.3.6.1). For multiple hypothesis correction shown in Extended Data Figs. 6e and 13b, P values obtained using Prism (v.8.4.2) were corrected using the padjust() function in base R (v.3.6.1).

**Peptide quantification mass spectrometry.** Mass spectrometry and peptide quantifications were performed following the standard practices of the Taplin Biological Mass Spectrometry facility, Harvard Medical School. Data were collected using a LTQ Orbitrap Velos Elite ion-trap mass spectrometer (Thermo Fisher Scientific). In brief, the program Sequest (Thermo Fisher Scientific) was used to compare peptides against protein databases with the acquired fragmentation pattern. Data were filtered to between a 1% and 2% peptide false discovery rate (FDR), and databases included a reversed version of all the sequences. Triplicate NPAS4 immunoprecipitation–mass spectrometry experiments were performed from hippocampal tissue. Duplicate experiments were performed on high molecular weight fractions using NPAS4–Flag-HA and TIP30–H3-Flag cortical lysates. Peptide counts for all mass spectrometry experiments are provided in Supplementary Table 1. The R (3.6.1) package EdgeR[61] (edgeR_3.28.1;limma_3.42.2) was used to identify proteins significantly enriched in NPAS4 or TIP60 immunoprecipitate samples relative to wild-type samples that did not express Flag-tagged proteins. Proteins that were identified in at least 2 out of 3 replicates were included in the EdgeR analysis. Peptides found strictly in wild-type control samples and not in NPAS4 or TIP60 samples (that is, background associated with the M2 resin) were removed before running the EdgeR glmFit() and glmLRT() functions.

**Sequencing and alignment.** All experiments were sequenced on an Illumina NextSeq 500 (Illumina Next-Seq Control Software v.4.0.2). Information on sequencing data is provided in Supplementary Table 2. Single-end reads (75 bp) were obtained for ATAC-seq, ChIP–seq, RNA-seq (cultured neurons) and sBLISS-seq. Paired-end reads (40 bp) were obtained for all CUT&RUN experiments, snRNA-seq experiments and hippocampal KA time course RNA-seq experiments. Single-end reads (162 bp) were obtained for amplicon libraries used in the mutation analysis.

For ATAC-seq and ChIP–seq samples (that is, NPAS4 ChIP and γH2AX ChIP), quality trimming of sequencing reads was performed with trimmomatic/0.36 (ref. [62]) using the following command: java -jar trimmomatic-0.33.jar SE -threads 1 -phred33 [FASTQ_FILE] ILLUMINACLIP:[ADAPTER_FILE]:2:30:10 LEADING:5 TRAILING:5 SLIDINGWINDOW:4:20 MINLEN:35. Nextera adapters were specified for ATAC-seq, and TruSeq adapters were specified for ChIP–seq samples. Samples were subsequently aligned to the mm10 genome using the Bowtie alignment software (vbowtie2/2.2.9) with the –very-sensitive setting. Reads mapping to the mitochondrial genome were removed. Duplicate reads were marked with Picard/2.8.0 with the command java -jar $PICARD/picard-2.8.0.jar MarkDuplicates REMOVE_DUPLICATES=false. Duplicates were subsequently removed using samtools/1.3.1 samtools view -b -F 1796.

For CUT&RUN samples, quality read trimming was performed using trimmomatic/0.36 using the following command: java -jar trimmomatic-0.33.jar PE -phred33 [FASTQ_FILE] ILLUMINACLIP: /n/app/trimmomatic/0.36/bin/adapters/TruSeq3-PE.fa:2:15:4:4:true LEADING:20 TRAILING:20 SLIDINGWINDOW:4:15 MINLEN:25. Adapters were further trimmed according to the pipeline and script (kseq_test) as previously described[63]. Paired-end reads were mapped to the mouse genome using bowtie2/2.2.9 with the following command: bowtie2 --local --very-sensitive-local --no-unal -x mm10 --dovetail --no-mixed --no-discordant --phred33 -I 10 -X 700. Yeast spike-in reads were mapped using the following command: bowtie2 --local --very-sensitive-local --no-unal –x sacCer3 --no-overlap --no-dovetail --no-mixed --no-discordant --phred33 -I 10 -X 700. Read depth (normalized to 1 million) or spike (normalized to 1) normalized bedgraph files were created using custom scripts modified from Spike_in_Calibration_v2.csh as previously described[63]. Finally, bedGraphToBigWig (ucsc-tools/363) was used to generate the bigWig files from either read depth normalized bedgraph files. Files are displayed on the IGV browser show read depth-normalized bigWigs. Mapping for sBLISS-seq samples were performed according to a previously described protocol[30]. The mapping pipeline was cloned from git clone at https://github.com/BiCroLab/blissNP.git.

For amplicon library analysis, adapter contamination was a significant problem for NeuN+-sorted datasets. Samples were therefore

trimmed to remove these adapters with the following trimming command: java -jar trimmomatic-0.33.jar SE -threads 1 -phred33 [FASTQ_FILE] ILLUMINACLIP:[ADAPTER_FILE]:2:30:10 LEADING:5 TRAILING:5 SLIDINGWINDOW:4:15 MINLEN:120. FASTQ files containing only trimmed, adapter-removed reads are provided in the Gene Expression Omnibus (GEO) submission. Full FASTQ files will be provided upon request. Mapping using BWA was performed using the default parameters outlined in the package Debarcer v.0.3.1 (https://github.com/oicr-gsi/debarcer/releases/tag/v0.3.1)[38].

END-seq reads were aligned to reference genome mm10 using subread (v.1.5.1) with parameters subread-align -t1 -T 6 -M 3. The Samtools (v.1.13) functions sort, markdup and index were used to create indexed .bam files with duplicate reads removed. Files were downsampled to the lowest number of reads in any sample within a replicate, and tag directories were compiled using the homer (v.4.9) command makeTagDirectory. Tag directories were used to generate aggregate plots centred on various genomic sites of interest using homer annotatePeaks.pl with parameters -size 10000 -hist 50. For statistical tests, signals were extracted within windows around the genomic sites of interest using homer annotatePeaks.pl with parameters -size 500.

**SAR-seq analysis.** Previously published[32] SAR-seq datasets were downloaded from the GEO database, accession number GSE167259. Three replicates of SAR-seq performed in iNeurons (postmitotic glutamatergic neurons derived from induced pluripotent stem cells) were retrieved (GSM5100400, GSM5100401 and GSM5100402). Tag directories were used to generate aggregate plots centred on various genomic sites of interest using homer annotatePeaks.pl with parameters -size 10000 -hist 50. For statistical tests, signals were extracted within windows around the genomic sites of interest using homer 'annotatePeaks.pl' with parameters -size 500. Site lists generated in mice (mm10) were lifted over to human hg19 for use with human SAR-seq data, using the UCSC Genome Browser liftOver tool. To generate sites with nonoverlapping signals for NPAS4 and FOS, NPAS4 and FOS summits were both extended to 1 kb. NPAS4-bound sites (no FOS) were generated using bedtools/2.27.1 intersect bed -v option to generate peaks with no NPAS4 and vice versa.

**bigWig visualization.** To generate bigWig files for ATAC-seq, ChIP–seq and CUT&RUN datasets, all aligned bam files for each replicate of a given experimental condition were pooled and converted to the BED format with the bedtools/2.27.1 bamtobed. For ATAC-seq and ChIP–seq data, the 75 bp reads were extended in the 3′ direction to 200 bp (average fragment length for ChIP–seq and ATAC-seq experiments as measured by bioAnalyzer) with the bedtools slop command using the following parameters: -l 0 -r 125 -s. For sBLISS-seq libraries, the mapping pipeline generates a single base pair cut site. Thus, for visualization purposes, reads were extended 75 bp in the 3′ direction to match the initial read sequence. Mm10 blacklisted regions were filtered out using the following command: bedops –not-element-of 1 [BLACKLIST_BED][60]. The filtered BED files were converted to coverageBED format using the bedtools genomecov command with the following options: -scale [NORM_FACTOR to scale each library to 20M reads for ChIP, 20M for ATAC, 10M for sBLISS-seq, and 1M for CUT&RUN] –bg. Finally, bedGraphToBigWig (ucsc-tools/363) was used to generate the bigWig files displayed on browser tracks throughout the manuscript using the IGV browser.

**Bulk RNA-seq quantification of gene expression in vitro RNA-seq.** The featureCounts package[64] was used to count reads in cultured neuron RNA-seq data using a custom filtered annotation file (gencode.v17.annotation.gtf filtered for feature_type="gene", gene_type="protein_coding" and gene_status="KNOWN") to obtain read counts along genes for each sample. For differential gene expression analysis, read count tables were TMM-normalized using the EdgeR software analysis package[61]. Any genes that were not expressed in at least three samples with TMM-normalized CPM > 1 were removed from further analysis. The voom and limma (edgeR_3.28.1;limma_3.42.2) analysis software packages were used to quantify differential gene expression (requiring FDR-corrected $q$ < 0.01). For analysis of the paired RNA-seq samples that matched sBLISS-seq, the DeSeq2 package was used to generate normalized counts. After running a standard DeSeq2 pipeline, normalized counts were generated with the function counts(dds, normalized=TRUE)[59].

**Gene ontology enrichment analysis.** Gene ontology (GO) enrichment analysis was performed using gProfiler2 in R (v.0.2.0)[65], with a custom background of expression-filtered genes from neuronal cell types across all scRNA-seq datasets in this manuscript and FDR < 0.05. Select GO terms are displayed in Extended Data Fig. 5e and a complete list of enriched GO terms for each cell type and dataset can be found in Supplementary Table 3.

**snRNA-seq analysis.** FASTQ files were created using the standard bcl2fastq pipeline from Illumina. Gene expression tables for each nuclear barcode were generated using the CellRanger 3.0.0 pipeline as designed by 10x Genomics. Samples were demultiplexed, and all *Npas4*[fl/fl] or *Tip60*[fl/fl] samples were merged using the CellRanger aggr function using default parameters. The datasets were loaded into R and analysed using the Seurat (v.3)[66] and Monocle3[67] packages. Nuclei were removed from the dataset if they contained fewer than 500 detected genes, displayed more than 5% of reads mapping to mitochondrial genes or had RNA counts detected at a level greater than 2 standard deviations higher than the average value in their assigned cell type (which probably reflect doublets and multiplets). To remove potential doublets from the datasets in a more stringent manner, we used the DoubletFinder package in R, which assesses which barcodes in a dataset are most likely to be doublets based on transcriptional similarity to distinct clusters[68]. We used default parameters and an estimated doublet rate of 7% based on guidance from 10x Genomics Chromium Single Cell 3′ Reagent kits v3 User Guide and the number of nuclei run in each reaction lane.

All nuclei in either *Npas4*[fl/fl] or *Tip60*[fl/fl] samples were considered together for clustering and dimensionality reduction. The 2,000 top variable genes across nuclei were identified using Seurat's (v.3) FindVariableFeatures function, which were used to perform PCA using the RunPCA function (both using default parameters). A shared nearest neighbour graph was constructed using the FindNeighbors function (considering the top 30 principal components), and clustering was assigned using the FindClusters function (resolution $n$ = 0.02). The following marker genes were used to assign cell type to the principal ten clusters identified: pan-neuronal (*Rbfox3*); pan-excitatory neurons (*Slc17a7*); pan-inhibitory neurons (*Gad2*); excitatory dentate gyrus neurons (*Prox1* and *C1ql2*); excitatory CA3 neurons (*Cpne4* and *Spock1*); excitatory CA1 neurons (*Mpped1*); excitatory subiculum neurons (*Tshz2*); oligodendrocytes (*Mog* and *Mag*); oligodendrocyte precursor cells (*Pdgfra*); astrocytes (*Gfap*); microglia (*Cx3cr1* and *C1qc*); and endothelial cells (*Cldn5*) (Extended Data Fig. 4). To assign infection status to each nucleus, we set a threshold of detecting more than eight mCherry or GFP transcripts in a given nucleus, which represented an inflection point above the background rate of detection for the distribution of these transcripts per nucleus, and reflected the expected infection patterns based on the known tropism of the AAV2/9 virus used in these experiments. Differential gene expression analysis between Cre and ΔCre-infected cells within each cluster was conducted using Wilcoxon rank-sum test using the FindMarkers function (logfc.threshold = 0.25), and significant genes were defined as those with an adjusted $P$ < 0.01. Heatmaps were generated using custom functions written in R. Violin plots were generated using Seurat VlnPlot function with default parameters.

We observed that in both datasets, within each neuronal cluster, the cells subclustered according to infection status (Cre versus ΔCre),

and this effect was not driven by expression of viral transcripts, as all viral features were removed from the gene expression matrix. This subclustering probably reflects the inability of Cre-infected cells to fully induce activity-dependent genes following depletion of either *Npas4* or *Tip60*. At 6 h after stimulation, these activity-induced genes were among the most highly expressed genes and differed across neuronal cell types, and therefore contributed significantly to the principal components used in dimensionality reduction for cell-type clustering. We used two independent analysis packages, Seurat (v.3) and Monocle3, to call differentially expressed genes and only considered significant genes (adjusted *P* < 0.01) from both analyses. Although we identified genes that are both upregulated and downregulated after acute *Npas4* or *Tip60* depletion, we focused our downstream analysis on downregulated genes that are more likely to be directly activated by this complex. To ensure our observed downregulation of putative target genes was not simply due to higher expression or detection of genes in the ΔCre-infected tissue, we randomly sampled from the top 10% of most highly expressed genes (to account for the target genes being highly expressed overall) and calculated the ΔCre − Cre difference for the set of randomly selected genes. We performed 10,000 iterations of this analysis to generate a distribution of sample differences using randomly selected genes. The actual observed differences (using the NPAS4–NuA4 target genes identified in each neuronal subtype) lay far outside this distribution (Extended Data Fig. 5c).

In general, NPAS4 induces a diverse set of pan-neuronal and cell-type-specific effector genes. Many targets were found to be common among 2 or 3 cell types, and about 10% of targets were specific to one cell class. Broadly, the set of all NPAS4 target genes were enriched for genes with functions in cell–cell adhesion, intercellular signalling and axon guidance, as well as a diverse set of metabotropic and ionotropic neurotransmitter receptor subunits. In addition to capturing known NPAS4 target genes such as *Nptx2*, *Plk2* and *Bdnf*, we identified 1,766 potentially new targets of NPAS4 in the hippocampus by this analysis (Supplementary Table 3).

**ATAC-seq and ChIP–seq peak calling.** ATAC-seq enriched peaks were determined using MACS2 (v.2.1.1) parameters –shift 100 -p 1e-5 --nolambda --keep-dup all --slocal 10000, as previously described[56,69]. ATAC-seq peaks from individual blacklist regions were removed as previously described[60]. sBLISS-seq peaks were called on UMI de-duplicated bed files in which the single cut site had been extended by the length of the sequencing read (75 bp). Peaks were called using MACS2 (v.2.1.1) with the following peak parameters: macs2 callpeak --nomodel --keep-dup all --format BED -g mm -p 1e-5. Reproducible peaks were considered those that overlap in 5 out of 8 replicates. Peaks mapping to the mitochondrial genome were removed. Final peak sets were extended 500 bp from the maximal sBLISS-seq summit in a reproducible peak. Peak calling on ChIP samples was performed using MACS2 (macs2/2.1.1)[70] using the following command macs2 callpeak -t (experimental bam) -c (input bam) -f BAM -g mm -p 1e-5.

**CUT&RUN peak calling.** All peak calling was performed using SEACR_1.1.sh[71]. For H3K27ac, FOS, CTCF and RAD50 CUT&RUN datasets, peak calling on individual replicates was performed using the spike-in normalized bedgraph files based on fragments 1–1,000 bp in length with the following command: SEACR_1.1.sh [target bedgraph] [control bedgraph] norm stringent. Paired control samples (either IgG or knockout control) are listed in Supplementary Table 2. To identify reproducible peak sets, SEACR peaks found in 3 out of 3 H3K27ac replicates (0 and 2 h KA stimulation), 3 out of 3 FOS replicates (0 and 2 h KA stimulation), 3 out of 3 CTCF replicates (0 and 2 h KA stimulation), 3 out of 3 RAD50 replicates (0 h) and 3 out of 4 RAD50 replicates (2 h KA stimulation) were intersected using bedtools/2.27.1 intersect bed. Peaks within 150 bp were merged. For NPAS4, ARNT2, EP400, MRE11

and ETL4 CUT&RUN datasets, peak calling on individual replicates was performed using SEACR_1.1.sh (ref. [71]) using the spike-in normalized bedgraph files based on fragments 1–1,000 bp in length with the following command: SEACR_1.1.sh [target bedgraph] [control bedgraph] norm relaxed. Paired control samples (either IgG or knockout control) are listed in Supplementary Table 2. To identify reproducible peak sets, SEACR peaks found in 4 out of 5 NPAS4 replicates (2 h KA stimulation), 3 out of 3 NPAS4 replicates (0 h KA stimulation), 2 out of 2 ARNT2 replicates (2 h KA stimulation), 2 out of 2 EP400 replicates (0 and 2 h KA stimulation; Abcam antibody), 2 out of 2 MRE11 replicates (0 and 2 h KA stimulation), 2 out of 2 ETL4 replicates (0 h KA stimulation) and 3 out of 4 ETL4 replicates (2 h KA stimulation) were intersected using bedtools/2.27.1 intersect bed. Peaks within 150 bp were merged. Finally, the maxima of CUT&RUN signal within 100 bp windows for each peak was calculated from spike-in normalized bigWig files using custom scripts. For FOS, ARNT2, EP400, ETL4 and MRE11, final peak calls were extended 200 bp upstream and downstream from this peak maxima to generate 500 bp peak calls for each factor and time point. Mm10 blacklisted regions[60] were filtered out using the following command: bedops/2.4.30 –not-element-of 1 [BLACKLIST_BED]. NPAS4 peaks were extended to 1 kb, as we found maximal enrichment for the Ebox (CAGATG) motif and bHLH–PAS motif (CGTG) in 1 kb regions extended from the peak maxima. During the revision process, an additional three replicates of NPAS4 CUT&RUN were added as confirmatory of the initial five replicates performed. These additional NPAS4 CUT&RUN replicates are available as bigWig files and raw FASTQ files in the GEO submission but were not used in the analysis of NPAS4 peak calling.

**Peak annotations and motif finding.** Peak annotations as enhancers, promoters or other were determined using the homer (v.4.9) function annotatePeaks.pl. NPAS4 peak annotations were performed on regions extending 1 kb from the peak maxima, as these regions were most enriched for the NPAS4 motif. FOS peak annotations were performed on regions extending 500 bp from the peak maxima. Active regulatory elements were defined as the union of reproducible ATAC-seq (in 3 out of 3 replicates) and H3K27ac peaks (in 3 out of 3 replicates) extending 500 bp from the maximal ATAC-seq signal in that regulatory element. Enhancers were defined as the union of intergenic and intronic binding site annotations. To find enriched motifs, the sequences underlying each peak were extracted using bedtools/2.27.1 getfasta command. Sequences of equal length (1,000 bp for NPAS4 and MRE11, 500 bp for sBLISS-seq) were processed using Meme-ChIP (https://meme-suite.org/meme/tools/meme-chip)[72] and tested against the motif background (HOCOMOCO v11) with significance reported as the *E*-value.

**Peak overlaps.** To determine the overlap between NPAS4 and additional factors, NPAS4 summits were extended to 1 kb. Peak overlaps were determined using bedtools/2.27.1 intersect bed allowing a minimum of 1 bp overlap. To generate sites with nonoverlapping signal for NPAS4 and FOS, NPAS4 and FOS summits were both extended to 1 kb. NPAS4-bound sites (no FOS) were generated using bedtools/2.27.1 intersect bed -v option to generate peaks with no NPAS4 and vice versa.

**Generation of fixed line plots and aggregate plots.** Fixed line plots were generated using homer(v4.9)'s annotatePeaks.pl [PEAK_BED] mm10 -d [INPUT_TAG_DIRS] -size 2000 –ghist -hist 25 -noann -nogene. Fixed line plots were generated from tag directories containing merged bam information from all replicates. Aggregate plots were generated using homer(v4.9)'s annotatePeaks.pl function with default parameter -hist 25, unless otherwise noted in the legend. Signal intensities were plotted using custom R scripts R (3.6.1). Aggregate plots show the average signal across replicates with the s.e.m. plotted. For statistical analysis of aggregate plots, signals for all replicates were extracted across the window specified in the figure legends using homer(v4.9)'s annotatePeaks.pl [PEAK_BED] mm10 -d [INPUT_TAG_DIRS] with homer's

default read depth normalization to 10 million reads. Signals were averaged across all replicates, and Wilcoxon rank-sum tests were used to compare average signals between different conditions across the specified windows.

**Identification of regulatory landscape in hippocampal neurons.** To define a set of regulatory elements across hippocampal neuronal tissue samples, we used ATAC-seq (in *CamkIIa*⁺ hippocampal neurons), together with CUT&RUN for the active histone modification H3K27ac, to characterize the constitutive and activity-responsive genomic regulatory element landscape in the hippocampus. We profiled hippocampal tissues in the basal state and 2 h following the synchronous induction of neuronal activity by low-dose KA administration. Reproducible ATAC-seq peaks were defined as MACS2 peaks identified in all 3 out of 3 replicates per time point. Sites from unstimulated and stimulated samples were concatenated and merged to generate a list of all possible ATAC-seq peaks across any stimulation condition. Reproducible H3K27ac peaks were defined as SEACR peaks identified in all 3 out of 3 replicates per time point. Sites from unstimulated and stimulated samples were concatenated and merged to generate a list of all possible H3K27ac peaks across any stimulation condition. The final landscape of elements included 179,841 elements, defined as the union of reproducible ATAC-seq and H3K27ac peaks found in our CUT&RUN and ATAC-seq datasets across all time points after removing mm10 blacklist regions. Note that blacklist removal was performed on the union of H3K27ac and ATAC-seq sites. To generate regions of comparable size, we determined the maximal ATAC-seq signal in each regulatory element and extended the regions 500 bp from this maximal chromatin accessibility summit.

To identify inducible ATAC-seq and H3K27ac peaks, we conducted a differential expression analysis using DeSeq2 (v.DESeq2_1.26.0)[59] on regions that had non-zero counts in at least two of the samples. We defined 'inducible elements' within this set as those sites exhibiting an increased ATAC signal (two-fold increase; adjusted $P < 0.05$ DeSeq2; $P$ values were determined by testing against a fold change threshold of 2) and/or an increased H3K27ac CUT&RUN signal (1.5-fold increase; adjusted $P < 0.05$ DeSeq2; $P$ values were determined by testing against a fold change threshold of 1.5) following stimulation (11,114 sites). Regulatory elements are provided in Supplementary Table 4. Note that elements that did not meet the threshold cut-off of non-zero counts in at least two of the samples will not be included in the respective DeSeq2 analysis tab in this table.

**Quantification of CUT&RUN and sBLISS-seq signals at gene regulatory elements.** To quantify transcription factor binding strength, the software package homer(v4.9) was first used to create tag directories for all replicates per factor for a given time point with the command makeTagDirectory [INPUT BED OR BAM]. For sBLISS-seq samples, bed files generated from a previously described pipeline[30] were expanded such that each UMI at each location was individually counted. For sBLISS-seq samples, the –len 0 tag was added to create tag directories to account for the fact that sBLISS-seq mapping generates bed files with single base-pair cut sites. Signals were extracted over sets of regulatory regions using the following command: annotatePeaks.pl [bed file] mm10 -size given -noann. For transcription factors (NPAS4, MRE11, ETL4, EP400, FOS and RAD50), signals were read-depth-normalized using homer(v4.9)'s default normalization to 10 million reads. To plot signals of various genomic features as a function of NPAS4 binding strength, we ranked all regulatory elements in our dataset (see above for the definition of 179,841 ATAC/H3K27ac elements in the hippocampus) according to NPAS4 IgG-normalized CUT&RUN signals and split regions into quartiles based on this ranking. We determined the NPAS4-normalized signal on a per site basis by dividing the aggregate signal for NPAS4 CUT&RUN (merge of NPAS4 replicates 1–5) by the aggregate signal for IgG (merged replicates A–F) at any given site in our regulatory landscape. NPAS4 Q4 (high) sites were determined as sites

that overlapped with a SEACR-defined peak of NPAS4 and were in the top quartile of the NPAS4 IgG-normalized CUT&RUN signal. NPAS4 Q1 (low) sites had no SEACR-defined peak and were in the lowest quartile of the NPAS4 IgG-normalized CUT&RUN signal.

To compare γH2AX, ATAC-seq, H3K27ac, MRE11 and sBLISS-seq signals within the set of 179,841 defined in our regulatory landscape across time points and conditions (for example, different genotypes), raw read counts were extracted using homer(v4.9)'s annotatePeaks. pl function with the following command: annotatePeaks.pl [bed file] mm10 -size given -noann –noadj. For sBLISS-seq signals, the –len 0 parameter was included to quantify only the cut site end (1 bp) and to prevent read shifting based on estimated fragment sizes. Raw signal counts were imported in R. Before running DeSeq2 (v.1.26.0), regions with low counts were excluded as follows: for ATAC-seq, H3K27ac and γH2AX, we required non-zero counts in at least two samples. For sBLISS-seq (both wild-type time course and Cre versus ΔCre datasets), which is sparser than ATAC-seq, H3K27ac CUT&RUN and γH2AX, we only eliminated regions with zero counts across all samples. DeSeq2-normalized counts were generated using the DeSeq2 function counts(dds, normalized=TRUE)[59] for our regulatory region set across all conditions (0 versus 2 h KA for ATAC, H3K27ac and γH2AX and 0, 2, 10 h KA for sBLISS-seq). We observed a batch processing effect for our numerous wild-type time course samples. Samples coming from all time points were included in each batch. However, to remove these effects computationally, we included the batch in our DeSeq2 design with the following command: DESeqDataSetFromMatrix(design ~Seq_ Batch + Condition). In addition, when exporting normalized counts, we used the limma(3.42.2)limma::removeBatchEffect(counts(dds object, normalized=TRUE). Batches for the processing samples can be found in Supplementary Table 2. Signal counts plotted in boxplots in the paper for ATAC-seq, H3K27ac, γH2AX and wild-type sBLISS-seq time course are based on DeSeq2-normalized values averaged across all replicates for a given peak.

For processing the of Cre and ΔCre datasets, DeSeq2 normalization of counts was performed within each genotype (that is, raw counts across Cre versus ΔCre datasets in *Npas4*^fl/fl^ were independently normalized from Cre versus ΔCre in wild-type datasets). We chose this design because these experiments were performed at separate times and comparisons were made between Cre and ΔCre within each genotype rather than across genotypes. To account for variability in sBLISS-seq datasets generated from very low cell numbers in the Cre versus ΔCre datasets, replicates consisting of independently infected animals were not averaged on a per peak basis but rather all replicate information was retained in plotting signals (Fig. 5b and Extended Data Fig. 12a–d).

**Quantification of unique breaks across the genome.** To compare breaks across the genome, input reads were downsampled to the lowest number of reads for all samples compared (NPAS4 Cre versus ΔCre and wild-type Cre versus ΔCre: 21,036,891 reads; TIP60 Cre versus ΔCre 14,262,877 reads). We then quantified the total number of UMIs in each sample, which is plotted in Fig. 5c and Extended Data Fig. 12e. Note that for our quantification of signals across different regulatory elements displayed in all other figures, which used DESeq2-normalized counts, we did not downsample inputs before running DESeq2. The DESeq2 algorithm accounts for sequencing depth differences in normalization pipeline.

**Quantification of mutational accumulation.** Debarcer output (bamPositionComposition) tables were obtained from the package Debarcer_v.0.3.1 (https://github.com/oicr-gsi/debarcer/releases/tag/ v0.3.1)[38]. These tables document the number of consensus ten families (that is, groups of ten reads with the same UMI in which all ten reads show the same base changes (or lack thereof)) for each base in a given amplicon. These tables also calculate the total number of UMI families

that either match the reference genome or show a change from the reference. To include a given amplicon in our analysis, we required that the average number of consensus ten families across all bases in the amplicons was >100. For information on primer pooling and the amplicons included in the final analysis, see Supplementary Table 5.

Using scripts in R, we calculated the mutation frequency for a given amplicon by totalling the sum of all base changes from the reference and dividing by the total number of bases assessed in the amplicon (that is, the sum of consensus depth ten families across all bases in our table). The first 22 bases of the sequence, which contains the regions at which the primer anneals to amplify the region, was excluded from analysis to avoid errors in primer production complicating results. This calculation gives a single mutation rate for a given amplicon in a given sample. The total mutation rate includes both insertions and deletions and single nucleotide changes. To calculate the frequency of select point mutations, we counted the total number of select changes (C>A/(G>T)) divided by the total number of the given base included in the amplicon. Because it is not possible to know on which strand a mutation occurred, complementary base changes were collapsed into a single category (for example, C>T was combined with G>A). Insertion and deletion frequency was also calculated as a separate category. We also calculated a per amplicon normalized mutation rate in which we divided the total mutation rate for each animal by the median total mutation frequency in young mice. For ageing gradient samples, wild-type mice aged 3 months old were considered young, 12 months old were considered middle aged and 23–27 months old were considered old. Extreme outlier points of both normalized mutation frequency and non-normalized frequency were removed across all samples using a ROUT's test at 0.1% confidence (Fig. 5e and Extended Data Fig. 13g). Outlier removal and statistical tests on mutational samples were performed in Prism (v.8.4.2).

### Reporting summary

Further information on research design is available in the Nature Portfolio Reporting Summary linked to this article.

## Data availability

All sequencing data have been deposited into the GEO with accession number GSE175965. Mass spectrometry data have been deposited to the ProteomeXchange Consortium through the PRIDE database with accession number PXD038718. Raw gel images are provided in Supplementary Fig. 1. Source data are provided with this paper.

## Code availability

Custom code used in this study is available upon request.

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

**Acknowledgements** We thank all Greenberg Laboratory members for valuable input; S. Hinshaw for advice and assistance with biochemistry experiments; J. Lough for *Tip60*[fl/fl] mice; T. Fazzio for *Tip60-H3F* mice; D. Ferguson for *Mre11*[fl/fl] mice; N. Crosetto and B. Bouwman for access to the sBLISS-seq protocols; T. Godfrey for advice on amplicon library production; T. Vierbuchen, J. Green, J. Tycko and A. Greben for manuscript comments and helpful insight throughout the course of the project; and J. Luquette for advice and assistance on mutational analysis. E.A.P. acknowledges a Good Ventures Life Sciences Research Fellowship and K99 from NIA 1K99AG064042-01A1. D.T.G. was supported by a Harvard Department of Neurobiology Graduate fellowship. C.P.D. was supported by NIH fellowships T32-NS007473 and F32-NS112455. E.-L.Y. was supported by a Stuart H. Q. and Victoria Quan fellowship, a Harvard Department of Neurobiology Graduate fellowship and an Aramont Fund for Emerging Science Research fellowship. M.A.N. acknowledges NIH fellowship T32GM007753. E.E.D. was supported by the Damon Runyon Cancer Research Foundation and a Warren Alpert Distinguished Scholar Fellowship Award. Histology imaging was performed through the Harvard Medical School Neuro Imaging Facility (NINDS P30 Core Center grant number NS072030). This work was supported by R01 NS028829, the Lefler Faculty Small Grant to M.E.G. and the Carol and Gene Ludwig Family Foundation through the Ludwig Neurodegenerative Disease Seed Grants Program at Harvard Medical School. The Greenberg Laboratory is supported by the Allen Discovery Center Program, a Paul G. Allen Frontiers Group advised programme of the Paul G. Allen Family Foundation and the Tang-Yang Autism Center at Harvard Medical School.

**Author contributions** E.A.P., D.T.G. and M.E.G. conceived and designed the study and wrote the manuscript. E.A.P. and D.T.G. designed, executed and analysed the majority of experiments performed in the paper. Specifically, E.A.P. designed and characterized the *Npas4–FH* and *Arnt2–FH* lines, performed and analysed biochemistry, mass spectrometry, RNA-seq, ATAC-seq, CUT&RUN, ChIP–seq, sBLISS-seq, lifespan, FACS sorting, amplicon and mutation library preparation and analyses. D.T.G. performed and analysed biochemistry, mass spectrometry, RNA-seq, snRNA-seq, sBLISS-seq, END-seq, amplicon and mutation library production, all stereotaxic surgery and assisted in electrophysiology experiments. A.T.L. performed electrophysiology experiments. C.L. performed biochemical assays, RNA-seq and assisted in amplicon library production. N.P. performed biochemical assays, RNA-seq and assisted in amplicon library production. D.A.H. performed additional mutational analysis. E.-L.Y. and C.P.D. performed FOS and H3K27ac CUT&RUN experiments. C.P.D. provided code for CUT&RUN data analyses. I.R.V. assisted with sBLISS-seq library production. M.A.N. assisted in the design and production of the *Npas4–FH* and *Arnt2–FH* tagged lines. E.L. performed FOS and EGR1 immunoprecipitation–mass spectrometry experiments. E.E.D. assisted in GO analysis of snRNA-seq data. E.C.G. contributed to project direction and manuscript writing. C.J.W. provided intellectual guidance and reagents for biochemical purification of NPAS4 complexes. B.L.S. provided intellectual guidance for electrophysiology experiments.

**Competing interests** The authors declare no competing interests.

**Additional information**
**Correspondence and requests for materials** should be addressed to Michael E. Greenberg.

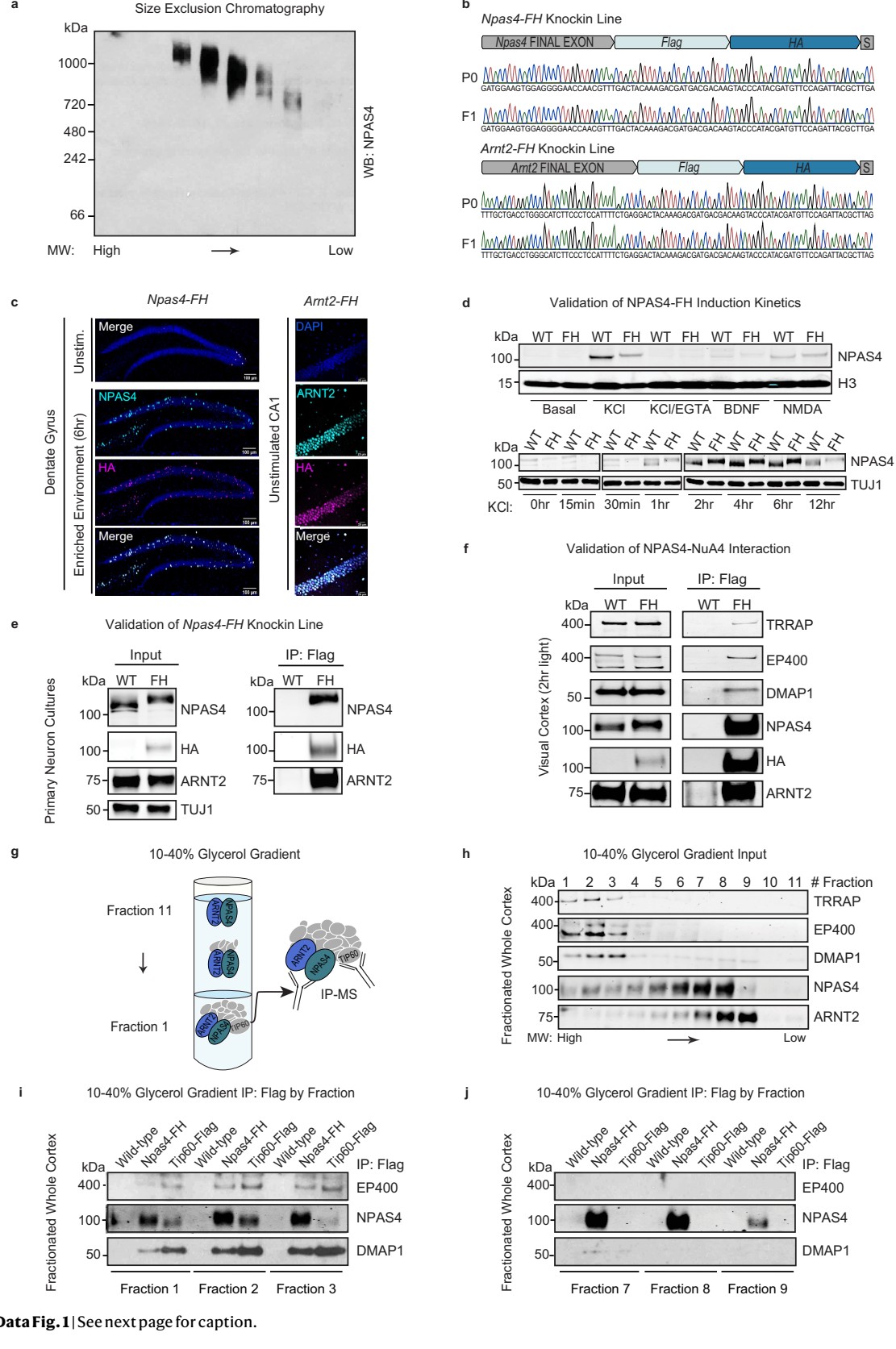

**Extended Data Fig. 1** | See next page for caption.

**Extended Data Fig. 1 | NPAS4 forms a complex with NuA4 across brain regions.** a. Lysate from the hippocampus of KA-stimulated mice, fractionated by molecular weight (MW) using gel filtration and non-denaturing size exclusion chromatography. Western blotting for NPAS4 in the different MW fractions confirms that NPAS4 resides in a high MW complex peaking ~1 MDa in size. Representative image from 3 experiments. See gel source data (Supplementary Fig. 1). b. Sequence validation of Flag HA epitope tag appended to the C-terminus of *Npas4* or *Arnt2* in the P0 or F1 generation in *Npas4-FH* or *Arnt2-FH* mice. c. Left: Immunohistochemistry of NPAS4 and HA antibody staining in hippocampus of *Npas4-FH* mice, 6 h enriched environment. Scale bars: 100 μm. Right: Validation of *Arnt2-FH* knockin mouse line by immunohistochemistry in CA1. Scale bar 25 μm. Representative images from 3 experiments. d. Validation by Western blot that NPAS4 and NPAS4-FH have the same induction kinetics and specificity for membrane depolarization-induced calcium signaling. To stimulate cultured cortical neurons, 55 mM KCl was applied for the indicated time points. Representative image from one experiment. See gel source data (Supplementary Fig. 1). e. Confirmation by Western blot of NPAS4-FH expression and Flag immunoprecipitation (IP) in cultured cortical neurons from either wild-type or *Npas4-FH* mice. Representative image from one experiment.

See gel source data (Supplementary Fig. 1). f. Confirmation via anti-Flag immunoprecipitation (IP) and Western blot that NPAS4-NuA4 also assembles in the visual cortex following 2 h light stimulation. Representative image from 2 experiments. See gel source data (Supplementary Fig. 1). g. Experimental diagram detailing glycerol gradient fractionation and immunoprecipitation of intact NPAS4-NuA4 complexes from high molecular weight (MW) fractions. h. Western blot of glycerol gradient fractions of NPAS4-FH in mouse cortical lysates segregated by MW. NPAS4 migrates in high MW fractions along with NuA4 components TRRAP, EP400, and DMAP1. Representative image from 3 experiments. See gel source data (Supplementary Fig. 1). i. Reciprocal validation of NPAS4-NuA4 interaction via Flag immunoprecipitation and Western blotting from high MW fractions of cortical lysates from wild-type, *Npas4-Flag-HA (Npas4-FH)*, and *Tip60-Flag (Tip60-F)* mice. Representative image from 2 experiments. See gel source data (Supplementary Fig. 1). j. Reciprocal validation of NPAS4-NuA4 interaction via Flag immunoprecipitation and Western blotting from low MW fractions of cortical lysates from wild-type, *Npas4-Flag-HA (Npas4-FH)*, and *Tip60-Flag (Tip60-F)* mice. Representative image from 2 experiments. See gel source data (Supplementary Fig. 1).

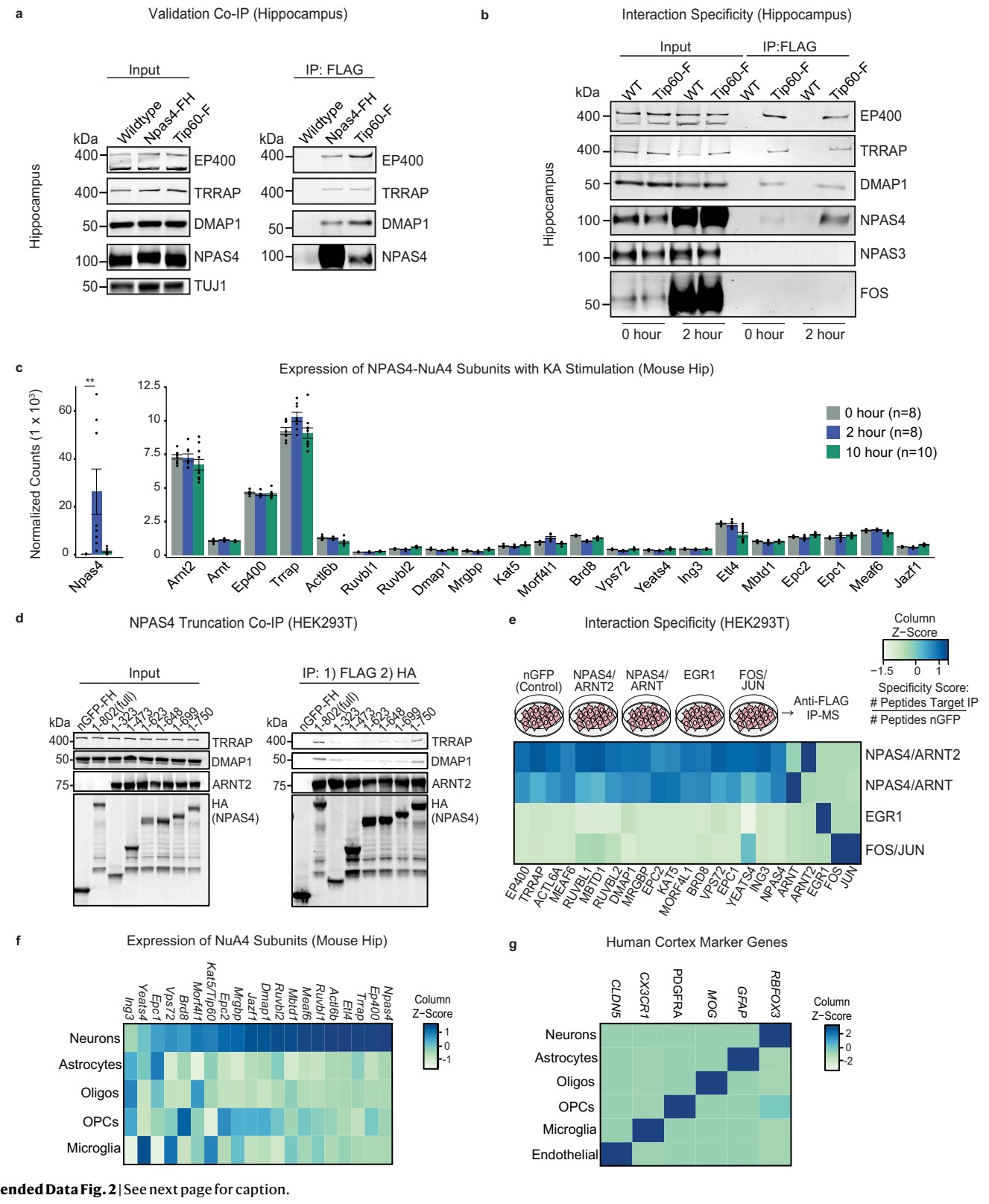

**Extended Data Fig. 2** | See next page for caption.

**Extended Data Fig. 2 | Validation of NPAS4-NuA4 interaction and neuronal specificity.** a. Confirmation that the NPAS4-NuA4 interaction can be observed via reciprocal Flag IP and Western blot from hippocampi of wild-type, *Npas4-FH*, and *Tip60-Flag (TIP60-F)* mice. Representative image from 2 experiments. See gel source data (Supplementary Fig. 1). b. Anti-Flag IP from hippocampi of *Tip60-Flag (TIP60-F)* mice in both unstimulated brains and 2 h post KA stimulation. Western blot demonstrates that components of the NuA4 complex interact in both the basal (0 h) and 2 h condition while NPAS4 interacts primarily in the stimulated state. Representative image from 2 experiments. See gel source data (Supplementary Fig. 1). c. Average RNA-seq DeSeq2 normalized counts ± s.e.m. of NPAS4-NuA4 components in the mouse hippocampus at 0 h (n = 8), 2 h (n = 8), and 10 h (n = 10), post KA stimulation. **$P$ = 3.19e-29. $P$ value determined from transcriptome-wide DeSeq2 analysis with Benjamini-Hochberg correction. See source data for individual $P$ values. d. Full length NPAS4 (amino acids 1-802) and the indicated truncations of the C-terminal portion of NPAS4 were expressed in HEK293T cells. Sequential Flag and HA IPs were performed (right), followed by Western blotting for NPAS4, ARNT2, and NuA4 subunits TRRAP and DMAP1. Flag-HA-tagged nuclear GFP (nGFP) was included as negative control. Representative image from 3 experiments. See gel source data (Supplementary Fig. 1). e. Heatmap of the column-normalized specificity score for each component of the NuA4 complex in anti-Flag IP-MS experiments conducted with NPAS4/ARNT, NPAS4/ARNT2, FOS/JUN, or EGR1 expressed in HEK293T cells. All proteins were Flag-HA tagged. Specificity score represents the ratio of the number of peptides identified in the IP over the number of peptides found in nGFP controls performed in parallel. Replicate numbers provided in Supplementary Table 1. f. Normalized counts (Seurat v3) of NPAS4-NuA4 components from hippocampal single-nucleus RNA sequencing in mouse hippocampus, normalized across column and displayed as a Z-score. g. Marker genes identifying cell types in human primary motor cortex single-nucleus RNA sequencing dataset published by the Allen Brain Institute (see Fig. 1d)[18], normalized across column and displayed as a Z-score.

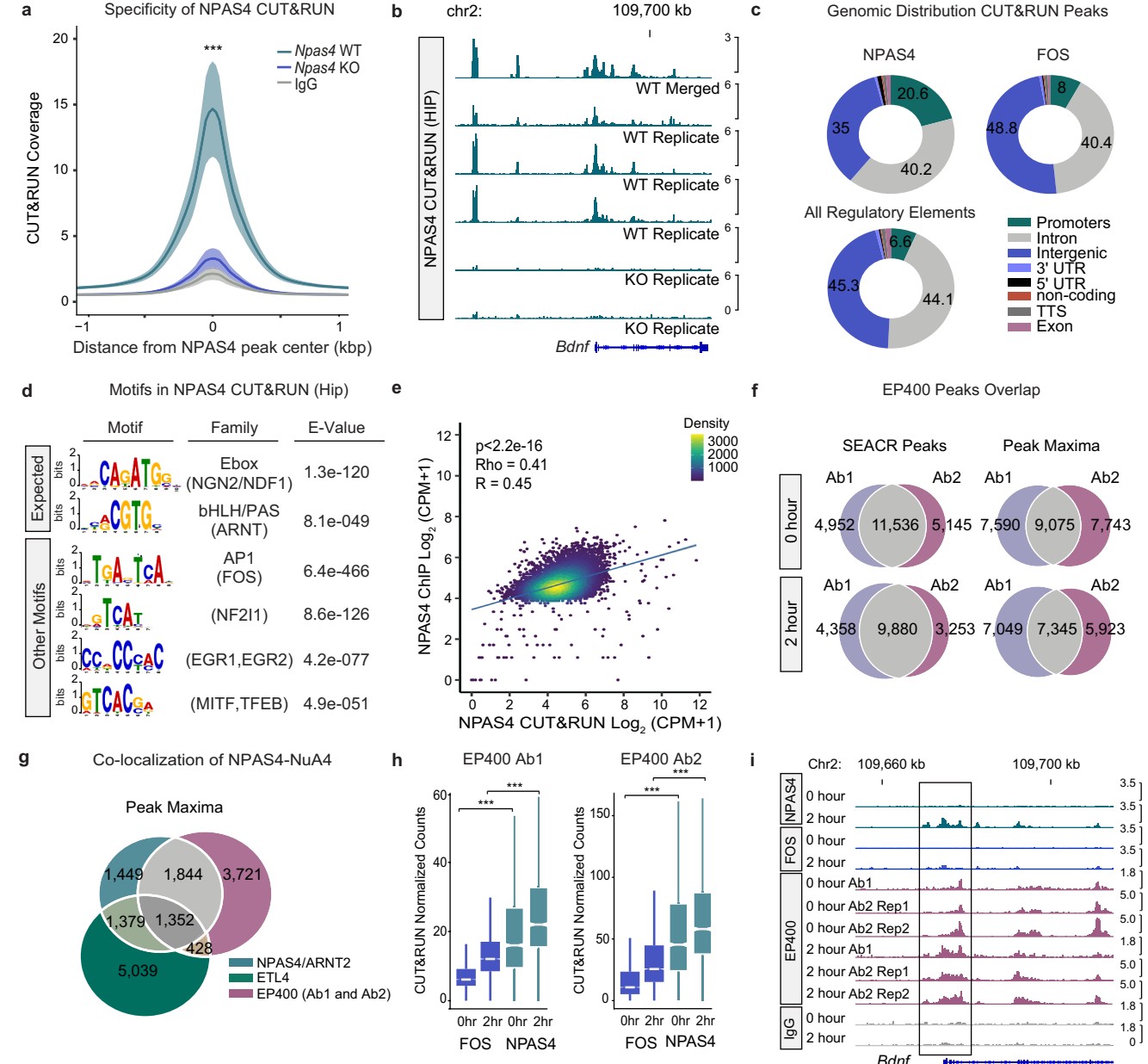

**Extended Data Fig. 3 | NPAS4 and NuA4 co-localize on chromatin across the genome.** a. Aggregate plot of average CUT&RUN coverage (fragment depth per bp/per peak) ± s.e.m. at NPAS4-binding sites in *Npas4* wild-type vs *Npas4* KO hippocampal tissue. (*Npas4* wild-type: n = 5, *Npas4* KO: n = 2, IgG: n = 8) 3-5 mice pooled per replicate. ***$P < 2.2e-16$, $P$ values were on calculated on average signal extracted in a 2 kb window centered on NPAS4 peak summits using unpaired, two-tailed Wilcoxon rank-sum tests. b. Integrative Genomics Viewer tracks of aggregated NPAS4 CUT&RUN signal (n = 5). 3-5 mice pooled per replicate. 3 additional replicates of NPAS4 CUT&RUN signal confirm the reproducibility of NPAS4 CUT&RUN. c. Distribution of NPAS4 (10,225 sites) and FOS (11,770 sites) CUT&RUN peak annotations relative to all regulatory elements (179,841 sites) in hippocampal tissue. d. Significant motifs enriched in 10,225 NPAS4 CUT&RUN peaks. E-value calculated using MEME-ChIP[72]. e. Correlation between NPAS4 CUT&RUN and NPAS4 ChIP-seq reads at NPAS4 ChIP-seq peaks (10,917 ChIP-seq peaks). Values represent the $\log_2$ read depth normalized counts for CUT&RUN vs ChIP-seq. Correlation was calculated by Pearson (R = 0.45) and Spearman (Rho = 0.41) tests, $P < 2.2e-16$ by two-tailed correlation tests. The reciprocal analysis of the correlation between NPAS4 CUT&RUN and NPAS4 ChIP-seq reads at NPAS4 CUT&RUN peaks (10,225 CUT&RUN peaks) yields a Pearson R = 0.46 and Spearman Rho = 0.43, $P < 2.2e-16$ by two-tailed correlation tests. f. Venn diagram of overlaps between SEACR peaks for anti-EP400 Ab1 (antibody 1, Bethyl Labs, A300-541A) and anti-EP400 Ab2 (antibody 2, Abcam Ab5201). Peak overlaps indicate at least 1 bp overlap between the entire SEACR-enriched region. Maxima indicates overlap between regions extended 500 bp out from the peak maxima for each factor. g. Venn diagram of overlaps between SEACR peaks for NPAS4/ARNT2 co-bound peaks with ETL4 and high-confidence EP400 peaks, defined as the union of peaks shared by both EP400 antibodies. Maxima indicates overlap between regions extended 500 bp out from the peak maxima for each factor. h. Boxplot of average EP400 normalized signal (counts per million) at sites bound by FOS but not NPAS4 (labeled 'FOS') and sites bound by NPAS4 but not FOS (labeled 'NPAS4'). FOS/No NPAS4 sites are defined as sites with a SEACR-determined peak of CUT&RUN signal for FOS but no peak for NPAS4 (6,998 sites) and vice versa (NPAS4/No FOS peaks: 5,550 sites). Boxplot shows median (line), IQR (box), 1.5x IQR (whiskers), notches indicate median ± 1.58× IQR/sqrt(n). 3-5 mice pooled per replicate. Replicate numbers provided in Supplementary Table 2. ***$P < 2.2e-16$. $P$ values were calculated using unpaired, two-tailed Wilcoxon rank-sum tests. i. Representative Integrative Genomics Viewer tracks of replicate EP400 CUT&RUN at the *Bdnf* promoter.

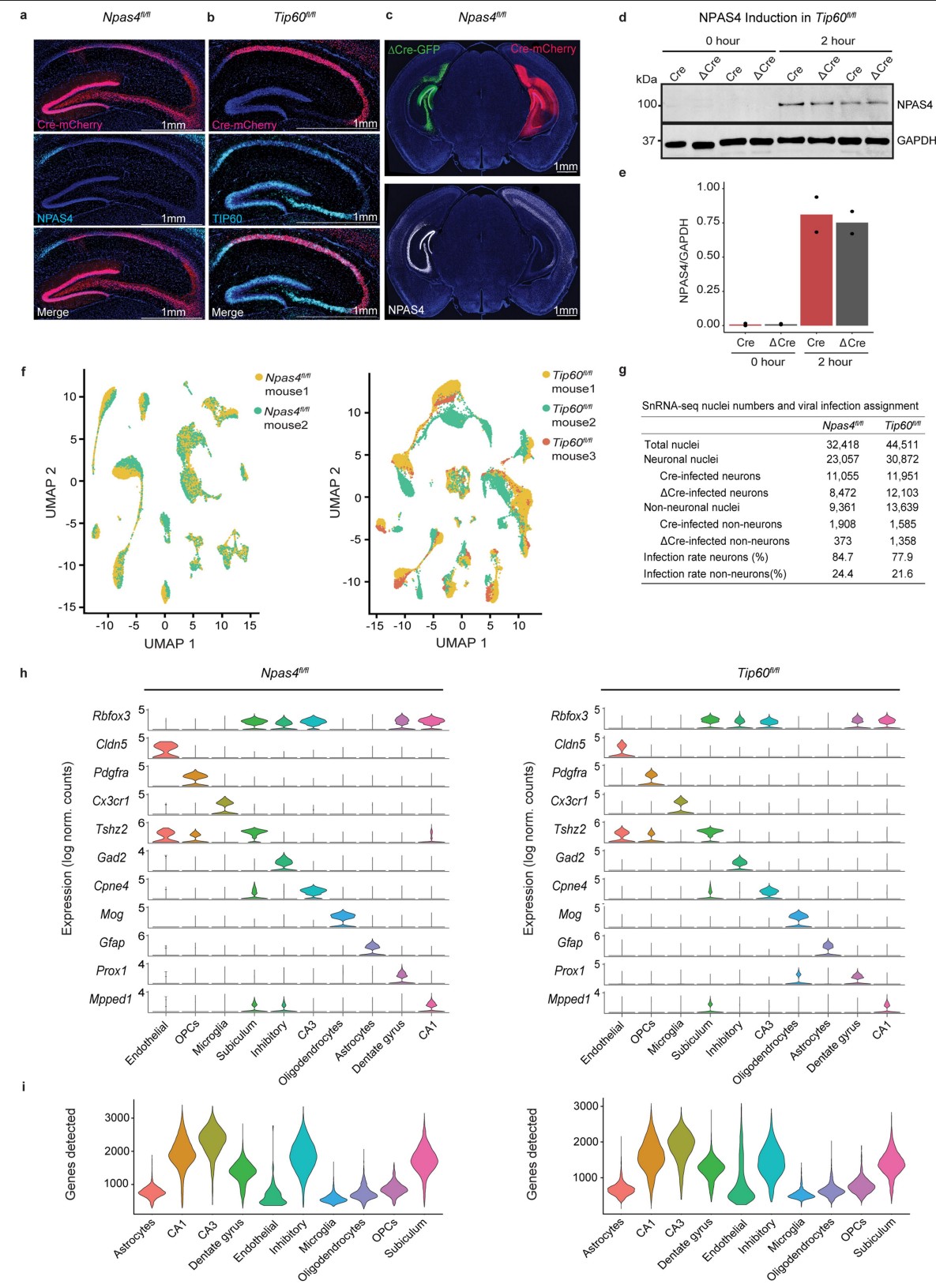

**Extended Data Fig. 4** | See next page for caption.

**Extended Data Fig. 4 | Acute deletion of NPAS4 or TIP60 by viral injection and additional quality control for single-nucleus RNA-seq datasets.**
a. Immunohistochemistry image of an *Npas4*^fl/fl^ mouse injected with AAV to express Cre-mCherry (shown in red) and collected 2 h post low-dose KA to induce NPAS4 (shown in cyan). Representative image from 3 animals. Scale bar = 1 mm. b. Immunohistochemistry image of a *Tip60*^fl/fl^ mouse injected with AAV to express Cre-mCherry (shown in red). TIP60 (shown in cyan). Representative image from 3 animals. Scale bar = 1 mm. c. Immunohistochemistry image of both hippocampal hemispheres of an *Npas4*^fl/fl^ mouse injected with Cre-mCherry and ΔCre-GFP in contralateral sides of the hippocampus and collected 2 h post low-dose KA to induce NPAS4 (shown in white). Representative image from 3 animals. Scale bar = 1 mm. d. Western blot from whole hippocampal tissue of *Tip60*^fl/fl^ mice injected with AAV expressing Cre-mCherry and ΔCre-GFP in contralateral sides of the hippocampus. Tissue was collected at either 0 or 2 h post KA stimulation. Cre and ΔCre tissue was collected from each individual mouse (0 h, n = 2; 2 h, n = 2). See gel source data (Supplementary Fig. 1). e. Quantification of the Western blot is shown in d, normalizing the NPAS4 signal to loading control GAPDH. f. Left: UMAP visualization of full *Npas4*^fl/fl^ snRNA-seq dataset. Nuclei are colored according to mouse of origin. 32,418 nuclei from 2 mice. Right: UMAP visualization of full *Tip60*^fl/fl^ snRNA-seq dataset. Nuclei are colored according to mouse of origin. 44,511 nuclei from 3 mice. g. Summary of final nuclei numbers in *Npas4*^fl/fl^ and *Tip60*^fl/fl^ snRNA-seq datasets, and quantification of infection rates with Cre-mCherry and ΔCre-GFP viruses. Higher infection rate in neurons reflects the known tropism of the AAV2/9 virus used in these experiments. h. Cell-type assignment in *Npas4*^fl/fl^ (left) and *Tip60*^fl/fl^ (right) snRNA-seq datasets using indicated marker genes. The y-axis denotes normalized expression (Seurat log$_e$ normalized counts). i. Distribution of number of genes detected per nucleus, by cell-type. *Npas4*^fl/fl^ (left) and *Tip60*^fl/fl^ (right).

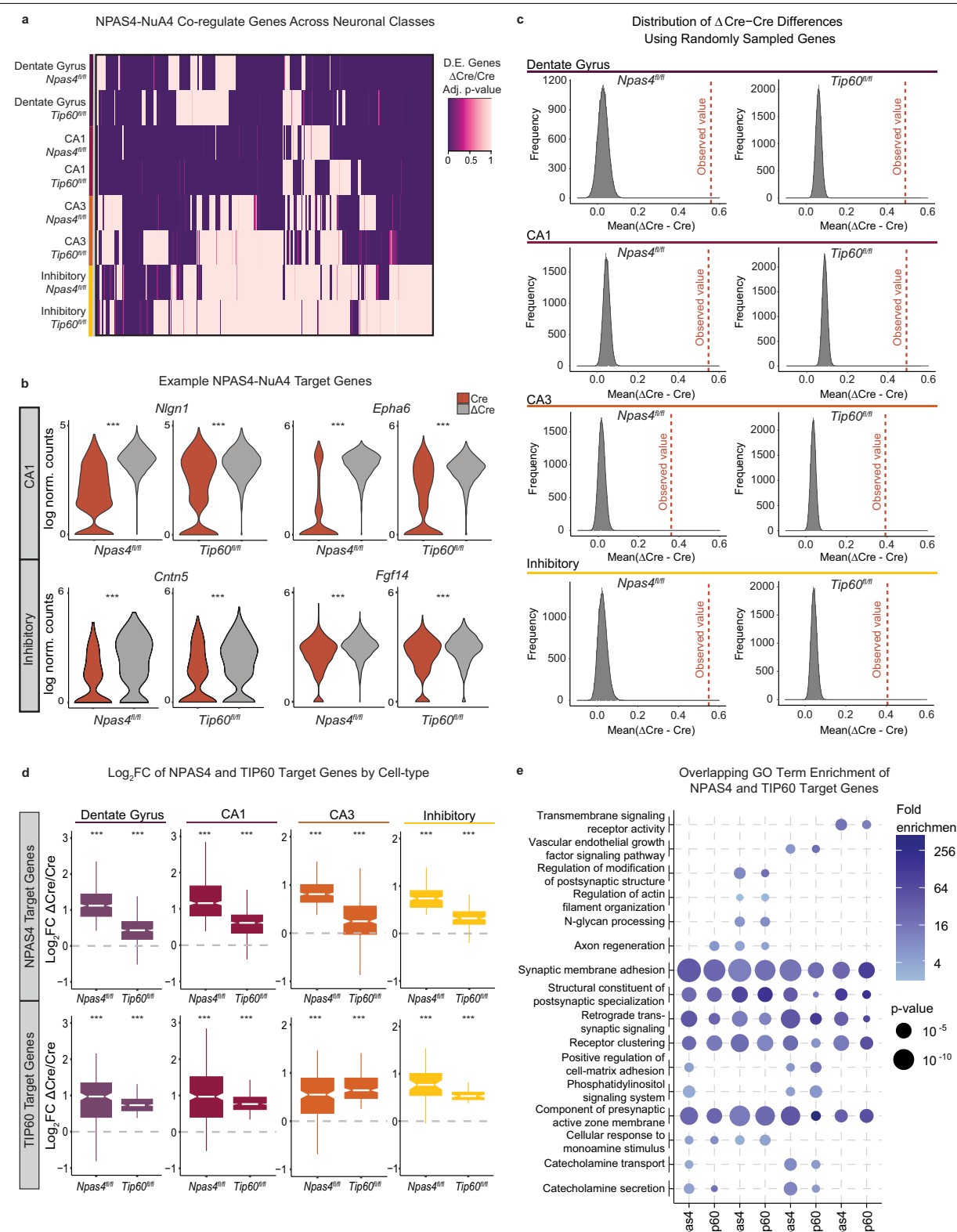

**Extended Data Fig. 5 |** See next page for caption.

**Extended Data Fig. 5 | NPAS4-NuA4 coordinate gene regulation across neuronal subtypes** *in vivo*. a. Heatmap showing coordinate regulation of NPAS4 target genes across the principal neuronal subtypes of the hippocampus in both *Npas4*[fl/fl] and *Tip60*[fl/fl] mice. NPAS4 target genes were identified in each cell-type using both Seurat (v3) and Monocle3 (see Methods). Bonferroni adjusted *P* values from Seurat (v3) differential expression testing (unpaired, two-tailed Wilcoxon rank-sum test) between ΔCre- and Cre-infected nuclei are shown in each cell-type. Each column represents adjusted *P* values for one gene. b. Violin plots showing the distribution of expression (Seurat $\log_e$ normalized counts) of the indicated gene across nuclei in the indicated cell-type. Bonferroni adjusted *P* value ***P* < 2.2e-16. Differential gene expression tests were conducted using an unpaired, two-tailed Wilcoxon rank-sum test implemented via the Seurat (v3) FindMarkers function. c. Comparison of the observed differences (ΔCre – Cre) in normalized counts for NPAS4 or TIP60 target genes in each neuronal subtype to the differences obtained when using an equal number of randomly selected genes. Genes were randomly selected from the top 10% of expressed genes (to account for NPAS4 or TIP60 target genes being highly expressed on average), and the average difference (ΔCre – Cre) in expression for each gene was calculated for each random sample. This sampling was repeated 10,000 times to generate sampling distributions (gray). In each subtype, the average difference (ΔCre – Cre) observed when using that subtype's NPAS4 or TIP60 target genes lies far outside the distribution obtained using randomly selected genes, suggesting the differences in expression of the target genes between ΔCre- and Cre-infected nuclei is not due to chance. d. Boxplots showing $\log_2$ fold change of NPAS4 or TIP60 target genes (see Methods) comparing ΔCre- and Cre-infected nuclei in dentate gyrus, CA1, CA3, and inhibitory neurons. Boxplot shows median (line), IQR (box), 1.5x IQR (whiskers), notches indicate median ± 1.58× IQR/sqrt(n). ****P* < 2.2e-16. *P* values were calculated using two-tailed, unpaired t-tests comparing to a null hypothesis of $\log_2$ fold change = 0 (no difference between Cre and ΔCre). Exact *P* values and cell numbers per cluster provided in source data. e. Select overlapping GO terms enriched in both NPAS4 target genes and TIP60 target genes across dentate gyrus, CA1, CA3, and inhibitory neurons. Circle size indicates the adjusted *P* value of enrichment determined by Fisher's one-tailed test using gProfiler2[65]. Color indicates the fold enrichment. See Methods for additional detail and Supplementary Table 3 for complete list of enriched GO terms for each cell-type.

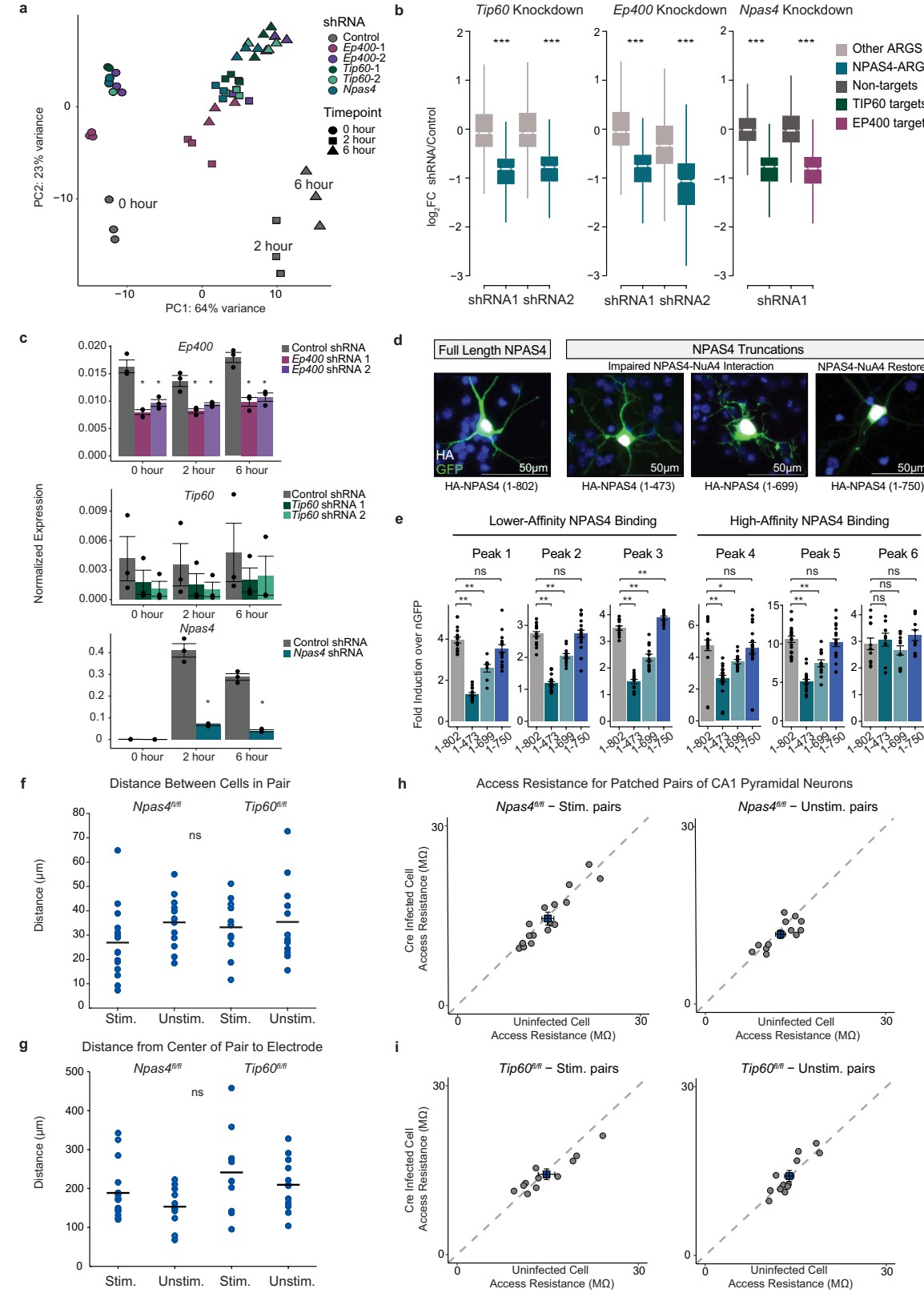

**Extended Data Fig. 6** | See next page for caption.

**Extended Data Fig. 6 | NPAS4-NuA4 coordinately regulates gene transcription and somatic inhibition.** a. Principal component analysis clustering of RNA-seq datasets from cultured mouse neurons expressing shRNAs targeting *Npas4*, *Tip60*, or *Ep400*, and stimulated with 55 mM KCl for either 0, 2, or 6 h. Control shRNA targets luciferase. b. Boxplots of $\log_2$ fold changes between the control shRNA (n = 3) and the indicated *Npas4* (n = 3), *Tip60* (n = 3) or *Ep400* (n = 3) shRNAs in neuronal cultures 6 h following membrane-depolarization by 55 mM KCl. Replicates consist of primary cultures generated on independent days. Boxplot shows median (line), IQR (box), 1.5x IQR (whiskers), notches indicate median ± 1.58× IQR/sqrt(n). Activity-regulated-genes (ARGs) are defined as genes upregulated at least 1.5-fold with a Benjamini-Hochberg adjusted $P < 0.01$ comparing the 6 h and 0 h time points in control shRNA-treated neurons. *Tip60* and *Ep400* targets are defined as all genes down-regulated by at least 1.5-fold with a Benjamini-Hochberg adjusted $P < 0.01$ by both shRNAs. Non-targets include all expressed genes not significantly affected by loss of *Tip60* or *Ep400*. ***$P < 2.2e-16$. *P* values were calculated using unpaired, two-tailed Wilcoxon rank-sum tests. c. Average expression ± s.e.m. of *Ep400* (n = 3), *Tip60* (n = 3), and *Npas4* (n = 3) by qPCR. Replicates consist of primary cultures generated on independent days. Expression normalized to both *Tubb3* and *Gapdh*. *Ep400* 0 h shRNA1: *$P = 0.0026$, shRNA2: *$P = 0.0072$; 2 h shRNA1: *$P = 0.0072$, shRNA2: *$P = 0.0154$; 6 h shRNA1: *$P = 0.0027$, shRNA2: *$P = 0.0038$. *Tip60* 0 h shRNA1: ns = 0.4021, shRNA2: ns = 0.2669; 2 h shRNA1: ns = 0.4507, shRNA2: ns = 0.3324; 6 h shRNA1: ns = 0.433, shRNA2: ns = 0.5431. *Npas4* 0 h shRNA1: *$P = 0.0214$; 2 h shRNA1: *$P = 0.0004$; 6 h shRNA1: *$P = 0.0001$. *P* values by two-tailed, unpaired t-tests.

d. HA staining of cultured cortical neurons infected with nuclear GFP (nGFP) and Flag-HA-tagged NPAS4 (full length and with indicated truncations) (see Extended Data Fig. 2d). Representative image from one experiment. Scale bar = 50 μm. e. Luciferase activity of NPAS4-bound enhancers in cultured neurons transfected with either nGFP, full length NPAS4, or indicated NPAS4 truncations (see Extended Data Fig. 2d). Colors represent the different NPAS4 truncations. Average luciferase activity normalized to nGFP-expressing control samples ± s.e.m. (see Methods). Each point represents a value from an independently transfected well collected from at least 3 independent primary neuronal cultures, except for Peak1 Npas4_1-699 (2 cultures). **$P < 0.0045$; *$P = 0.049$. *P* values were calculated using two-tailed, unpaired t-tests with Benjamini-Hochberg correction for multiple hypothesis testing. Individual *P* values and replicate numbers provided in source data. f. Points represent distance between cells in an uninfected:Cre-infected cell pair. *Npas4*$^{fl/fl}$: Stim: n = 16 pairs from 4 mice, Unstim: n = 12 pairs from 5 mice. *Tip60*$^{fl/fl}$: Stim: n = 11 pairs from 2 mice, Unstim: n = 12 pairs from 2 mice. $P = 0.305$. *P* value was calculated using a one-way ANOVA. g. Lateral distance from center of each pair to the stimulating electrode placed in the center of stratum pyramidale. *Npas4*$^{fl/fl}$: Stim: n = 16 pairs from 4 mice, Unstim: n = 12 pairs from 5 mice. *Tip60*$^{fl/fl}$: Stim: n = 11 pairs from 2 mice, Unstim: n = 12 pairs from 2 mice. $P = 0.055$. *P* value was calculated using a one-way ANOVA. h,i. Access resistance in MΩ for each pair of simultaneously patched CA1 pyramidal neurons in (h) *Npas4*$^{fl/fl}$: Stim: n = 16 pairs from 4 mice, $P = 0.7396$; Unstim: n = 12 pairs from 5 mice, $P = 0.9677$ and (i) *Tip60*$^{fl/fl}$: Stim: n = 11 pairs from 2 mice, $P = 0.7533$; Unstim: n = 12 pairs from 2 mice, $P = 0.8906$. *P* values by unpaired, two-tailed t-tests.

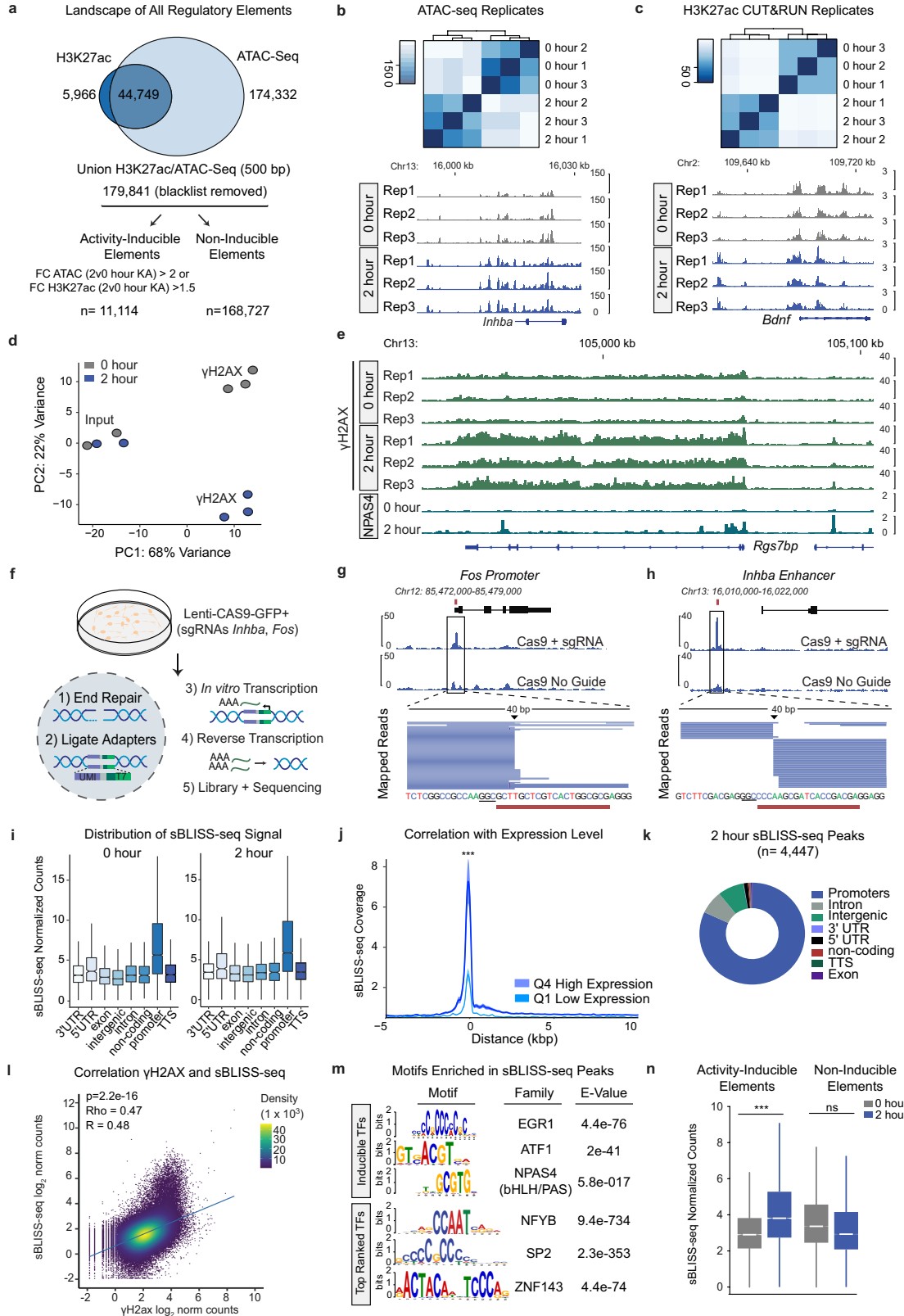

**Extended Data Fig. 7 |** See next page for caption.

**Extended Data Fig. 7 | NPAS4-NuA4 binds to regions undergoing recurrent DNA breaks *in vivo*.** a. Definition of all regulatory elements used throughout the manuscript. Venn Diagram depicting the overlap of reproducible ATAC-seq peaks (merged between 0 and 2 h peaks) and reproducible H3K27ac CUT&RUN (merged between 0 and 2 h peaks). ATAC-seq and CUT&RUN peak sets were defined as peaks consistently found in 3 of 3 replicates per timepoint. 3-5 mice pooled per replicate. All regulatory elements are defined as the union of reproducible ATAC-seq and H3K27ac peaks across all timepoints. Activity-inducible regulatory elements are defined as elements that exhibit a greater than two-fold change in ATAC-seq signal (2 vs 0 h stimulation; adjusted $P < 0.05$) and/or a 1.5-fold change in H3K27ac CUT&RUN signal (2 vs 0 h stimulation; adjusted $P < 0.05$). $P$ values calculated by DeSeq2's Wald test with default Benjamini-Hochberg correction. b. Upper Panel: Heatmap of Euclidean distance between replicates of ATAC-seq in hippocampal nuclei isolated from unstimulated (0 h) or 2 h post stimulation across all regulatory elements defined in Extended Data Fig. 7a. Lower panel: Representative Integrative Genomics Viewer tracks of individual ATAC-seq replicates at activity-inducible gene *Inhba*. c. Upper Panel: Heatmap of Euclidean distance between replicates of H3K27ac CUT&RUN in hippocampal nuclei isolated from unstimulated brains (0 h) or 2 h post stimulation across all regulatory elements defined in Extended Data Fig. 7a. Lower panel: Representative Integrative Genomics Viewer tracks of individual H3K27ac CUT&RUN replicates at activity-inducible gene *Bdnf*. d. Principal component analysis of γH2AX ChIP-seq signal across all regulatory elements (see Extended Data Fig. 7a) in unstimulated and stimulated hippocampal nuclei. Replicates cluster together and separate by stimulation state. e. Representative Integrative Genomics Viewer tracks of individual γH2AX ChIP-seq replicates and aggregate NPAS4 CUT&RUN signal (n = 5) at activity-inducible gene *Rgs7bp*. 3-5 mice pooled per replicate. f. Schematic of sBLISS-seq on cultured neurons infected with either Cas9-only viruses or Cas9 virus + gRNAs. g. Integrative Genome Browser tracks displaying sBLISS-seq signal at the *Fos* promoter in cultured neurons infected with either Cas9+gRNA to the *Fos* locus or a Cas9-only control. Red line indicates the position of the gRNA. Zoomed-in perspective shows the reads mapping on either side of the predicted gRNA cut site, indicated by the arrow. PAM sites are underlined in the DNA sequence. Representative image from 3 experiments. Replicates consist of independent cultures generated on separate days.

h. Integrative Genome Browser tracks displaying sBLISS-seq signal at the *Inhba* enhancer in cultured neurons infected with either Cas9+gRNA to the *Inhba* enhancer locus or a Cas9-only control. Representative image from 3 experiments. Replicates consist of independent cultures generated on separate days. i. Boxplots showing sBLISS-seq normalized signal (see Methods) across all regulatory elements in wild-type neurons at 0 (n = 8) and 2 h (n = 8) after KA stimulation. 1 mouse per replicate. Boxplot shows median (line), IQR (box), 1.5x IQR (whiskers), notches indicate median ± 1.58× IQR/sqrt(n). Promoters = 11,853, Introns = 79,438, Intergenic = 81,385, 3'UTR = 1,574, 5'UTR = 695, non-coding = 445, TTS = 1,831, Exons = 2,620. See Extended Data Fig. 7a. j. Aggregate plot of average sBLISS-seq coverage (fragment depth per bp per peak) ± s.e.m. (n = 8) at the TSS of genes in the highest quartile of expression (Q4) vs genes in the lowest quartile (Q1) in 2 h stimulated hippocampi. Gene expression quartiles were determined from DeSeq2 normalized counts of paired RNA-seq samples collected from the same tissue as sBLISS-seq samples at the 2 h KA stimulation timepoint (see Fig. 3d). ***$P < 2.2e$-16 was calculated on average signal extracted in the 20 kb window around gene TSS using an unpaired, two-tailed Wilcoxon rank-sum test. k. Distribution of DSB peak annotations at 2 h of stimulation (4,447 peaks). Reproducible sBLISS-seq peaks were defined by the MACS2 peak calling algorithm and were found in at least 5 of 8 replicates. l. Correlation between sBLISS-seq signal ($\log_2$ normalized counts) and γH2AX signal ($\log_2$ normalized counts) across all regulatory elements in the hippocampus (179,841 sites; see Extended Data Fig. 7a). Correlation was calculated by Pearson (R = 0.47) and Spearman (Rho = 0.48), with $P < 2.2e$-16 for both two-tailed correlation tests. m. Most significant motifs enriched in reproducible sBLISS-seq peaks at 2 h of stimulation. Motifs of activity-inducible transcription factors (ATF1, EGR1, and NPAS4/AHR) are enriched in sBLISS-seq peaks. E-value calculated using MEME-ChIP[72]. n. Boxplots of average sBLISS-seq normalized counts at activity-inducible elements (11,114 sites) vs non-inducible elements (168,727 sites) at 0 h (n = 8 mice) and 2 h (n = 8 mice) post stimulation. Boxplot shows median (line), IQR (box), 1.5x IQR (whiskers), notches indicate median ± 1.58× IQR/sqrt(n). Non-inducible elements include all elements that do not fall within the inducible peak set in our regulatory landscape. ***$P = 2.2e$-16, ns: $P = 1$. $P$ values by unpaired, one-tailed Wilcoxon rank sum test (2 h > 0 h).

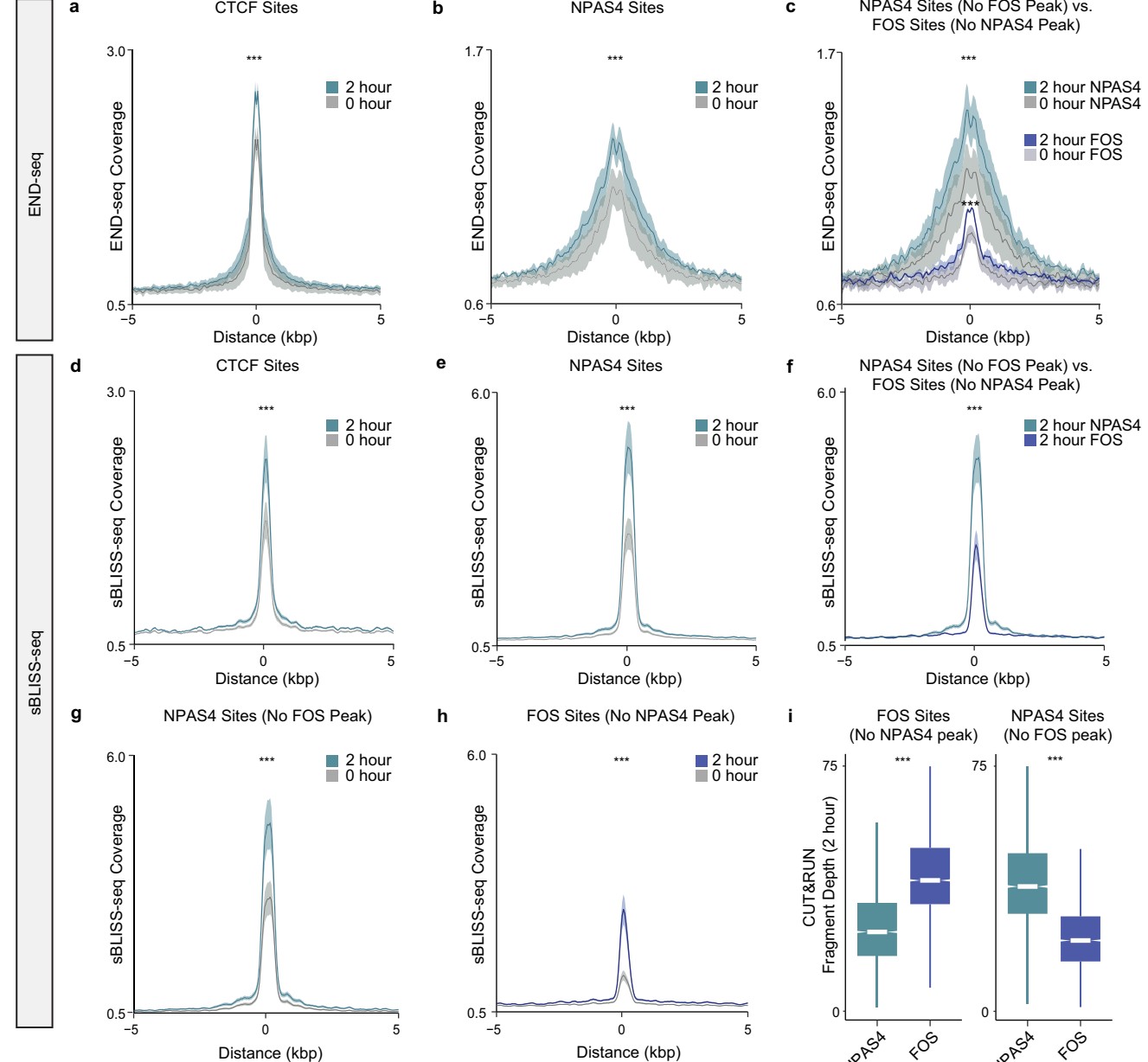

**Extended Data Fig. 8 | END-seq mapping of DSB signal in cultured cortical neurons shows enrichment at NPAS4-bound sites and activity-dependent dynamics.** a. Aggregate plots (bin size 50 bp) of average END-seq coverage (fragment depth per bp per peak) ± s.e.m. (n = 2) in cultured mouse cortical neurons at CTCF-bound sites (13,228 sites) at either 0 or 2 h of stimulation with 55mM KCl. See Methods for full description of END-seq method and experimental parameters. CTCF-bound sites are defined as sites with a SEACR-defined peak of CTCF CUT&RUN signal in hippocampal datasets. ***P < 2.2e-16. For a-c, replicates consist of independent primary cultures. P values by unpaired, two-tailed Wilcoxon rank-sum tests on signal extracted in a 500 bp window centered on each peak. b. Aggregate plots (bin size 50 bp) of average END-seq coverage ± s.e.m. (n = 2) in cultured mouse cortical neurons at NPAS4-bound sites (10,225 sites) at either 0 or 2 h post stimulation with 55mM KCl. NPAS4-bound sites are defined as sites with a SEACR-defined peak of NPAS4 CUT&RUN in *in vivo* hippocampal datasets. ***P < 2.2e-16. c. Aggregate plots (bin size 50 bp) of average END-seq coverage ± s.e.m. (n = 2) in cultured mouse cortical neurons at NPAS4 sites that lack a FOS peak (5,550 sites) and FOS sites that lack an NPAS4 peak (6,998 sites), at both 0 and 2 h post stimulation with 55mM KCl. ***P < 2.2e-16 for NPAS4 0 vs 2 h, ***P = 2.3e-15 for FOS 0 vs 2 h. d. Aggregate

plots (bin size 50bp) of average sBLISS-seq coverage ± s.e.m. (n = 8) at CTCF-bound sites at 0 and 2 h post stimulation with low-dose KA. ***P < 2.2e-16. For d-h, replicates derived from individual mice. P values by unpaired, two-tailed Wilcoxon rank-sum tests on signal extracted in a 500 bp window centered on each peak. e. Aggregate plots (bin size 50 bp) of average sBLISS-seq coverage ± s.e.m. (n = 8) at NPAS4-bound sites at 0 and 2 h post stimulation. **P < 2.2e-16. f. Aggregate plots (bin size 50 bp) of average sBLISS-seq coverage ± s.e.m. (n = 8) at NPAS4 sites that lack a FOS peak and FOS sites that lack an NPAS4 peak at 2 h post stimulation. ***P < 2.2e-16. g. Aggregate plots (bin size 50 bp) of average sBLISS-seq coverage ± s.e.m. (n = 8) at NPAS4 sites that lack a FOS peak at 0 and 2 h post stimulation. ***P < 2.2e-16. h. Aggregate plots (bin size 50 bp) of average sBLISS-seq coverage ± s.e.m. (n = 8) at FOS sites that lack an NPAS4 peak at 0 and 2 h post stimulation. ***P < 2.2e-16. i. Boxplots showing average NPAS4 (n = 5) and FOS (n = 3) CUT&RUN signal (counts per million) at NPAS4 sites that lack a FOS peak and FOS sites that lack an NPAS4 peak at 2 h post stimulation in hippocampus. 3-5 mice pooled per replicate. Boxplot shows median (line), IQR (box), 1.5x IQR (whiskers), notches indicate median ± 1.58× IQR/sqrt(n). ***P < 2.2e-16. P values were calculated using unpaired, two-tailed Wilcoxon rank-sum tests.

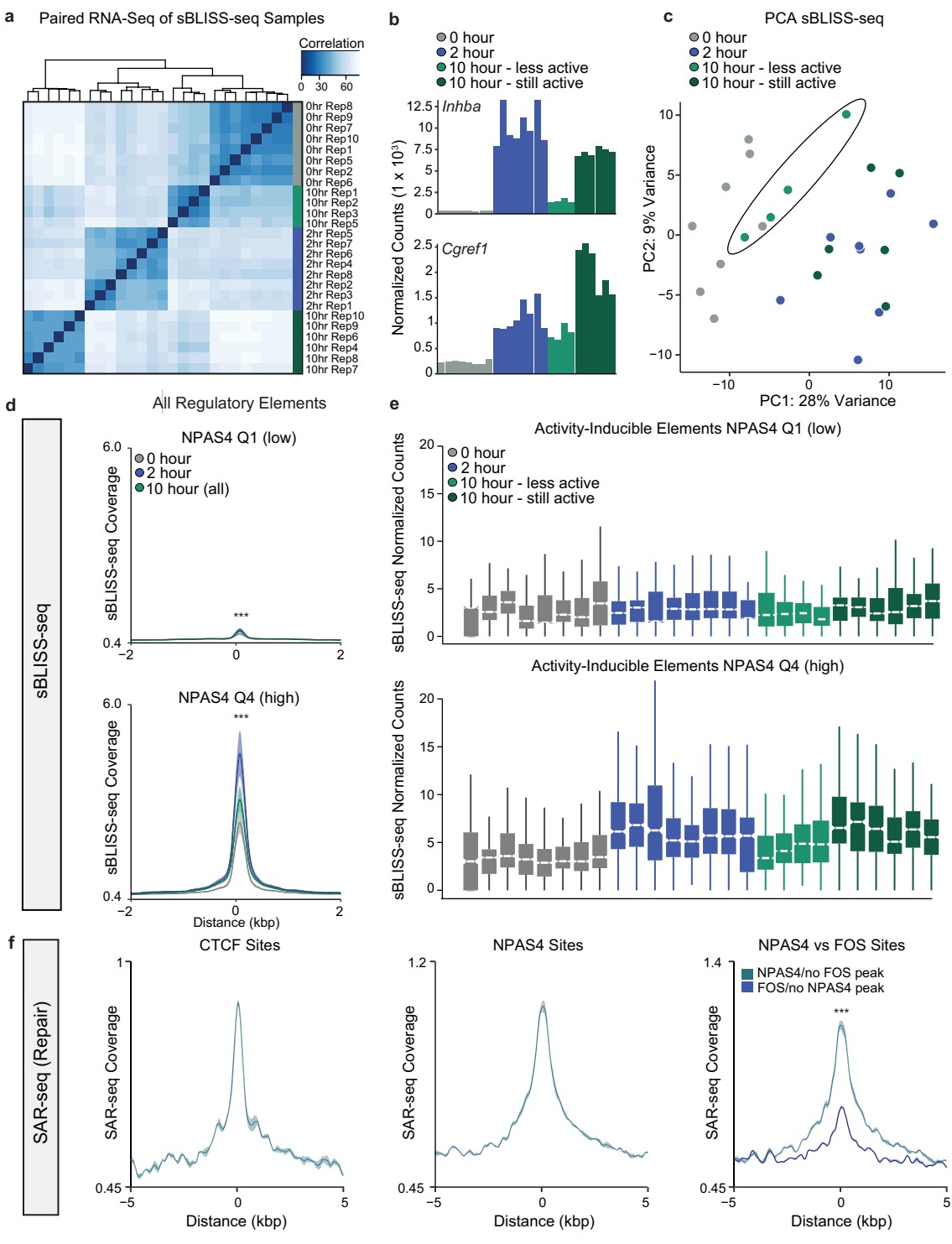

**Extended Data Fig. 9 | NPAS4-NuA4-bound sites undergo DNA repair as inducible transcription subsides.** a. Heatmap of Euclidean distance between replicates of RNA-seq prepared from the same tissue used to generate the sBLISS-seq timecourse. The samples cluster primarily according to stimulation state, with the 10 h post stimulation samples falling into two groups with either dampening or sustained transcriptional induction. b. DeSeq2 normalized counts of additional inducible genes, *Inhba* and *Cgref1*, are shown for each replicate. c. Principal component analysis clustering of sBLISS-seq samples at 0 h, 2 h and 10 h post stimulation. The sBLISS-seq samples cluster according to stimulation state, with the 10 h post stimulation samples clustering either with the 0 h or 2 h samples. Paired RNA-seq analysis indicates that this separation is driven by altered levels of transcriptional induction in these samples. d. Aggregate plots (bin size 50 bp) of average sBLISS-seq coverage (fragment depth per bp per peak) ± s.e.m. at all regulatory elements, subset by quartiles

of NPAS4 CUT&RUN signal. Q1 = 44,864 sites, Q4 = 7,378 sites. 0 h (n = 8), 2 h (n = 8), 10 h (n = 10). 1 mouse per replicate. *P* values were calculated using unpaired, two-tailed Wilcoxon rank sum tests. NPAS4 Q1: 0vs2 h: *P* < 2.2e-16, 2 vs 10 h: *P* = 1.7e-15, 0 vs 10 h: *P* < 2.2e-16; NPAS4 Q4: 0 vs 2 h *P* < 2.2e-16, 2 vs 10 h: *P* < 2.2e-16, 0 vs 10 h: *P* < 2.2e-16. e. Boxplots of sBLISS-seq DeSeq2 normalized counts in 0 h, 2 h, 10 h 'less active', and 10 h 'still active' samples at activity-inducible regulatory elements, subset by quartiles of NPAS4 binding. Boxplot shows median (line), IQR (box), 1.5x IQR (whiskers), notches indicate median ± 1.58×IQR/sqrt(n). Each boxplot represents an individual replicate consisting of an individual mouse. Boxplots of average signal across all replicates are shown in Fig. 4b. f. Aggregate plots (bin size 50 bp) of average SAR-seq[32] coverage ± s.e.m. (n = 3) at CTCF binding sites, all NPAS4 binding sites, and NPAS4 vs FOS binding sites. See Extended Data Fig. 8i for definition of NPAS4 vs FOS binding sites.

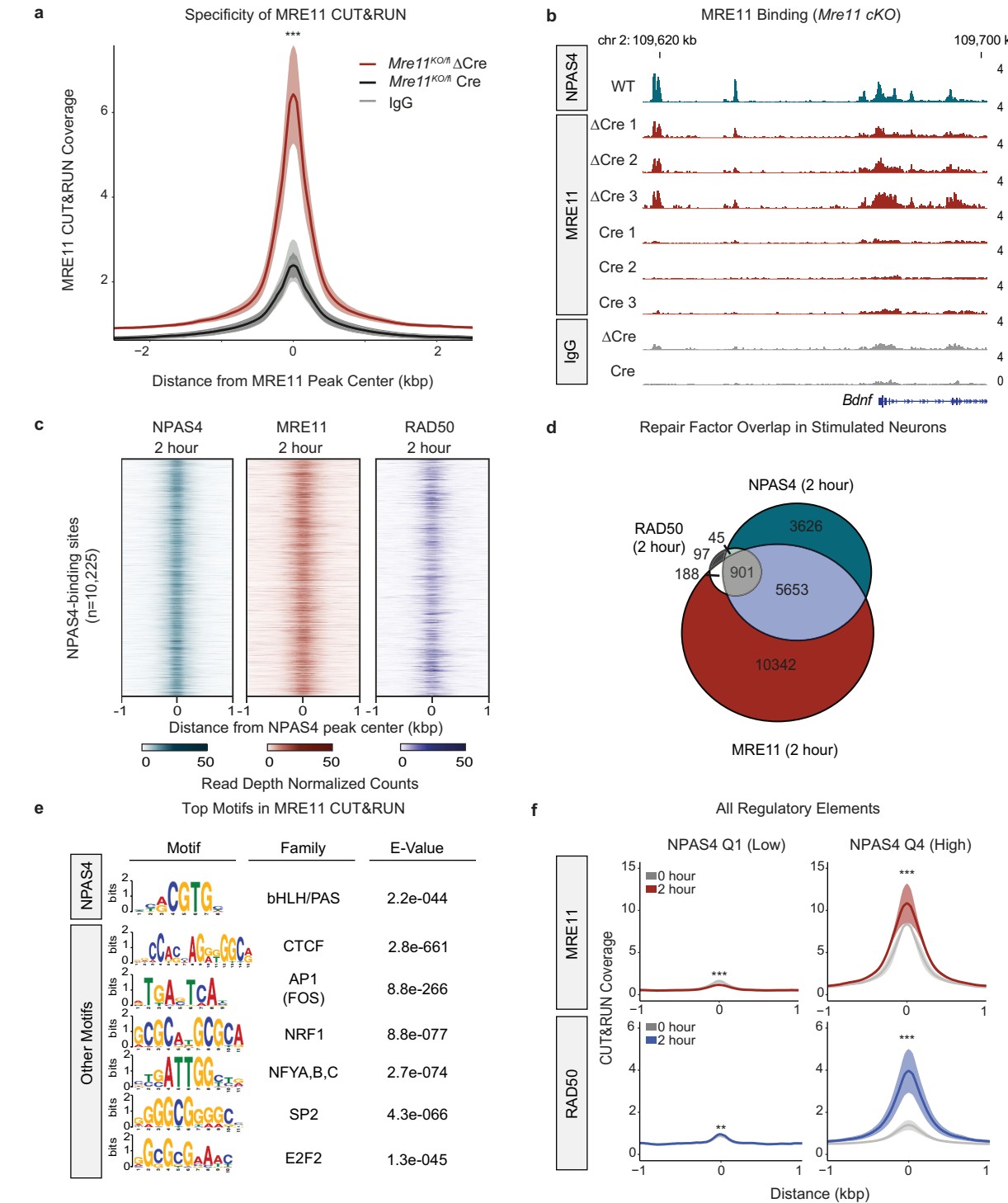

**Extended Data Fig. 10** | See next page for caption.

**Extended Data Fig. 10 | NPAS4-NuA4 co-binds the genome with DNA repair sensors MRE11 and RAD50 in stimulated neurons.** a. Aggregate plot of average MRE11 CUT&RUN coverage (fragment depth per bp per peak) ± s.e.m. in KA-stimulated nuclei isolated from *Mre11^{KO/fl}* infected with either ΔCre-GFP virus (Control) or Cre-mCherry (*Mre11* cKO) at MRE11 binding sites (17,084 sites). MRE11 binding sites were determined by the SEACR peak calling algorithm. IgG indicates the CUT&RUN signal from a nonspecific IgG control and represents the average IgG across both ΔCre-GFP and Cre conditions. MRE11 (n = 3 Cre and n = 3 ΔCre-GFP), IgG (n = 3 Cre and n = 3 ΔCre-GFP). 1 mouse per replicate. ***P < 2.2e-16. *P* values were calculated using unpaired, two-tailed Wilcoxon rank-sum tests. b. Integrative Genomics Viewer browser image displaying CUT&RUN signal for NPAS4, MRE11 (Cre or ΔCre), and IgG (Cre or ΔCre) in 2 h KA-stimulated nuclei at the *Bdnf* gene. c. Aggregate CUT&RUN signal for NPAS4, RAD50 and MRE11 in 0 h vs 2 h stimulated hippocampal tissue at NPAS4-binding sites. Each NPAS4-binding site is represented as a single horizontal line centered at the peak summit and extended out ±1 kb. Intensity of color correlates with sequencing signal as indicated by the scale bar for each factor (0 to 50 read-depth normalization). MRE11(n = 2), RAD50 (n = 4), NPAS4 (n = 5). 3-5 mice pooled per replicate. d. Venn diagram of overlaps between binding sites of MRE11, RAD50 and NPAS4 in 2 h KA-stimulated neurons. Peaks for each factor were determined using the SEACR peak calling algorithm and represent peaks found reproducibly across replicates (2 of 2 MRE11 replicates, 3 of 4 RAD50 replicates, and 4 of 5 NPAS4 replicates). 3-5 mice pooled per replicate. e. Most significant motifs enriched in MRE11 CUT&RUN peaks (2 h KA-stimulated). Motif enrichment was performed on 1 kb peaks extended 500 bp up and downstream from the peak maxima. Notable motifs include the NPAS4/bHLH-PAS motif, the AP1 family, and CTCF motifs. f. Aggregate of average CUT&RUN coverage (fragment depth per bp per peak) ± s.e.m. at all regulatory elements, subset by quartiles of NPAS4 CUT&RUN signal. Q1 = 44,864 sites, Q4 = 7,378 sites. MRE11(0 h and 2 h, n = 2), RAD50 (0 h, n = 3), RAD50 (2 h, n = 4). ***P < 2.2e-16; **P = 5.78e-06. *P* values by unpaired, two-tailed Wilcoxon rank-sum tests. 3-5 mice pooled per replicate.

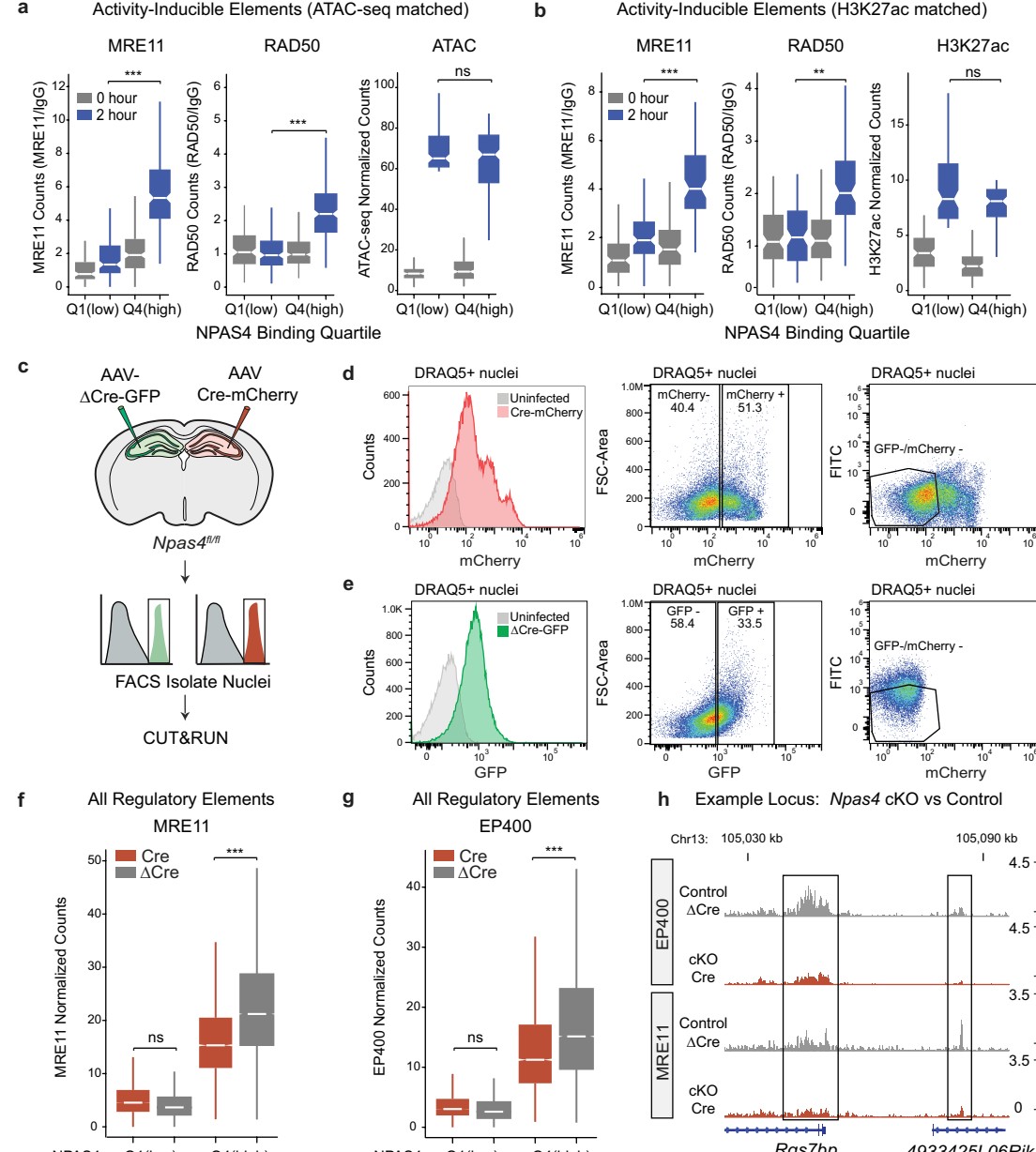

**a** Activity-Inducible Elements (ATAC-seq matched)

MRE11    RAD50    ATAC

**b** Activity-Inducible Elements (H3K27ac matched)

MRE11    RAD50    H3K27ac

**c** AAV-ΔCre-GFP    AAV Cre-mCherry

*Npas4^fl/fl*

FACS Isolate Nuclei

CUT&RUN

**d** DRAQ5+ nuclei

**e** DRAQ5+ nuclei

**f** All Regulatory Elements
MRE11

**g** All Regulatory Elements
EP400

**h** Example Locus: *Npas4* cKO vs Control

**Extended Data Fig. 11** | See next page for caption.

**Extended Data Fig. 11 | NPAS4-NuA4 recruits repair factors to chromatin in stimulated neurons.** a. Boxplots of average IgG-normalized MRE11 and RAD50 CUT&RUN signal (left) or ATAC-seq normalized counts (right) at activity-inducible elements plotted, subset by quartiles of NPAS4 CUT&RUN signal. The sites displayed were selected to have equivalent or higher inducible ATAC-seq signal in low quartiles (Q1) compared to top NPAS4-binding sites (Q4). Q1 = 111 sites, Q4 = 200 sites. Boxplot shows median (line), IQR (box), 1.5x IQR (whiskers), notches indicate median ± 1.58× IQR/sqrt(n). MRE11(0 h and 2 h, n = 2), RAD50 (0 h, n = 3), RAD50 (2 h, n = 4), ATAC-seq (0 and 2 h, n = 3). 3-5 mice pooled per replicate. MRE11 2 h Q1 vs Q4: ***$P$ < 2.2e-16; RAD50 2 h Q1 vs Q4: ***$P$ < 2.2e-16; ATAC 2 h NPAS4 Q1 vs NPAS4 Q4: $P$ = 0.105. $P$ values by unpaired, two-tailed Wilcoxon rank-sum tests. b. Boxplots of average IgG-normalized MRE11 and RAD50 CUT&RUN signal (left) or H3K27ac normalized counts (right) at activity-inducible elements plotted, subset by quartiles of NPAS4 CUT&RUN signal. The sites displayed were selected to have equivalent or higher inducible H3K27ac signal in low quartiles (Q1) compared to top NPAS4-binding sites (Q4). Q1 = 79 sites, Q4 = 88 sites. Boxplot shows median (line), IQR (box), 1.5x IQR (whiskers), notches indicate median ± 1.58× IQR/sqrt(n). MRE11(0 h and 2 h, n = 2), RAD50 (0 h, n = 3), RAD50 (2 h, n = 4), ATAC-seq (0 and 2 h, n = 3). 3-5 mice pooled per replicate. MRE11 2 h Q1 vs Q4: ***$P$ < 2.2e-16; RAD50 2 h Q1 vs Q4: **$P$ = 2.603e-12; H3K27ac 2 h Q1 vs Q4: $P$ = 0.0567. $P$ values by unpaired, two-tailed Wilcoxon rank-sum tests. c. Experimental design used to isolate Cre-mCherry- and ΔCre-GFP-infected nuclei using florescence-activated cell sorting (FACS). Nuclei are subsequently used for CUT&RUN of EP400, MRE11. d. FACS of Cre-mCherry-positive nuclei (gated from DRAQ5, a fluorescent DNA dye used to identify nuclei). Left panel shows the histogram distribution of Cre-mCherry signal in infected tissue relative to an uninfected control tissue sample. Middle panel shows the sorting scheme, with negative events defined by a DRAQ5-stained, uninfected tissue sample run in parallel on the day of sorting. Right panel shows the FITC vs mCherry signal for all DRAQ5+ nuclei and demonstrates no doubly-infected cells in the samples. See Supplementary Fig. 2 for gating scheme. e. FACS of ΔCre-GFP nuclei (gated from DRAQ5+ nuclei). Left panel shows the histogram distribution of ΔCre-GFP signal in infected tissue relative to an uninfected control tissue sample. Middle panel shows the sorting scheme, with negative events defined by a DRAQ5-stained, uninfected tissue sample run in parallel on the day of sorting. Right panel shows the FITC vs mCherry signal for all DRAQ5+ nuclei and demonstrates no doubly-infected cells in the samples. See Supplementary Fig. 2 for gating scheme. f. Boxplots of average MRE11 CUT&RUN normalized counts in Cre or ΔCre-infected hippocampi of *Npas4*$^{fl/fl}$ mice at activity-inducible sites, subset by quartiles of NPAS4 binding. Q1 = 44,864 sites, Q4 = 7,378 sites. Boxplot shows median (line), IQR (box), 1.5x IQR (whiskers), notches indicate median ± 1.58× IQR/sqrt(n). MRE11 CRE (n = 3), MRE11 ΔCre (n = 3). 2-3 mice pooled per replicate. MRE11 Q1: $P$ = 1, Q4: ***$P$ < 2.2e-16. $P$ values by unpaired, one-tailed Wilcoxon rank-sum tests. g. Boxplots of average EP400 CUT&RUN normalized counts in Cre or ΔCre-infected hippocampi of *Npas4*$^{fl/fl}$ mice at activity-inducible sites, subset by quartiles of NPAS4 binding. Q1 = 44,864 sites, Q4 = 7,378 sites. Boxplot shows median (line), IQR (box), 1.5x IQR (whiskers), notches indicate median ± 1.58× IQR/sqrt(n). EP400 ΔCre (n = 3), EP400 Cre (n = 4). 2-3 mice pooled per replicate. EP400 Q1: $P$ = 1, Q4: ***$P$ < 2.2e-16. $P$ values by unpaired, one-tailed Wilcoxon rank-sum tests (ΔCre > Cre). h. Integrative Genomics Viewer browser image displaying CUT&RUN signal for MRE11 and EP400 in *Npas4*-cKO (Cre) or Control (ΔCre) in 2 h KA-stimulated nuclei at the *Rgs7bp* gene. MRE11 CRE (n = 3), MRE11 ΔCre (n = 3), EP400 ΔCre (n = 3), EP400 Cre (n = 4). 2-3 mice pooled per replicate.

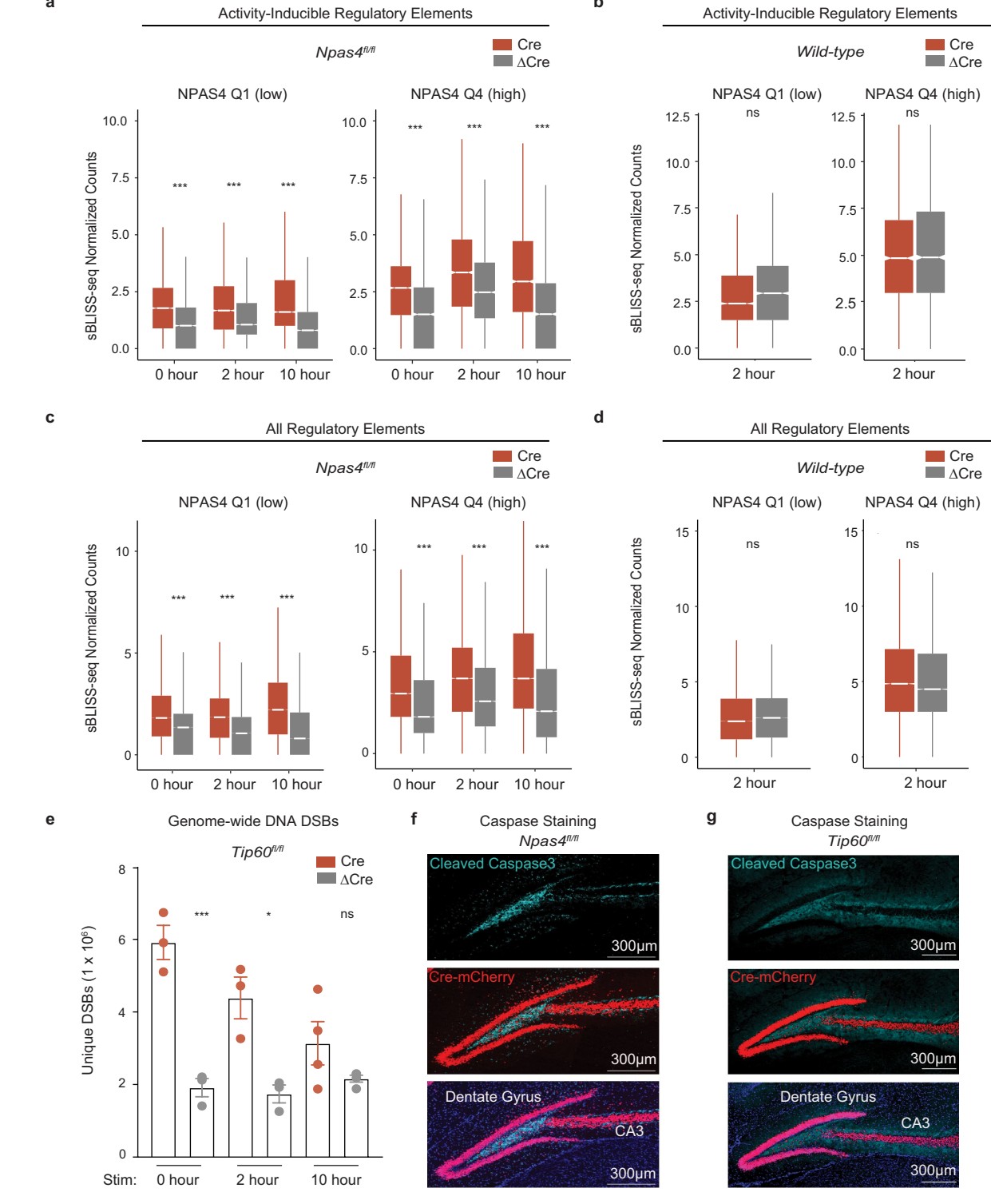

**Extended Data Fig. 12** | See next page for caption.

**Extended Data Fig. 12 | Loss of NPAS4-NuA4 increases DNA breaks across the neuronal genome.** a. Boxplots of sBLISS-seq normalized counts in nuclei isolated from *Npas4*$^{fl/fl}$ mice injected with Cre-mCherry (*Npas4*-cKO) or ΔCre-GFP virus (Control) at activity-inducible regulatory elements, subset by quartiles of NPAS4 CUT&RUN signal (see Methods, Extended Data Fig. 7a). Q1 = 1,017 sites, Q4 = 764 sites. Data plotted includes all datapoints coming from 5 replicates per genotype; no averaging across replicates was performed. Boxplot shows median (line), IQR (box), 1.5x IQR (whiskers), notches indicate median ± 1.58× IQR/sqrt(n). ***$P$ < 2.2e-16 $P$ values by unpaired, one-tailed Wilcoxon rank-sum tests (Cre > ΔCre). b. Boxplots of sBLISS-seq normalized counts in nuclei isolated from wild-type mice injected with Cre-mCherry or ΔCre-GFP virus at activity-inducible regulatory elements, subset by quartiles of NPAS4 CUT&RUN signal. Q1 = 1,017 sites, Q4 = 764 sites. Data plotted includes all datapoints coming from 3 replicates per genotype; no averaging across replicates was performed. Boxplot shows median (line), IQR (box), 1.5x IQR (whiskers), notches indicate median ± 1.58× IQR/sqrt(n). Q1: $P$ = 1; Q4 $P$ = 0.995. $P$ values by unpaired, one-tailed Wilcoxon rank-sum tests (Cre > ΔCre). c. Boxplots of sBLISS-seq normalized counts in nuclei isolated from *Npas4*$^{fl/fl}$ mice injected with Cre-mCherry (*Npas4*-cKO) or ΔCre-GFP virus (Control) at all regulatory elements, subset by quartiles of NPAS4 CUT&RUN signal (see Methods for regulatory site definition). Q1 = 44,864 sites, Q4 = 7,378 sites.

Data plotted includes all datapoints coming from 5 replicates per genotype. Boxes represent the interquartile range with line at the median. Boxplot shows median (line), IQR (box), 1.5x IQR (whiskers), notches indicate median ± 1.58× IQR/sqrt(n). ***$P$<2.2e-16. $P$ values by unpaired, one-tailed Wilcoxon rank-sum tests (Cre > ΔCre). d. Boxplots of sBLISS-seq DeSeq2 normalized counts in nuclei isolated from wild-type injected with Cre-mCherry or ΔCre-GFP virus at all regulatory elements, subset by quartiles of NPAS4 CUT&RUN signal (see Methods for regulatory site definition). Data plotted includes all datapoints coming from 3 replicates per genotype. Boxplot shows median (line), IQR (box), 1.5x IQR (whiskers), notches indicate median ± 1.58× IQR/sqrt(n). Q1: $P$ = 1; Q4 $P$ = 0.1092. $P$ values by unpaired, one-tailed Wilcoxon rank-sum tests (Cre > ΔCre). e. Average ± s.e.m. of genome-wide breaks in hippocampal nuclei isolated from *Tip60*$^{fl/fl}$ mice (0 and 2 h; n = 3, 10 h; n = 4) infected with Cre or ΔCre virus. To compare across samples, input reads were downsampled to the lowest value among all conditions, and numbers of unique DNA breaks are shown. *Tip60*$^{fl/fl}$: 0 h: $P$ = 0.0017, 2 h: $P$ = 0.013, 10 h: $P$ = 0.158. $P$ values by two-tailed, unpaired t-tests. f. Cleaved caspase 3 (apoptosis marker) staining in *Npas4*$^{fl/fl}$ mice injected with AAV to express Cre-mCherry. Representative image from 3 animals. Scale bar = 300 μm. g. Cleaved caspase 3 staining in *Tip60*$^{fl/fl}$ mice injected with AAV to express Cre-mCherry. Representative image from 3 animals. Scale bar = 300 μm.

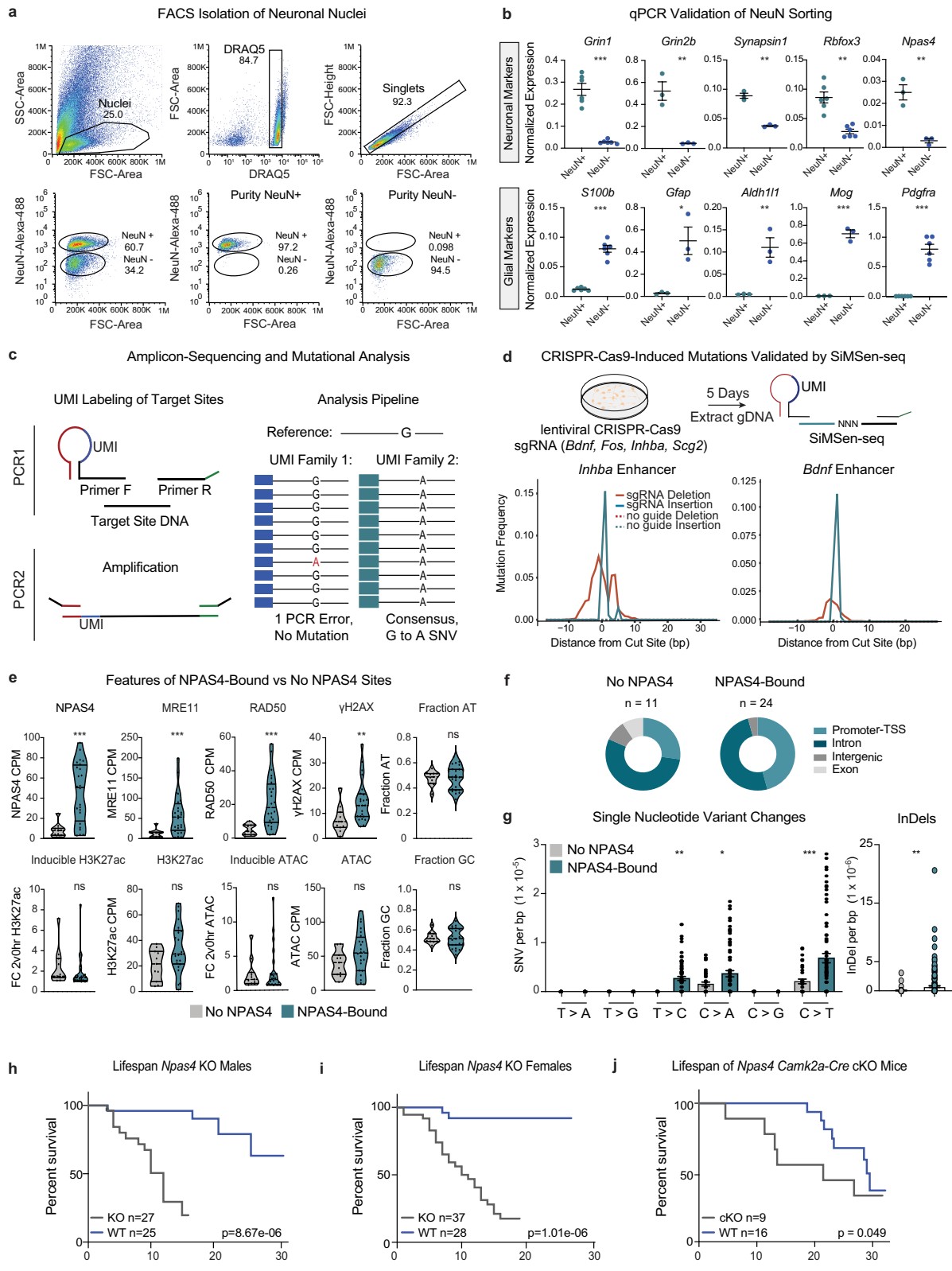

**a** FACS Isolation of Neuronal Nuclei

**b** qPCR Validation of NeuN Sorting

**c** Amplicon-Sequencing and Mutational Analysis

**d** CRISPR-Cas9-Induced Mutations Validated by SiMSen-seq

**e** Features of NPAS4-Bound vs No NPAS4 Sites

**f** No NPAS4 / NPAS4-Bound

**g** Single Nucleotide Variant Changes / InDels

**h** Lifespan *Npas4* KO Males

**i** Lifespan *Npas4* KO Females

**j** Lifespan of *Npas4 Camk2a-Cre* cKO Mice

**Extended Data Fig. 13** | See next page for caption.

**Extended Data Fig. 13 | Mutational Analysis at NPAS4-NuA4 sites during aging and *Npas4 KO* lifespan analysis.** a. FACS sorting scheme to isolate NeuN-expressing neurons from aging *Npas4* wild-type vs KO hippocampal tissue. Sorted cells from NeuN+ and NeuN- gates were re-analyzed for purity. See Supplementary Fig. 2 for gating scheme. b. Average normalized gene expression ± s.e.m. (relative to housekeeping gene *Gapdh*) of marker genes for neurons (NeuN+) or glial cell types (NeuN-). Each dot represents data from an individual mouse collected across 2 independent sorting days. ***$P$ < 0.001; **$P$ < 0.01; *$P$ < 0.05. $P$ values by unpaired, two-tailed t-tests with Benjamini-Hochberg multiple hypothesis correction. Individual $P$ values and replicates as follows: *Grin1* (n = 6): $P$ = 2.06e-05, *Grin2b* (n = 3): $P$ = 5.97e-03, *Syapsin1* (n = 3): 5.72e-04, *Rbfox3* (n = 6): 3.89e-04, *Npas4* (n = 3): $P$ = 5.21e-03, *S100b* (n = 6): $P$ = 3.85e-06, *Gfap* (n = 3): $P$ = 1.89e-02, *Aldh1l1* (n = 3): $P$ = 9.50e-03, *Mog* (n = 3): $P$ = 2.29e-04, *Pdgrfa* (n = 6): $P$ = 1.16e-05. c. Diagram of the SiMSen-seq amplicon sequencing approach used to identify somatic mutations that occur in individual DNA templates. d. Diagram of positive control experiment (top) used to test SiMSen-seq mutation detection method. Cultured cortical neurons were infected with lentivirus to express Cas9 with or without guide RNAs (*Bdnf, Fos, Inhba, Scg2*). SiMSen-seq was used to detect mutations at the gRNA cut sites at *Inhba* and *Bdnf*. Mutation frequency (that is, the number of insertions/deletions per UMI family) is plotted across a 40 bp window surrounding the cut site. e. Violin plots of read depth-normalized sequencing signal for NPAS4, MRE11, RAD50, H3K27ac CUT&RUN, γH2AX ChIP-seq and ATAC-seq in 2 h stimulated neurons at NPAS4-bound (24) and No NPAS4 (11) sites selected for mutational analysis. Each point represents a site. Inducible H3K27ac and Inducible ATAC plots show the distribution of fold changes (2 vs 0 h) in ATAC-seq and H3K27ac at these same sites. Dashed line represents the median; solid lines represent quartiles. ***$P$ < 0.0005; **$P$ < 0.005. $P$ values by unpaired, two-tailed Wilcoxon rank sum tests. NPAS4: $P$ = 0.002, MRE11: $P$ = 0.0009, RAD50: $P$ = 1.33e-04, γH2AX $P$ = 0.0027, Percent GC $P$ = 0.76; Percent AT $P$ = 0.76; Inducible H3K27ac $P$ = 0.25, H3K27ac $P$ = 0.07; ATAC $P$ = 0.19; Inducible ATAC $P$ = 0.53. Replicates per factor included in Supplementary Table 2. f. Genomic annotations of NPAS4-bound (24) vs No NPAS4 (11) sites. g. Left panel: Average SNV frequency ± s.e.m. in NPAS4-Bound and No NPAS4 in young (3-month-old) animals. Each point represents a single site sampled from a mouse and data from 4 mice are shown. *$P$ < 0.05; **$P$ < 0.005, ***$P$ < 0.001; T>A $P$ = 0.99, T>G $P$ = 0.99, T>C $P$ = 0.0049, C>A $P$ = 0.027, C>G $P$ = 0.99, C>T $P$ = 1.85e-09. $P$ values by a one-way ANOVA with Holm-Sidak's correction for multiple hypothesis testing. Right panel: Average Insertion/Deletion frequency ± s.e.m. in NPAS4-Bound and No NPAS4 sites in young (3-month-old) animals. Each point represents a single site sampled from a mouse and data from 4 mice are shown. ***$P$ = 2.96e-05. $P$ value by an unpaired, two-tailed Wilcoxon rank-sum test. h. Lifespan analysis on *Npas4* wild-type (n = 25) vs *Npas4* KO (n = 27) male littermates. Median lifespan KO = 12 months; Median lifespan of wild-type not determined. $P$ = 8.67e-06 by two-tailed Gehan-Breslow-Wilcoxon test; $P$ = 1.37e-06 by two-tailed log-rank Mantel-Cox test. i. Lifespan analysis on *Npas4* wild-type (n = 28) vs *Npas4* KO (n = 37) female littermates. Median lifespan KO = 11 months; Median lifespan of wild-type not determined. $P$ = 1.01e-06 by two-tailed Gehan-Breslow-Wilcoxon test; $P$ = 8.48e-08 by two-tailed log-rank Mantel-Cox test. j. Lifespan analysis on *Npas4* wild-type (n = 16, *Npas4*[+/+]; *Camk2a-Cre+; Sun1*[fl/+]) vs *Npas4* cKO (n = 9, *Npas4*[fl/fl]; *Camk2a-Cre+; Sun1*[fl/+]). Median lifespan cKO = 21.46 months; Median lifespan of wild-type = 29 months. $P$ = 0.049 by two-tailed Gehan-Breslow-Wilcoxon test; $P$ = 0.19 by a two-tailed log-rank Mantel-Cox test.

| | |
|---|---|

# Reporting Summary

## Statistics

For all statistical analyses, confirm that the following items are present in the figure legend, table legend, main text, or Methods section.

| n/a | Confirmed | |
|---|---|---|
| ☐ | ☒ | The exact sample size (*n*) for each experimental group/condition, given as a discrete number and unit of measurement |
| ☐ | ☒ | A statement on whether measurements were taken from distinct samples or whether the same sample was measured repeatedly |
| ☐ | ☒ | The statistical test(s) used AND whether they are one- or two-sided<br>*Only common tests should be described solely by name; describe more complex techniques in the Methods section.* |
| ☒ | ☐ | A description of all covariates tested |
| ☐ | ☒ | A description of any assumptions or corrections, such as tests of normality and adjustment for multiple comparisons |
| ☐ | ☒ | A full description of the statistical parameters including central tendency (e.g. means) or other basic estimates (e.g. regression coefficient) AND variation (e.g. standard deviation) or associated estimates of uncertainty (e.g. confidence intervals) |
| ☐ | ☒ | For null hypothesis testing, the test statistic (e.g. *F*, *t*, *r*) with confidence intervals, effect sizes, degrees of freedom and *P* value noted<br>*Give P values as exact values whenever suitable.* |
| ☒ | ☐ | For Bayesian analysis, information on the choice of priors and Markov chain Monte Carlo settings |
| ☒ | ☐ | For hierarchical and complex designs, identification of the appropriate level for tests and full reporting of outcomes |
| ☐ | ☒ | Estimates of effect sizes (e.g. Cohen's *d*, Pearson's *r*), indicating how they were calculated |

*Our web collection on statistics for biologists contains articles on many of the points above.*

## Software and code

Policy information about availability of computer code

| | |
|---|---|
| Data collection | VS ASW-FL (Image acquisition software for VS120 Slide Scanner Microscope). FlowJo (10.0.8r1). Sony SH800Z FACS acquisition software. Illumina Next-Seq Control Software (v4.0.2) |
| Data analysis | Homer(v4.9), Prism (v8.4.2), R(3.6.1), EdgeR(3.28.1), Limma(3.42.2), Trimmomatic (0.36), vbowtie2(2.2.9), sBLISS-seq mapping pipeline (https://github.com/BiCroLab/blissNP.git), CellRanger(3.0.0), Seurat(v3), Monocle3, MACS2 (v 2.1.1), SEACR_1.1, bedtools (2.27.1), DESeq2 (1.26.0), Subread(v1.5.1), Debarcer_v0.3.1 (https://github.com/oicr-gsi/debarcer/releases/tag/v0.3.1), UCSC Genome Browser LiftOver (https://genome.ucsc.edu/cgi-bin/hgLiftOver), Meme-ChIP (https://meme-suite.org/meme/tools/meme-chip), Sequest (Thermo Fisher Scientific, Waltham, MA) |

For manuscripts utilizing custom algorithms or software that are central to the research but not yet described in published literature, software must be made available to editors and reviewers. We strongly encourage code deposition in a community repository (e.g. GitHub). See the Nature Portfolio guidelines for submitting code & software for further information.

# Data

Policy information about [availability of data](availability of data)

All manuscripts must include a [data availability statement](data availability statement). This statement should provide the following information, where applicable:

- Accession codes, unique identifiers, or web links for publicly available datasets
- A description of any restrictions on data availability
- For clinical datasets or third party data, please ensure that the statement adheres to our [policy](policy)

All sequencing data for RNA-seq, ATAC-seq, ChIP-seq, CUT&RUN, sBLISS-seq, END-seq, snRNA-seq, and amplicon sequencing has been deposited in the Gene Expression Omnibus with accession number GSE175965. Mass spectrometry data has been deposited in PRIDE repository with accession number PXD038718. Raw gel images are provided in Supplementary Information Fig. 1. Additional data is provided as source data throughout the manuscript.

# Human research participants

Policy information about [studies involving human research participants and Sex and Gender in Research.](studies involving human research participants and Sex and Gender in Research.)

| | |
|---|---|
| Reporting on sex and gender | N/A |
| Population characteristics | N/A |
| Recruitment | N/A |
| Ethics oversight | N/A |

Note that full information on the approval of the study protocol must also be provided in the manuscript.

# Field-specific reporting

Please select the one below that is the best fit for your research. If you are not sure, read the appropriate sections before making your selection.

☒ Life sciences  ☐ Behavioural & social sciences  ☐ Ecological, evolutionary & environmental sciences

For a reference copy of the document with all sections, see [nature.com/documents/nr-reporting-summary-flat.pdf](nature.com/documents/nr-reporting-summary-flat.pdf)

# Life sciences study design

All studies must disclose on these points even when the disclosure is negative.

| | |
|---|---|
| Sample size | No statistical methods were used to predetermine sample size. Sample sizes were determined according to standards of practice in the field for each assay and generally adhere to guidelines of the ENCODE consortium (https://www.encodeproject.org/about/experiment-guidelines/). Sample size details are included in the manuscript with experimental description. |
| Data exclusions | For mutational analysis experiments, to include a given amplicon in our analysis, we required that the amplicon be found in greater than 1/3 of all samples and that the average number of consensus 10 families across all bases in the amplicons was >100. For information on primer pooling and the amplicons included in the final analysis, see Supplementary Table 5. For mutational analysis, extreme outlier points were removed across all conditions using a ROUT's test at 0.1% confidence performed in Prismv8.4.2. |
| Replication | Replicate details for each assay are provided in the manuscript. All attempts at replication were successful. For specific replicate numbers per assay, please see Supplementary Table 2. In general, in vitro RNA-seq experiments were performed in triplicate. RNA-seq from hippocampal tissue paired with sBLISS-seq data was performed with 8-10 replicates per timepoint. Single-nucleus RNA-seq experiments were performed on 2 independent Npas4fl/fl (Cre vs ΔCre) animals and 3 independent Tip60fl/fl (Cre vs ΔCre) animals. γH2AX ChIP-seq was performed in triplicate. ATAC-seq was performed in triplicate. All CUT&RUN experiments were conducted at least twice and for most antibodies in triplicate, with the exception of one EP400 dataset generated using an antibody from Bethyl Laboratories. This additional dataset corroborates data using a second EP400 antibody from Abcam, which has been conducted in duplicate in wild-type tissue and in triplicate in ΔCre (Control) infected animals. NPAS4 CUT&RUN was performed 5 times in wild-type mice and in duplicate in NPAS4 KO mice. IP-MS experiments on hippocampal tissue were performed in triplicate for the initial isolation of the complex. IP-MS experiments conducted from fractionated lysates were performed in duplicate. BLISS-seq experiments were conducted using 8-10 replicates in wild-type mice and three to five times in our Npas4 and Tip60 cKO (Cre vs ΔCre). |
| Randomization | All our data are derived from mice or cultured HEK293T cells. In general samples were grouped according to mouse age and genotype. Care was taken to include mice of both sexes equally in experimental groups. |
| Blinding | For genomic assays (ATAC-seq, RNA-seq, ChIP-seq, CUT&RUN, amplicon sequencing for mutational analysis, BLISS-seq, END-seq) and biochemistry, blinding of animal/tissue genotype was not feasible due to constraints in sample processing. All genomic assays are treated to the same bioinformatic pipelines. For electrophysiology experiments, the experimenter was blinded to animal genotype and treatment. |

# Reporting for specific materials, systems and methods

We require information from authors about some types of materials, experimental systems and methods used in many studies. Here, indicate whether each material, system or method listed is relevant to your study. If you are not sure if a list item applies to your research, read the appropriate section before selecting a response.

## Materials & experimental systems

| n/a | Involved in the study |
|---|---|
| ☐ | ☒ Antibodies |
| ☐ | ☒ Eukaryotic cell lines |
| ☒ | ☐ Palaeontology and archaeology |
| ☐ | ☒ Animals and other organisms |
| ☒ | ☐ Clinical data |
| ☒ | ☐ Dual use research of concern |

## Methods

| n/a | Involved in the study |
|---|---|
| ☐ | ☒ ChIP-seq |
| ☐ | ☒ Flow cytometry |
| ☒ | ☐ MRI-based neuroimaging |

## Antibodies

| | |
|---|---|
| Antibodies used | HA (Cell Signaling Technology, C29F4, RRID AB_1549585), HA (Sigma-Aldrich, ROAHAHA); rabbit anti-NPAS4 (in house (Lin et al., Nature 2008); rabbit anti-ARNT2 (in house, Sharma et al., Neuron 2019); rabbit anti-KAT5 (Proteintech, 10827-1-AP, RRID AB_2128431); rabbit anti-Cleaved Caspase-3 (Cell Signaling Technology, 9664S, RRID AB_2070042); rabbit anti-EP400 (Bethyl Labs, A300-541-A, RRID AB_2098208);  rabbit anti-EP400 (Abcam, Ab5201, RRID AB_304780), rabbit anti-EP400 (Abcam, 70301, RRID AB_1269644); rabbit anti-DMAP1 (Cell Signaling Technology, 13326, RRID AB_2798180); mouse anti-DMAP1 (Santa Cruz, sc373949, RRID AB_10918457); rabbit anti-TRRAP (Bethyl, A301-132A, RRID AB_2209668); rabbit anti-cFOS (in house); mouse anti-β-TUBULIN3 (Covance, MMS-435P, RRID AB_2313773), rabbit anti-GAPDH (Sigma-Aldrich, G9545, RRID AB_796208); rabbit anti-NPAS3 (Gift of Steven McKnight; Erbel-Sieler et al., PNAS 2004); rabbit anti-H3K27ac (Abcam, ab4729, RRID AB_2118291); rabbit anti-MRE11 (Novus Biologicals, Novus NB100-142, RRID AB_10077796, Lot R and S); rabbit anti-ETL4 (Bethyl Laboratories, A304-928A, RRID AB_2621122); anti-M2-FLAG 1224 resin (Sigma-Aldrich, A2220, RRID AB_10063035); Mouse anti-NeuN-Alexa488 (Millipore, MAB377X, clone A60, RRID AB_2149209); Mouse IgG-Alexa488 (Life Technologies, MA518167, RRID AB_2539541); rabbit anti-γH2AX (Abcam, ab2893, RRID AB_303388); rabbit anti-RAD50 (Novus Biologicals, NB100-154, RRID AB_2177080); rabbit anti-CTCF (Active Motif, 61311, RRID AB_2614975). For validation, see manufacturer's details for specificity using RRID from antibody registry. |
| Validation | Antibody validation was tailored to application. For Western blots, the following antibodies were used: rabbit anti-EP400 (Bethyl Labs, A300-541-A; Abcam, Ab5201; Abcam, 70301; 1:1000), rabbit anti-DMAP1 (Cell Signaling Technology, 13326; 1:500 or 1:1000) mouse anti-DMAP1 (Santa Cruz, sc373949; 1:500 or 1:1000), rabbit anti-TRRAP (Bethyl, A301-132A; 1:500 or 1:1000), rabbit anti-ARNT2 (in house; 1:1000), rabbit anti-NPAS4 (in house; 1:1000), rabbit anti-FOS (in house; 1:1000), mouse anti-β-TUBULIN3 (Covance, MMS-435P, 1:5000), rabbit anti-GAPDH (Sigma-Aldrich, G9545; 1:5000), rabbit anti-Histone H3 (Abcam 1791, 1: 10,000), rabbit anti-NPAS3 (Gift of Steven McKnight; 1:1000), rabbit anti-HA (Cell Signaling Technology, C29F4; 1:1000). Knockout or knockdown controls were performed to validate NPAS4, EP400, DMAP1, ARNT2, MRE11. HA validation was performed by blotting in wild-type mice lacking a transgenic epitope. Rabbit anti-NPAS4 was previously validated (Lin et al., Nature 2008). Rabbit anti-ARNT2 was previously validated (Sharma et al., Neuron, 2019). Rabbit anti-FOS in house antibody was previously validated (Yap et al., Nature 2021). For CUT&RUN, the following antibodies were used: rabbit anti-FOS (in house), rabbit anti-H3K27ac (Abcam, ab4729), rabbit anti-NPAS4 (in house), rabbit anti-EP400 (Bethyl Laboratories, A300541A; Abcam ab5201), rabbit anti-ARNT2 (in house), rabbit anti-MRE11 (Novus Biologicals, Novus NB100-142), rabbit anti-RAD50 (Novus Biologicals, NB100-154), rabbit anti-CTCF (Active Motif, 61311), rabbit anti-ETL4 (Bethyl Laboratories, A304-928A). For NPAS4 and MRE11, CUT&RUN signal was analyzed relative to knockout conditions. When KO conditions were difficult to obtain, multiple independent antibodies were used for a target protein (e.g. EP400).  For immunohistochemistry, the following antibodies were used:  rat anti-HA (Sigma-Aldrich, ROAHAHA, 1:250); rabbit anti-NPAS4 (in house, 1:1,000); rabbit anti-ARNT2 (in house, 1:1,000); rabbit anti-KAT5(TIP60) (Proteintech, 10827-1-AP, 1:250); rabbit anti-Cleaved Caspase-3 (Cell Signaling Technology, 9664S, 1:1,000). Antibody staining was validated in conditional knockout tissue or other negative controls (e.g. wild-type mice lacking a transgenic epitope).  For FACS with staining, the following antibodies were used: mouse anti-NeuN-Alexa488 (Millipore, MAB377X, clone A60); Mouse IgG-Alexa488 (Life Technologies, MA518167). A non-targeting isotype control antibody was used as a negative control. NeuN staining and sorting was validated by qPCR for neuronal vs non-neuronal markers from sorted populations. For additional validation, see manufacturer's details for specificity using RRID from antibody registry listed above. |

## Eukaryotic cell lines

Policy information about cell lines and Sex and Gender in Research

| | |
|---|---|
| Cell line source(s) | HEK293T (Thermo Fisher Scientific # 50188404FP) |
| Authentication | Not tested |
| Mycoplasma contamination | Not tested |
| Commonly misidentified lines (See ICLAC register) | No commonly misidentified cell lines were used in this study. |

# Animals and other research organisms

Policy information about <u>studies involving animals</u>; <u>ARRIVE guidelines</u> recommended for reporting animal research, and <u>Sex and Gender in Research</u>

| | |
|---|---|
| Laboratory animals | Animal use was approved and overseen by Harvard University Institutional Animal Care and Use Committee and Harvard Center for Comparative Medicine. The following mouse lines were used: wild-type C57/BL6 (Jackson Labs Stock 000664), Npas4fl/fl (Lin et al., 2008), Npas4 KO (Lin et al., Nature 2008), Tip60fl/fl (Fisher et al., Plos One 2016), Npas4-FLAG-HA (this manuscript), Arnt2-FLAG-HA (this manuscript), Tip60-H3F (Chen et al., eLife 2013), Mre11fl/fl (Buis et al., Cell 2008, Nat Struct Mol Biol 2012), B6;129-Gt(ROSA)26Sor<tm5(CAG-Sun1/sfGFP)Nat>/J (Jackson labs Stock 021039), B6.Cg-Tg(Camk2a-cre)T29-1Stl/J (Jackson labs Stock 005329). Mice were housed in a temperature and humidity-controlled environment using ventilated micro-isolator cages. Mice were kept under a standard 12 hr light/dark cycle, with food and water provided ad libitum. Male and female littermate mice were used in similar proportions and divided between control and experimental groups for all experiments conducted. In the case of NPAS4 KO lifespan analysis, data is also separated by sex. See Extended Data 13. For biochemistry and genomic experiments, animals were collected at 4-6 weeks of age throughout the manuscript. For physiology experiments, animals were dissected and patched at P24-P28 . For aging experiments, animals were collected at 3-4 months, 12 months, and 23-27 months of age. Details of animal age and sex are detailed within each protocol. |
| Wild animals | Did not involve wild animals |
| Reporting on sex | Findings apply to mice of both sexes as male and female littermate mice were used in similar proportions and divided between control and experimental groups for all experiments conducted. In the case of the Npas4KO mice lifespan studies, we report individual results by sex. |
| Field-collected samples | No field samples collected |
| Ethics oversight | Animal use was approved overseen by Harvard University Institutional Animal Care and Use Committee and Harvard Center for Comparative Medicine. |

Note that full information on the approval of the study protocol must also be provided in the manuscript.

# ChIP-seq

## Data deposition

☒ Confirm that both raw and final processed data have been deposited in a public database such as <u>GEO</u>.

☒ Confirm that you have deposited or provided access to graph files (e.g. BED files) for the called peaks.

| | |
|---|---|
| Data access links<br>*May remain private before publication.* | Sequencing data have been deposited in the Gene Expression Omnibus with accession number GSE175965. Additional data are provided in source data files. |
| Files in database submission | Sequencing data have been deposited in the Gene Expression Omnibus with accession number GSE175965. |
| Genome browser session<br>(e.g. <u>UCSC</u>) | Graph files (bigWig) are included in Gene Expression Omnibus GSE175965. |

## Methodology

| | |
|---|---|
| Replicates | Replicate information provided in Supplementary Table 2, Methods, and Figure Legends. Briefly, CUT&RUN experiments were conducted at least twice and for most antibodies in triplicate, with the exception of one EP400 dataset generated using an antibody from Bethyl Laboratories A300541A. This additional dataset corroborates data using a second EP400 antibody from Abcam, which has been conducted in duplicate in wild-type tissue and in triplicate in ΔCre (Control) infected animals. NPAS4 CUT&RUN was performed 5 times in wild-type mice and in duplicate in NPAS4 KO mice. γH2AX ChIP-seq was performed in triplicate. |
| Sequencing depth | Sequencing depth information provided in Supplementary Table 2. Briefly, all CUT&RUN experiments were sequenced on average with 22 million adapter trimmed pairs, with a minimum of 8 million adapter trimmed pairs. anti-γH2AX ChIP-seq samples were sequenced to a minimum depth of 20 million reads. |
| Antibodies | ChIP-seq: rabbit anti-γH2AX (Abcam, ab2893); rabbit anti-NPAS4 (in house); CUT&RUN: rabbit anti-NPAS4 (in house), rabbit anti-FOS (in house), rabbit anti-H3K27ac (Abcam, ab4729), rabbit anti-EP400 (Bethyl Laboratories, A300541A; Abcam ab5201), rabbit anti-ARNT2 (in house), rabbit anti-MRE11 (Novus Biologicals, NB100-142), rabbit anti-RAD50 (Novus Biologicals, NB100-154), rabbit anti-CTCF (Active Motif, 61311), rabbit anti-ETL4 (Bethyl Laboratories, A304-928A). |
| Peak calling parameters | Peak calling on ChIP samples was performed using MACS2 (macs2/2.1.1) using the following command macs2 callpeak –t (experimental bam) –c (input bam) -f BAM -g mm -p 1e-5. Required p less than 1e-5 in peak calling for ChIP-seq. All peak calling for CUT&RUN was performed using SEACR_1.1.sh (Meers et al., 2019). For H3K27ac, FOS, CTCF, and RAD50 CUT&RUN datasets, peak calling on individual replicates was performed using the spikein normalized bedgraph files based on fragments 1 to 1000 bp in length with the following command: SEACR_1.1.sh [target bedgraph] [control bedgraph] norm stringent. For NPAS4, ARNT2, EP400, MRE11, and ETL4 CUT&RUN datasets, peak calling on individual replicates was performed using SEACR_1.1.sh using the spikein normalized bedgraph files based on fragments 1 to 1000 bp in length with the following command: SEACR_1.1.sh [target bedgraph] [control bedgraph] norm relaxed. Paired control samples (either IgG or KO control) are listed in Supplementary Table 2. |

| Data quality | To identify reproducible peak sets for CUT&RUN, SEACR peaks found in 3 of 3 H3K27ac replicates (0 and 2hr KA stimulation), 3 of 3 FOS replicates (0 and 2hr KA stimulation), 3 of 3 CTCF replicates (0 and 2hr KA stimulation), 3 of 3 RAD50 replicates (0hr) and 3 of 4 RAD50 replicates (2hr KA stimulation), 4 of 5 NPAS4 replicates (2hr KA stimulation), 3 of 3 NPAS4 replicates (0hr KA stimulation), 2 of 2 ARNT2 replicates (2hr KA stimulation), 2 of 2 EP400 replicates (0 and 2hr KA stimulation; Abcam antibody), 2 of 2 MRE11 replicates (0 and 2hr KA stimulation), 2 of 2 ETL4 replicates (0hr KA stimulation) and 3 of 4 ETL4 replicates (2hr KA stimulation) were intersected using bedtools/2.27.1 intersect bed. Peaks within 150 bp were merged. Finally, the maxima of CUT&RUN signal within 100 bp windows for each peak was calculated from spikein normalized bigWig files using custom scripts. For FOS, ARNT2, EP400, ETL4, and MRE11, final peak calls were extended 200 base pairs up and downstream from this peak maxima to generate 500 base pair peak calls for each factor and timepoint. Mm10 blacklisted regions were filtered out using the following command: bedops/2.4.30 –not-element-of 1 [BLACKLIST_BED]. NPAS4 peaks were extended to 1 kb, as we found maximal enrichment for the Ebox (CAGATG) motif and bHLH/PAS motif (CGTG) in 1 kb regions extended from peak maxima. |
|---|---|
| Software | Peak calling on ChIP samples was performed using MACS2 (macs2/2.1.1). ChIP-seq samples were aligned to the mm10 genome using the Bowtie alignment software(vbowtie2/2.2.9) with the –very-sensitive setting. CUT&RUN peak calling was performed with SEACR_1.1.sh. |

# Flow Cytometry

## Plots

Confirm that:

☒ The axis labels state the marker and fluorochrome used (e.g. CD4-FITC).

☒ The axis scales are clearly visible. Include numbers along axes only for bottom left plot of group (a 'group' is an analysis of identical markers).

☒ All plots are contour plots with outliers or pseudocolor plots.

☒ A numerical value for number of cells or percentage (with statistics) is provided.

## Methodology

| Sample preparation | For GFP and mCherry FACS: Dissected hippocampal tissue was examined under a florescent scope to detect GFP or mCherry. Tissue that was uninfected, or in rare cases showed infection of both fluorophores in a single hemisphere, was discarded. Hippocampi were placed in 0.5 mL of buffer HB (0.25 M sucrose, 25 mM KCl, 5 mM MgCl2, 20 mM Tricine-KOH, pH 7.8, 1 mM DTT, 0.15 mM spermine, 0.5 mM spermidine) and dounced 5X with a loose pestle and 10X with a tight pestle. 5% IGEPAL CA-630 (32 μL) was added prior to douncing with a tight pestle 5-8 more times and filtering through a 40-μm strainer. DRAQ5 nuclear dye (Abcam; ab108410) was added (1:500), and nuclei expressing either mCherry or GFP were sorted on a SONY SH800. Negative gates were determined using uninfected tissue. Nuclei were collected in 1 mL of CUT&RUN Wash Buffer containing 2 mM EDTA.<br>For NeuN-FACS: To sort neuronal nuclei for DNA isolation, dissected hippocampal tissue was placed in 1 mL of buffer HB (0.25 M sucrose, 25 mM KCl, 5 mM MgCl2, 20 mM Tricine-KOH, pH 7.8, 1 mM DTT, 0.15 mM spermine, 0.5 mM spermidine) and dounced 5X with a loose pestle and 10X with a tight pestle. 5% IGEPAL CA-630 (32 μL) was added prior to douncing again with a tight pestle 5-8 times. The nuclei suspension in HB was filtered through a 40-μm strainer. Nuclei were then pelleted by centrifugation at 500 g for 5 min and resuspended in 400 μL of FACS Block/Stain Buffer (1% BSA, 0.05% Igepal-630, 3 mM MgCl2 in 1X PBS). Nuclei were incubated for 15 min with gentle rotation at 4°C. Following this blocking step, nuclei were pelleted and resuspended in FACS Block/Stain Buffer containing 1:1000 dilution of mouse anti-NeuN-Alexa488 (Millipore, MAB377X). An isotype control Mouse IgG-Alexa488 (Life Technologies, MA518167) was included as a negative control along with an unstained sample. Samples were incubated in antibody mix for 1 hour with gentle rotation at 4°C. Nuclei were washed 1X with FACS Block/Stain Buffer, and DRAQ5 nuclear dye (Invitrogen) was added (1:500) prior to sorting. NeuN-high-expressing nuclei were separated from NeuN-low-expressing nuclei using a SONY SH800. |
|---|---|
| Instrument | SONY SH800 |
| Software | Sony SH800Z Cell Sorter software was used during acquisition of data. Data were subsequently analyzed using FlowJo (10.0.8r1) |
| Cell population abundance | Singlet DRAQ5-positive nuclei represented roughly 25% of the initial population. NeuN+ nuclei gated from the singlet DRAQ5-positive population represented roughly 60% of the population. Cre-mCherry+ and ΔCre-GFP+ nuclei abundance depended on the viral injection and tissue microdissection and ranged from 20% to 75% of the singlet DRAQ5-positive population. See Extended Data Figs. 11d,e and 13a and Supplementary Fig. 2 for gating strategy. |

Gating strategy

For NeuN-FACS: Unstained sample without DRAQ5 nuclear dye was used to establish the APC-Cy7-positive gate for DRAQ5-positive nuclei. Nuclei stained with DRAQ5 were initially selected based on APC-Cy7 signal, followed by selection of nuclei with linearly proportional FSC area and height signal to isolate singlet nuclei. NeuN+ gate was determined using both a DRAQ5-stained sample that was not stained with Mouse anti-NeuN-Alexa488 (no primary control) and a DRAQ5-stained sample that was stained with a Mouse IgG-Alexa488 isotype control. See Extended Data 13a and Supplementary Fig. 2 for gating strategy.

For mCherry and GFP FACS: Unstained sample without DRAQ5 nuclear dye was used to establish the APC-Cy7-positive gate for DRAQ5-positive nuclei. Nuclei stained with DRAQ5 were initially selected based on APC-Cy7 signal, followed by selection of nuclei with proportional APC-Cy7 area and SSC signal to isolate singlet nuclei. mCherry+ and GFP+ gates were determined using a DRAQ5-positive sample from an uninfected mouse. See Extended Data 11d,e and Supplementary Fig. 2 for gating strategy.

☒ Tick this box to confirm that a figure exemplifying the gating strategy is provided in the Supplementary Information.

