## [Peer Review File · Nature]

Manuscript Title: An NPAS4:NuA4 complex couples synaptic activity to DNA repair

Reviewer Comments & Author Rebuttals

Reviewer Reports on the Initial Version:

Referees' comments:

Referee #1:

The authors discover an association of NPAS4 with a large protein NuA4 complex previously implicated to DNA repair. Through elegant genetic and molecular experiments they confirm a functional association and recruitment of NuA4 components to neuronal NPAS4-bound enhancers and promoters. They then test

This manuscript contains a wealth of data, combining an array of quantitative approaches. Overall, the work is very well done, timely and exciting. While dsDNA breaks have been proposed to be linked to activity-dependent transcription in neurons in previous work, this study far extends the original findings and adds multiple new critical layers. Thus, the work reports major discoveries regarding the mechanism, coupling to NPAS4, and life-long impact of impairment of this DNA repair system.

There are a few specific points that should be addressed in a revision:

1) One surprising finding in this work is the activity-dependent recruitment of DNA repair machinery by a neuron-specific transcription factor. However, the selectivity of this phenomenon raises some questions:

a) Data by the authors indicates that after kainic acid stimulated brain lysates and NuA4/TIP60 and NPAS4 form a high molecular weight complex. Are NuA4/TIP60 components elevated in response to depolarization? Or do NPAS4/ARNT2 heterodimers join a pre-existing complex upon activity-dependent production of NPAS4?

b) The relationship between NuA4 complex and NPAS4 is somewhat unclear. While NPAS4 is activity-dependent and acutely recruited to enhancers and promoters, NuA4 components likely bind more broadly to genomic regions – independent of neuronal activity. The loss of EP400 from NPAS4-recruiting enhancers in *Npas4* KO mice is modest. The authors suggest that this may reflect perdurance of EP400 at previously NPAS4-bound sites – but other interpretations are possible. The authors should probe selectivity for NPAS4 by correlating cFOS and EP400 binding sites (using existing ChIP or cut-and-run datasets). This should reveal whether there is specificity for NPAS4 or rather general recruitment to genes regulated by IEG transcription factors.

2) TIP60 cKOs and NPAS4 cKOs share certain neuronal phenotypes. The authors interpret this to show that TIP60 is required for NPAS4 function. Another possibility is that TIP60 is required for NPAS4 expression or stabilization. This could be easily tested/excluded by comparing NPAS4 levels in tissue from wild-type and TIP60 KO mice without and with kainite stimulation.

3) The most innovative part of this paper links dsDNA breaks and mutation of *Npas4*. One caveat for the interpretation of these experiments is the synaptic phenotype of *Npas4* itself. Based on the

data from the authors, loss of NPAs4 will impair the ability of neurons to counter strong stimulation by forming additional GABAergic synapses. Thus, in mice with a life-long loss of NPAS4 one would expect that neurons are more frequently exposed to high levels of excitation that can not be scaled down through increasing GABAergic synapse formation. A potential genetic experiment to address this issue would be the knock-in of a mutation that selectively impairs coupling of NPAS4 to the DNA repair functions but which maintains the ability to activate transcription of NPAS4 target genes that promote GABAergic synapse formation. This might be a new project by itself – but in case the authors have data to address the caveat noted above this would further strengthen the paper. If not, this needs to be discussed in the manuscript.

Minor points:

Line 812: “not bound by NPAS4 versus those not bound by NPAS4”?

Ext. Data Fig. 7e: What do the colors/columns represent? I assume its the different NPAS4 deletion constructs? Please explain in figure or legend.

Referee #2:

The manuscript by Pollina et al. presents the discovery of a novel role for the neuronal activity-regulated gene, NPAS4, in association with the DNA repair complex NuA4 in maintaining genome stability and integrity in neurons. The authors used a wide range of methods, including extensive genomic assays in wild-type and knockout neurons as well as electrophysiology, to establish that NPAS4 protein associates with multiple components of the NuA4 complex in neurons. They show that these components are co-localized with NPAS4 at activity-induced "peaks" -- i.e. cis-regulatory elements that can be identified by chromatin accessibility (ATAC), histone modifications (H3K27ac), as well as binding of NPAS4 and NuA4 components (CUT&RUN). Using sBLISS-Seq, they link these activity-induced genomic regions with DNA double-strand breaks (DSBs), which have recently been proposed to affect the genome at neuronal activity-induced genes. Because DSBs may represent a threat to the integrity of the genome in post-mitotic, long-lived neurons, Pollina and co-authors hypothesized that the NPAS4:NuA4 complex may play a role in supporting DNA repair at these sites. They show that a DNA repair protein, MRE11, co-localizes with NPAS4 at activity-induced peaks, and is reduced following KO of NPAS4. Finally, the authors present evidence that somatic mutations accumulate at regions affected by DSBs throughout the mouse lifespan, increasing from young (3 month) to old (24 month) mice at sites that lack NPAS4-mediated genome protection. They argue that the genome-protective role of NPAS4 may explain the shortened lifespan of NPAS4 KO mice.

Overall this is a very impressive, novel and interesting study. It presents a compelling case for what is a provocative and timely new hypothesis regarding the role of NPAS4 and of DSBs in the life of neurons. The wide range and high quality of genomic assays, computational analyses, and functional validation make this a strong paper that will be of wide interest to neuroscientists and genomicists alike.

My main critiques concern the clarity of presentation of some of the findings and methods. Because so many different, complex techniques are used, it can be difficult to comprehend each of the findings. In particular, some of the analyses are described in the text and figures using simplified short-hand, so that the reader must hunt through the Methods section to understand what is actually being shown. To take one example, it would be helpful when showing "activity-inducible genes" (in Fig. 3c) that are "bound or not bound by NPAS4" to include at least a bit more detail about how genes were assigned to these categories.

A second general comment is that there is not enough clarity about the biological replicates that were used. The figures generally show a single value or track for each experimental group, and I could not determine how many independent biological replicates were used for the various analyses. In cases where multiple replicates can be shown without adding too much complexity/crowding to the figure, that would greatly help the reader to appreciate the reproducibility and reliability of the data.

Specific comments:

Fig. 1b: Does the top "Home" image show a merge of all fluorescence channels? This was unclear.

Fig. 1f: It would be helpful to (also) show these data in an unnormalized format, especially because this figure is used to support that the NuA4 expression is limited to neurons.

Fig. 1h: The color scale shows "read depth" for each CUT&RUN experiment -- but it is not clear if the read depth has been normalized. Specify if this showing RPKM (i.e. normalized for library size).

Fig. 1i: Need statistical analysis, based on replicates, for the difference between 0 and 2 hr.

Fig. 2b: Single-cell transcriptome analysis was a little hard to follow here. What is driving the large separation between KO (Cre) and control neurons of the same type (e.g. CA1 pyramidal cells)? Showing the UMAP on its own is not sufficient, especially since it looks like the plot points are overlapping and obscuring each other such that it's hard to tell where each of the populations is present. Moreover, the two shades of gray (DeltaCRE and "uninfected") are very hard to distinguish.

-- It was not very easy to understand how the set of "NPAS4 target genes" and "TIP60 target genes" was defined.

-- In Ext. Data Fig. 6, explain the units for gene expression ($\log_{10}(\text{TPM})$?).

Fig. 2c: It is not really very useful to show the distribution of "expression" level (presumably in something like $\log_{10}(\text{TPM})$ units, but this is not actually labeled) across a bunch of different genes. Instead I would suggest showing the distribution of the $\log(\text{fold-change})$ across genes. This should not affect the conclusions.

Fig. 2f: How many different animals (biological replicates) were used here? Does each pair of cells come from an independent experiment?

Lines 188/189: Explain more clearly the test that was used to determine the 1766 differentially-expressed genes. Some information about thresholds, p-values, FDR correction, effect size, etc., should be provided in the main text, not only in the methods.

Line 202: I don't think the data show that NPAS4 and NuA4 are obligate partners. What is the evidence that they don't also have additional independent roles?

Fig. 3e - What is the statistical analysis here? How many animals (replicates) are used? What is the statistical basis of this statement: "While non-inducible sites did not increase 278 the number of DNA breaks with neuronal activity, inducible regulatory elements 279 displayed an increased number of breaks at 2 hours post-stimulation and a return to 280 baseline at 10 hours post-stimulation, suggesting that these sites are breaking and 281 subsequently undergoing repair (Fig. 3e)."?"

Ext. Data Fig. 10f: It is hard to see the colors. What are the criteria for defining MRE11 or NPAS4 peaks?

Fig. 3i,j: MRE11 binding is reduced but not completely abolished by lack of NPAS4 or TIP60. Does this suggest there are additional, NuA4-independent mechanisms for recruiting DNA repair?

Fig. 3i: The Delta-CRE mice should not be labeled as "WT". Instead they could be labeled "control".

Fig. 4: The evidence for mutation accumulation in aging was intriguing but relatively thin compared with the rest of the paper. Mutations were analyzed in NPAS4 cKO mice at only one time point, making it hard to directly assess aging.

Fig. 4d,e: What is the absolute (not normalized) rate of mutations?

Fig. 4e: "Each point represents one site sampled from one mouse." How many mice are shown here? The analysis should take into account this nested data structure, rather than treating all the data points as independent.

Line 214 missing a word: "we ranked all regulatory in our dataset".

Signed: Eran A. Mukamel

Referee #3:

General assessment:

Overall, this manuscript contains extremely interesting data that represent an enormous amount of excellent work, involving an impressive arsenal of complex technologies. The results on the NPAS4 interaction with the NuA4/TIP60 chromatin modifier complex in neurons are consistent and in all regards convincing. The DNA repair connection, on the other hand, raises a number of questions of experimental and conceptual nature.

Major comments:

I should start by complimenting the authors for the high scientific quality of their study, including the use of proper controls in all of the experiments. Striking examples are:

1. The approach to generate *Npas4*-FLAG-HA and *Arnt2*-FLAG-HA knock-in transgenic mouse lines using Crispr-Cas9 technology, in which FLAG and HA epitope tags are fused in frame to the 3' end of the endogenous gene for the isolation of NPAS4 protein complex directly from the brain is very powerful as it ensures physiological expression levels and normal regulation. The subsequent interaction between NPAS4 and all known subunits of the NuA4/TIP60 chromatin modifier complex and one new subunit is absolutely stunning (Fig. 1 and Ext. Data Fig. 1).

2. Another elegant step is the nested PCR strategy employed in the library preparation to amplify viral transcripts in order to identify individual infected cells and discriminate them from non-infected cells, which strongly improves the power of the analysis (Fig. 2a,b, Ext. Data Fig. 5e).

3. A third nice example of scientific rigor is the generation of a mutant form of NPAS4 that cannot

interact with NuA4 (Ext. Data Fig. 1f), used in luciferase reporter assays (Ext. Data Fig. 7d,e). This is indeed a very powerful experimental approach.

Also, the stereotactic injections of AAV transgenic CRE-marker constructs and the subsequent analyses, including single-nucleus RNA-sequencing and current recordings of neural circuits as in Fig. 2 are truly impressive.

Altogether the paper presents very solid and conclusive evidence that NPAS4 and NuA4 can function together as a discrete protein complex in neurons upon stimulation. For this part of the manuscript there are no comments, only compliments.

The DNA repair connection

As reported before (and cited in the paper) the NuA4 chromatin modifier complex on its own exerts two functions: control of gene expression and helping in coordinating DSB repair, although this latter activity appears to this referee still rather ill-defined in current literature. Obviously, the DNA repair machinery has to operate in the context of chromatin and repair transactions involve all known chromatin modifications. Therefore, it is a priori not very surprising that for virtually all DNA repair systems, including DSB repair pathways, many connections with chromatin-modifying protein complexes have emerged in the past two decades. However, these are always transient and hence dynamic in nature, which is intrinsic to the requirements of repair of damage that is stochastic, random and can occur at any time at any location in the genome. I presume that this is the same in the case of the NPAS4/NuA4 complex and the repair proteins investigated here and that some of the specific questions raised below relate to this phenomenon. Presumably, in this light the link between genuine DNA damage induction, intrinsically dynamic DNA repair proteins and the Npas4/Nu4/TIP60 chromatin modifier complex has to be considered.

Also, it has been amply demonstrated (as cited in the manuscript) that transcription is linked to induction of DSBs, which are usually transient as well because persistent DSBs are very potent activators of cell cycle checkpoints and the DNA damage response, and e.g., lead to SCARS, which are associated with cellular senescence. Presumably, part of these 'DSBs' might simply be intermediates of normal DNA metabolism that are captured in the middle of a DNA transaction at the moment of cell lysis, which usually involves detergents. Examples could be topoisomerase-DNA intermediates involved in transcription-related unwinding reactions. They would have been sealed moments later. A small fraction of the transcription-related DSBs are undoubtedly toxic. However, this raises the question how to discriminate the artificial breaks of DNA metabolism intermediates from the long-lived, toxic DSBs and how to quantify them in a reliable manner?

Specific comments:

1. To assess the effect of NPAS4 and TIP60 on the formation and repair of DNA breaks at NPAS4- or TIP60-bound sites and across the neuronal genome, Cre-expressing or (inactive) Δ Cre-expressing AAVs were injected in the brain of Npas4^{fl/fl} or Tip60^{fl/fl} mice and nuclei isolated at 0, 2, or 10 hr post-stimulation for sBLISS-seq. The use of a recombination-deficient version of Cre (Δ Cre-GFP) is an excellent control. However, one potential complication that might interfere with some of the assays in this study is that CRE will induce DSBs by itself, not only in flox-sequences but also off-target, at other sites in the genome as amply demonstrated (e.g., see Loonstra A, et al.2001. Growth inhibition and DNA damage induced by Cre recombinase in mammalian cells. Proc. Natl Acad. Sci. USA 98:9209-14. doi: 10.1073/pnas.161269798, PMID: 11481484). In wt cells the effects may be limited, as these lesions generally can be quite efficiently repaired but in case of DSB-repair mutants, as in this study, this may exert significant negative effects including activation of DNA damage signaling and increased DSBs. This may explain at least in part the generally higher number

of DSBs in Fig. 3f and in Ext. Data Fig. 9h-j after KA induction and even before KA induction in the mutant which have active Cre, versus wt, which have the inactive Δ Cre. It would also explain the elevated number of genome-wide DSBs at all time points with and without KA activation in floxed mutant brain only with CRE expression and in Npas4 mutant conditions, but not in wt, as displayed in Fig. 4f. and for the Tip60-conditional inactivation by Cre presented in Ext. Data Fig. 14a. This is relevant for the discussion on p. 13 (lines 291 and further), as well as the discussion on p. 18 (lines 401 etc.), which is speculative and to me not very plausible. On the other hand, this would argue that Npas4 and Tip60 mutations affect (Cre-induced) DSB repair. Are mutant cells sensitive to DSB-inducing agents such as ionizing radiation, bleomycin or topoisomerase inhibitors?

2. MRE11 is a suitable marker for DSBs in vivo, and the co-localization and presumed interaction of MRE11 and Npas4/NuA4 complex reported here is sound, but it raises a number of questions. The nature of this interaction is still unclear and could well be indirect, transient and non-selective. Affinity of MRE11 with chromatin modifiers and accessible genomic regions and with ongoing DNA transactions would explain “that MRE11 is often present at NPAS4-bound elements even in unstimulated neurons” (lines 318-319), but also at other sites in the genome as observed in this study. Slight differences in affinity to specific binding sites (e.g., chromatin modifications) could explain why e.g., MRE11 is found more at NPAS4 sites than at FOS sites in the genome as found here. Photobleaching experiments of fluorescently-tagged MRE11 and other partners would be required to determine whether the interactions are dynamic, based on short-lived affinities.

To get better insight it would be informative to determine the presence of other components of the MRN complex: NBS1 and/or Rad50. Even more critical: DSB are very genotoxic, can frequently cause chromosomal aberrations, but are generally not particularly mutagenic (i.e., causing base substitutions), because of their low abundance, compared to much more abundant oxidative lesions and abasic sites due to spontaneous hydrolysis. MRE11 is important for DSB repair. This raises the question what about the link with other DNA repair systems such as base excision, nucleotide excision and transcription-coupled repair, which are important for preserving long-term neurofunction? Were any of these proteins or repair pathways enriched in the Mass spec analysis for NPAS4 interactors in Fig.1d that identified all components of the NuA4/TIP60 chromatin modifier complex?

3. Mutation analysis (Fig. 4): Questions that emerge: how do absolute (not relative) mutation frequencies of the different classes of mutations (base substitutions, small deletions/insertions, large deletions, translocations, etc.) compare with mutation frequencies known in aging for brain and other organs/tissues? Are they in the same order of magnitude? What type of mutation signatures are observed? The Npas4-KO mice show a quite unusual survival curve in Fig. 4h, with equal mortality rates over their entire lifespan. What is the explanation and what is the main cause of death? How is neural functioning affected in physiology and behavior?

4. Conceptually, the authors argue that the NuA4/TIP60 chromatin modifier complex in neurons may function as a guardian of genome integrity through activity-dependent DNA repair at promoters and enhancers. However, this would only apply to MRE11-dependent DSB repair processes. Moreover, this concept would account only for part of the genome (determined to be 1766 activity-regulated genes) as other promoters and enhancers of e.g., constitutively expressed genes, which generally are important for cell viability, would not enjoy the same protection and hence might limit neuronal lifespan.

Minor comments

1. Ext. Data Fig. 1d: The MW marker for the TUJ1 loading control is indicated to be 15 kDa, whereas

it is 50 kDa.

2. It is unclear why the reader is referred to Ext. Data Fig. 1f in the context of the NPAS4 mutant which has lost the interaction with NuA4 (line 197), as this panel only shows the wt NPAS4. Moreover, it is clearer to readers to refer to this mutant in the main text as “a truncation mutant of NPAS4”, instead of “a mutant form of NAPS4”, to discern it from a point mutant (which probably would have been as a better separation-of-function mutant).

3. Fig. 3e: Colors to distinguish the line for promoters and for introns could be made more distinct.

4. Fig. 4a: The authors “aged cohorts of *Npas4* knockout (KO) mice and their wild-type littermates and collected hippocampal tissue from young (3 month), wild-type mice, middle-aged (12-17 month) wild-type and *Npas4* KO mice, and old (23-27 month) wild-type mice raised under standard housing conditions” (line 344). Why was only one age (12 months) used for the *Npas4* KO mouse cohort? To distinguish developmental from progressive degenerative symptoms associated with aging it would have been informative to have at least one interim cohort at young age.

Overall, I am impressed by the large amount data and high quality of the work. The part on the NPAS4 link with NuA4 chromatin modifier complex is sound. The repair connection raises a number of questions and will need significant strengthening.

Referee #4:

In the manuscript from Pollina et al., the authors generate a mouse model to investigate the NPAS4 transcription factor in neurons. They find that NPAS4 interacts with the NuA4 chromatin modifying complex and that this complex mediates transcription of an overlapping set of target genes and functions in the same subset of cell types. In addition, they propose that NPAS4 works together with NuA4 to mediate rapid repair of DNA double-strand breaks (DSBs) in the vicinity of NPAS4 binding, and that this confers protection to genome integrity. In Fig. 4, the authors show differences in mutation numbers between regions with NPAS4 binding compared with no binding, and between NPAS4 KO and WT, suggesting that NPAS4 plays a protective function in these genome locations during aging.

The data on the relationship between NPAS4 and NuA4, showing a physical interaction and evidence of common transcriptional activities is robust, and the mouse work is impressive. The major issue with this manuscript is that the data presented (primarily in Fig. 3 and supporting supplementary data) do not support the conclusions around double strand breaks and their repair. Specifically:

1. Without further validation or clarification, it's not clear whether the sBLISS data can be reliably compared across samples in a quantitative manner. In the paper where this technique was developed, this limitation is highlighted (Bouwman et al., Nat. Protoc., 2020), and the data here show considerable noise and apparent inconsistencies (for example, why are there so many more DSBs at non-induced sites in Fig. e?). Moreover, the recent paper from Nussenzweig and colleagues (Nature, 2021) mapped sites of repair in post-mitotic neurons and found no evidence of DSBs (albeit in a very different system) using a similar end mapping approach, raising the possibility that many of the sites mapped here are not physiologically relevant. At a minimum, all sBLISS data should be presented (all time points, all samples) with information about the number of unique reads in each sample and which of these intersect with locations where gH2AX was enriched and/or regions of transcriptional activity. It would also be important to see how these sites compare with

the patterns of mapped repair sites in Reid et al. (Science, 2021) and the Nussenzweig Nature paper.

2. Related to this, the authors use multiple different approaches to categorize and compare genomic sites that make it incredibly difficult to assess the data, and not all results (samples or time points) are shown in each analysis. For example, in Fig. 3c, it appears that genes that are actively transcribed in a NPAS-dependent manner are plotted, but in the rest of the figure, it is a different set of sites (defined by ATAC-seq and H327Ac data). And in Fig. 3f, the data from the KO from non-induced sites should be shown in the same way. If the model is that NuA4 functions to mediate repair at NPAS4-dependent transcriptionally-active genes, then that should be how the data is divided for comparison in all analyses. The differences in gH2AX ChIP levels, DSB (BLISS) numbers, and MRE11 binding should then be shown according to the same classification at the same sites for the WT, Npas4fl/fl KO, and TIP60fl/fl KO for all three time points.

3. The NuA4/TIPO complex has been implicated in DNA repair in human cells in a number of studies, and this work is not cited here. In particular, Tang et al. (NSMB, 2013) and Jacquet et al. (Mol. Cell, 2016) showed that this complex is important for promoting the use of homologous recombination (HR) in the S and G2 phases of the cell cycle to repair DNA DSBs and prevents 53BP1 binding. This raises the question of what repair mechanism could be in use in these post-mitotic cells. There is data around use of RNA-templated recombination (see Meers, et al., DNA Repair, 2016), which could be relevant here. This should be discussed, and the relevant papers cited.

The mutation data in Fig. 4, while showing relatively modest effects, is compelling. Again, interesting to see how this compares with the repair sites mapped in the Science/Nature papers, which mention G4 structures at NPAS4 sites. However, it doesn't directly implicate failed DSB repair at these sites and could reflect altered SSB repair (or other pathways) instead.

Author Rebuttals to Initial Comments:

REFEREE #1

The authors discover an association of NPAS4 with a large protein NuA4 complex previously implicated to DNA repair. Through elegant genetic and molecular experiments they confirm a functional association and recruitment of NuA4 components to neuronal NPAS4-bound enhancers and promoters.

This manuscript contains a wealth of data, combining an array of quantitative approaches. Overall, the work is very well done, timely and exciting. While double-stranded DNA breaks have been proposed to be linked to activity-dependent transcription in neurons in previous work, this study far extends the original findings and adds multiple new critical layers. Thus, the work reports major discoveries regarding the mechanism, coupling to NPAS4, and life-long impact of impairment of this DNA repair system.

We thank the reviewer for their thoughtful review of our work and are encouraged that they find the paper to be timely and exciting.

There are a few specific points that should be addressed in a revision:

1) One surprising finding in this work is the activity-dependent recruitment of DNA repair machinery by a neuron-specific transcription factor. However, the selectivity of this phenomenon raises some questions:

a) Data by the authors indicates that after kainic acid stimulated brain lysates and NuA4/TIP60 and NPAS4 form a high molecular weight complex. Are NuA4/TIP60 components elevated in response to depolarization? Or do NPAS4/ARNT2 heterodimers join a pre-existing complex upon activity-dependent production of NPAS4?

*The reviewers raise an important question. Our biochemical data so far suggests that NPAS4:ARNT2 dimers join and recruit a pre-existing complex. We show that Tip60 interacts with several components of the complex (TRRAP, DMAP1, EP400) before stimulation and that the NPAS4 association with NuA4 is dependent on stimulation (**Extended Data Fig. 2b**). We also show this complex may contain neuronal-enriched subunits, for example the new subunit ETL4, which seems to be recruited to the genome in an activity-dependent manner (**Fig. 1 h-j**).*

*We now also provide evidence that NuA4 components are not induced at the RNA level using a new timecourse of RNA-seq data from hippocampus (**Extended Data Fig. 2g**).*

b) The relationship between NuA4 complex and NPAS4 is somewhat unclear. While NPAS4 is activity-dependent and acutely recruited to enhancers and promoters, NuA4 components likely bind more broadly to genomic regions – independent of neuronal activity. The loss of EP400 from NPAS4-recruiting enhancers in NPAS4 KO mice is modest. The authors suggest that this may reflect perdurance of EP400 at previously NPAS4-bound sites – but other interpretations are possible. The authors should probe

selectivity for NPAS4 by correlating cFOS and EP400 binding sites (using existing ChIP or cut and run datasets). This should reveal whether there is specificity for NPAS4 or rather general recruitment to genes regulated by IEG transcription factors.

The reviewer raises an important point. Our biochemical data strongly suggests that FOS and one other IEG, EGR1, do not directly interact with NuA4 either in a heterologous system nor in the brain (Extended Data Fig. 2b,d). However, we now analyze our CUT&RUN data for FOS and NPAS4 from the hippocampus and show that sites bound by FOS but not NPAS4 have lower levels of EP400 than sites with the converse (Extended Data Fig. 3h,i).

We now also acknowledge in the text that the NuA4 complex contains readers of histone acetylation so it is possible that EP400, a proxy for the complex, is targeting these acetylated regions across the genome independent of NPAS4. We now clarify on lines 158-165 in the text that EP400 may also be recruited to the genome by alternative mechanisms.

2) TIP60 cKOs and NPAS4 cKOs share certain neuronal phenotypes. The authors interpret this to show that TIP60 is required for NPAS4 function. Another possibility is that TIP60 is required for NPAS4 expression or stabilization. This could be easily tested/excluded by comparing NPAS4 levels in tissue from wild-type and TIP60 KO mice without and with kainite stimulation.

We now show that at the protein level, NPAS4 induction is not significantly altered in Tip60 cKO unstimulated neurons or 2 hours post-stimulation (Extended Data Fig. 4d).

3) The most innovative part of this paper links double-stranded DNA breaks and mutation of NPAS4. One caveat for the interpretation of these experiments is the synaptic phenotype of NPAS4 itself. Based on the data from the authors, loss of NPAS4 will impair the ability of neurons to counter strong stimulation by forming additional GABAergic synapses. Thus, in mice with a life-long loss of NPAS4 one would expect that neurons are more frequently exposed to high levels of excitation that cannot be scaled down through increasing GABAergic synapse formation. A potential genetic experiment to address this issue would be the knock-in of a mutation that selectively impairs coupling of NPAS4 to the DNA repair functions but which maintains the ability to activate transcription of NPAS4 target genes that promote GABAergic synapse formation. This might be a new project by itself – but in case the authors have data to address the caveat noted above this would further strengthen the paper. If not, this needs to be discussed in the manuscript.

Thank you for this excellent experimental suggestion. We initially conducted truncation experiments to define a region of NPAS4 that cannot interact with the NuA4 complex, with the intention of identifying a point mutant (Extended Data Fig. 2c and Fig. 7d,e) that might disrupt the repair while leaving transcription intact. However, we have reason to suspect that NPAS4 interacts with multiple surfaces of the NuA4 complex. We are hopeful that manipulating select subunits of this complex may give more insight into

repair vs transcription. However, given that there are 20 subunits of this complex, we believe undertaking this experiment lies beyond the scope of the current manuscript but is an area of active investigation in the future. We now add this caveat to our discussion of the repair and lifespan data taking into account how lifelong loss of inhibitory control might affect these phenotypes (see text lines 651-658).

Minor points:

Line 812: “not bound by NPAS4 versus those not bound by NPAS4” ?

This sentence has now been corrected.

Extended Figure 7e: what do the colors/columns represent? I assume its the different NPAS4 deletion constructs? Please explain in figure or legend.

We apologize for the lack of clarity and we now specify in the relevant figure legend (Extended Data Fig. 7e) that the colors correspond to specific deletion constructs of NPAS4.

REFEREE #2

The manuscript by Pollina et al. presents the discovery of a novel role for the neuronal activity-regulated gene, NPAS4, in association with the DNA repair complex NuA4 in maintaining genome stability and integrity in neurons. The authors used a wide range of methods, including extensive genomic assays in wild-type and knockout neurons as well as electrophysiology, to establish that NPAS4 protein associates with multiple components of the NuA4 complex in neurons. They show that these components are co-localized with NPAS4 at activity-induced "peaks" -- i.e. cis-regulatory elements that can be identified by chromatin accessibility (ATAC), histone modifications (H3K27ac), as well as binding of NPAS4 and NuA4 components (CUT&RUN). Using BLISS-Seq, they link these activity-induced genomic regions with DNA double strand breaks (DSBs), which have recently been proposed to affect the genome at neuronal activity-induced genes. Because DSBs may represent a threat to the integrity of the genome in post-mitotic, long-lived neurons, Pollina and co-authors hypothesized that the NPAS4:NuA4 complex may play a role in supporting DNA repair at these sites. They show that a DNA repair protein, MRE11, co-localizes with NPAS4 at activity-induced peaks, and is reduced following KO of NPAS4. Finally, the authors present evidence that somatic mutations accumulate at regions affected by DSBs throughout the mouse lifespan, increasing from young (3 month) to old (24 month) mice at sites that lack NPAS4-mediated genome protection. They argue that the genome-protective role of NPAS4 may explain the shortened lifespan of NPAS4 KO mice.

Overall this is a very impressive, novel and interesting study. It presents a compelling case for what is a provocative and timely new hypothesis regarding the role of NPAS4 and of DSBs in the life of neurons. The wide range and high quality of genomic assays, computational analyses, and functional validation make this a strong paper that will be of wide interest to neuroscientists and genomicists alike.

We thank the reviewer for their comments and are pleased that they find the study "impressive, novel, and compelling."

My main critiques concern the clarity of presentation of some of the findings and methods. Because so many different, complex techniques are used, it can be difficult to comprehend each of the findings. In particular, some of the analyses are described in the text and figures using simplified short-hand, so that the reader must hunt through the Methods section to understand what is actually being shown. To take one example, it would be helpful when showing "activity-inducible genes" (in Fig. 3C) that are "bound or not bound by NPAS4" to include at least a bit more detail about how genes were assigned to these categories.

We apologize for the brevity of our explanations in the text and figure legends. Throughout the text, we have now made an effort to carefully explain how select genomic locations were chosen and our rationale for performing an analysis in a particular way. For example, to the reviewer's point about how genes were assigned as NPAS4-bound inducible genes, we have now updated this section of the text. We appreciate, as the reviewer suggests, that assigning NPAS4 binding to select sets of genes can be nuanced, as many transcription factors bind many kilobases away from their target genes and can be involved in complex looping interactions in three-dimensional space.

*Therefore, to enhance clarity of presentation, we now focus the majority of our analyses in Figures 3 and 4 (and associated Extended Data) on regulatory elements themselves, (defined as regions of ATAC-seq and/or H3K27ac). We now provide the definitions of these sites in a new figure, **Extended Data Fig. 9**. Furthermore, we subset these regulatory elements by high or low levels of NPAS4 binding (quartile 4 of IgG normalized NPAS4 CUT&RUN signal with a statistical peak of NPAS4 binding, versus quartile 1 of IgG normalized NPAS4 CUT&RUN signal with no statistical peak of NPAS4 binding). We believe this analysis provides a more direct assessment of how the NPAS4:NuA4 complex may influence damage and repair at its target sites. In addition to looking at all regulatory elements subset by NPAS4 binding quartiles, we further examine a subset of these sites that increase in ATAC and H3K27ac following neuronal stimulation (termed 'activity-inducible' elements; n=11,114 sites), and again subset by NPAS4 binding quartiles. We focus on these inducible elements to gain a better understanding of how DNA damage and repair are influenced at sites with dynamic changes to chromatin.*

*We describe in each figure legend which set of sites is being examined and how those sites were defined. For example, see legend of **Fig. 3c**.*

A second general comment is that there is not enough clarity about the biological replicates that were used. The figures generally show a single value or track for each experimental group, and I could not determine how many independent biological replicates were used for the various analyses. In cases where multiple replicates can be

shown without adding too much complexity/crowding to the figure, that would greatly help the reader to appreciate the reproducibility and reliability of the data.

*We appreciate the reviewer's concern about the lack of clarity for our biological replicates. We now provide all information about replicates in **Extended Data Table 1** where we detail the number of replicates and provide information on how these replicates are used in our Materials and Methods section. We also now provide replicate information in all figure legends.*

*We have performed all in vitro RNA-seq experiments in triplicate. A new timecourse of RNA-seq from hippocampal tissue paired with our BLISS-seq (DNA break) data was performed with 8-10 replicates per timepoint. Single-nucleus RNA-seq experiments were performed on 2-3 independent animals in NPAS4^{fl/fl} and Tip60^{fl/fl} animals. γ H2AX ChIP-seq was performed in triplicate. ATAC-seq was performed in triplicate. All CUT&RUN experiments were conducted at least twice and for most antibodies in triplicate, with the exception of one EP400 dataset generated using an antibody from Bethyl Laboratories. This dataset was intended to corroborate our other EP400 datasets, which have been conducted in duplicate with an antibody raised against EP400 from Abcam. NPAS4 CUT&RUN was performed 5 times in wild-type mice and in duplicate in NPAS4 KO mice. In addition, during the revision process we have added 3 additional replicates of NPAS4 CUT&RUN that show concordance with our merged NPAS4 CUT&RUN shown in **Extended Data Fig. 3b**. IP-MS experiments on hippocampal tissue were performed in triplicate for the initial isolation of the complex. IP-MS experiments conducted from fractionated lysates were performed in duplicate. BLISS-seq experiments were conducted using 8-10 replicates in our timecourse in wild-type mice and three to five times in our Npas4 and Tip60 cKO (Cre vs Δ Cre).*

We now further demonstrate the reproducibility of our data by:

- a) Where feasible, we include data from individual replicates in supplemental data figures. For examples, see **Extended Data Fig. 3b, 3i, 9b-c, 10a,b, 12e, 13b**.*
- b) Providing Information on replicate numbers for each dataset in **Extended Data Table 1**.*
- c) Providing more details on replicate numbers in all Figure legends*
- d) Showing the average \pm standard error of the mean for all aggregate plots shown in the paper and indicating how many replicates contribute to the aggregate.*
- e) Showing that our single-cell RNA-seq data was collected from 2-3 independent animals and that there is no difference in cell clustering between individual replicates in **Extended Data Fig. 5c,d**.*

Specific comments:

Fig. 1b - Does the top "Home" image show a merge of all fluorescence channels? This was unclear.

We apologize for not making this clear in the figure legend. Home cage does indicate a merge of all 3 florescent channels. We have now amended the figure legend to reflect this. The new legend reads:

“Representative images showing complete overlap of anti-NPAS4 and anti-HA antibody signals in hippocampus, following either 6 hr enriched environment (left) or 2 hr kainic acid (right) to induce NPAS4 expression. Scale bars: 100 μ m (left) and 20 μ m (right). Merge indicates combination of DAPI, anti-NPAS4 endogenous antibody staining and HA staining. Little to no NPAS4-FH is expressed in standard housing in the hippocampus (merge of all 3 channels shown)”

Fig 1f - It would be helpful to (also) show these data in an unnormalized format, especially because this figure is used to support that the NuA4 expression is limited to neurons.

We have modified the text to note that expression is not limited to neurons but is enriched in neurons. We provide the non-normalized expression matrix for NuA4 subunits in **Rebuttal Fig. 1** and demonstrate that normalization does not affect our finding that NuA4 subunits are most highly expressed in neurons. To appreciate differences across the cell types for each gene in the complex, we found it useful to normalize to expression of each gene by calculating a row Z score. Without normalization, the most highly expressed genes drive heat map scaling and obscure cell type expression patterns.

Figure 1. Non-normalized expression of NuA4 subunits in human cortex.

a. Single-nucleus RNA-seq expression levels for NuA4 components across cell types in human primary motor cortex, presented in row-normalized format on the left and non-normalized format on the right. Normalized values are displayed as a row Z-score to normalize for baseline differences in expression between genes and reveal consistent enrichment in neurons. Non-normalized data show log₂(CPM) counts values of the trimmed means for each gene in each cell type (trimmed means exclude the upper 25% and lower 25% of cells ranked by expression to calculate arithmetic mean). Data published and curated by Allen Brain Institute (see **Fig. 1f**).

b. Non-normalized expression of select lowly expressed NuA4 subunits, showing enriched expression in neurons.

Fig 1h - The color scale shows "read depth" for each CUT&RUN experiment -- but it is not clear if the read depth has been normalized. Specify if this showing RPKM (i.e. normalized for library size)

We now specify that read-depth has been normalized to 10 million reads to account for differences in sequencing depth in the legend and have amended the figure to read "Read-Depth Normalized Counts."

New Figure legend now states:

CUT&RUN signal for NPAS4 and complex components ETL4 and EP400 at NPAS4-binding sites in either unstimulated (0 hour) or stimulated (2 hour) hippocampal tissue. NPAS4-binding sites were defined using the peak calling algorithm SEACR¹ and were consistently identified in 4 of 5 NPAS4 CUT&RUN experiments. Each NPAS4-binding site (n=10,225 sites) is represented as a single horizontal line centered at the peak summit and extended out \pm 1kb. Intensity of color correlates with sequencing signal as indicated by the scale bar for each factor (0 to 50 read-depth normalization). Signal has been normalized to 10M reads for each factor. Plotted signal represents the average signal of at least 2 replicates per factor, except for EP400 Ab1 (n =1).

Fig 1i - Need statistical analysis, based on replicates, for the difference between 0hr and 2hr.

We now show \pm SEM intervals for our aggregate plots based on replicates. For statistical analysis we extracted the relevant CUT&RUN signal at NPAS4 sites and performed a two-sided Wilcoxon test comparing the signal at 0 and 2 hour timepoints. In the manuscript, we report the results for this test when averaging across the replicates but have also performed this analysis for each individual replicate. In all pairwise comparisons, the p-value is $p < 2.2e-16$.

Fig. 2b - Single cell transcriptome analysis was a little hard to follow here. What is driving the large separation between KO (Cre) and control neurons of the same type (e.g. CA1 pyramidal cells)? Showing the UMAP on its own is not sufficient, especially since it looks like the plot points are overlapping and obscuring each other such that it's hard to tell where each of the populations is present. Moreover, the two shades of gray (Δ Cre and "uninfected") are very hard to distinguish.

We apologize for the lack of clarity on the color choice and have updated the figures using better color contrast. As the reviewer noted, we were interested to find that depletion of either NPAS4 or Tip60 results in substantial shift in gene expression in our single-nucleus RNA-seq datasets, causing cells lacking these factors to subcluster within each cell type. We note that this effect is not driven by expression of viral transcripts because we have removed all viral features from the gene expression matrix.

The separation of Cre- and Δ Cre-infected cells within cell-type clusters is primarily driven by impaired expression of activity-induced genes in the Cre-infected cells, reflecting the inability of these cells to fully induce activity-dependent genes upon

depletion of either NPAS4 or TIP60. At this timepoint (6 hours post-stimulation), these activity-dependent genes are among the most highly expressed genes and differ across cell types, and therefore can contribute significantly to the PCs used in dimensionality reduction. For example, within dentate gyrus the cells primarily cluster according to dentate gyrus marker genes but lower-ranked PCs contain large numbers of inducible genes affected by loss of NPAS4 or Tip60, driving the Cre- and Δ Cre-infected cells to subcluster within the larger dentate gyrus cluster. We apologize for not pointing out this interesting finding the main text and have now added a short discussion highlighting this point (lines 201-208).

-- It was not very easy to understand how the set of “NPAS4 target genes” and “TIP60 target genes” was defined.

We apologize for the lack of clarity on defining NPAS4 and TIP60 target genes. For this analysis NPAS4 and TIP60 target genes were defined in each of the primary neuronal clusters as the genes downregulated in Cre-expressing cells upon deletion of Npas4 or Tip60, respectively. We now add sentences in the text to clarify this analysis.

“Across each of the neuronal subtypes, NPAS4 target genes and TIP60 target genes were defined for each principal neuronal cell-type as the genes downregulated in that cell-type by deletion of either Npas4 or Tip60.”

-- In Extended Figure 6, explain the units for gene expression ($\log_{10}(\text{TPM})$)?

Expression level data was generated by the Seurat package, using the \log_1 normalization of the Seurat-normalized counts for each gene (accessed via the “data” slot in the final Seurat objects). We have now indicated the units as $\log(\text{normalized counts})$ on the relevant axes and have added more information to the relevant figure legends and methods sections.

Fig. –c - It is not really very useful to show the distribution “f “express”on” level (presumably in something like $\log_{10}(\text{TPM})$ units, but this is not actually labeled) across a bunch of different genes. Instead I would suggest showing the distribution of the $\log(\text{fold-change})$ across genes. This should not affect the conclusions.

*The reviewer makes an important point about the difference between expression levels and the fold-change for each gene. We have plotted the expression level of the NPAS4 or Tip60 target genes in each cluster because we felt the visual comparison where we directly show that the Δ Cre expression level is consistently higher in each case clarifies the purpose and result of the experiment. As the reviewer pointed out, the \log_{FC} distributions are also important because they capture changes in expression on a per-gene basis. We now show \log_{FC} distributions in **Extended Data Fig. 6d**, and, as the reviewer anticipated, none of the conclusions are affected by this analysis.*

Fig. –f - How many different animals (biological replicates) were used here? Does each pair of cells come from an independent experiment?

We now indicate in the Figure legend how many mice were used for each experiment (N=16 cell pairs from 4 mice for NPAS4^{fl/fl} with kainate; N=12 cell pairs from 5 mice for NPAS4^{fl/fl} with PBS; N=12 cell pairs from 2 mice for Tip60^{fl/fl} with kainate; N=11 cell pairs from 4 mice for Tip60^{fl/fl} with PBS). Typically, several cell pairs (neighboring WT and cKO) can be obtained in each slice from a single animal. We have now indicated the number of animals in each experiment in the figure legend.

Line 188/189: Explain more clearly the test that was used to determine the 1766 differentially expressed genes. Some information about thresholds, p-values, FDR correction, effect size, etc., should be provided in the main text, not only in the methods.

We now explain more detail in the relevant figure legends and main text on how these 1766 genes were obtained in lines 231-232.

Line 202: I don't think the data show that NPAS4 and NuA4 are obligate partners. What is the evidence that they don't also have additional independent roles?

We agree that this wording is too strong given the data we have presented and have toned down this sentence in the text (line 246-248).

*We do however believe that NuA4 is required for NPAS4-dependent transcriptional induction. For example, our luciferase data indicates that truncations of NPAS4 which cannot interact with NuA4 do not transactivate enhancer/promoter regions (**Extended Data Fig. 7d,e**). Our mass spectrometry data from both brain and overexpression in HEK-293Ts implicates the NuA4 complex as the primary interactor of NPAS4 and ARNT2 (**Fig. 1d, Extended Data Fig. 2d, Extended Data Table 2**). It is possible that NuA4 may have additional interactors in the brain. However, our analysis of protein associated with NuA4 in the brain, obtained by pull-down of NuA4 component TIP60, implicates the NPAS4:ARNT2 dimer as a primary validated interactor (See **Extended Data Table 2**).*

Fig. –e - What is the statistical analysis here? How many animals (replicates) are used? What is the statistical basis of this statement: "While non-inducible sites did not increase 278 the number of DNA breaks with neuronal activity, inducible regulatory elements 279 displayed an increased number of breaks at 2 hours post-stimulation and a return to 280 baseline at 10 hours post-stimulation, suggesting that these sites are breaking and 281 subsequently undergoing repair (Fig. 3)"."

A new dataset generated during the revision process is now the basis for the BLISS-seq data presented in Figure 3. We have now added this dataset mapping DNA DSBs in the brain using more replicates in wild-type mice, allowing us to more comprehensively examine activity-inducible DNA break dynamics at genomic regulatory elements in the hippocampus. The revised experiment includes 8-10 mice at each of three stimulation timepoints (n=8 at 0 hr, n=8 at 2 hr, and n=10 at 10 hr). We now use Wilcoxon rank sum tests to compare the average signal across our replicates at the three timepoints. On

this basis we find that our new analysis confirms our previous findings that DSBs accumulate at activity-inducible regulatory elements in neurons, and expands upon our prior results to demonstrate how these dynamics relate to transcription (Fig. 3 g,h).

Our new wild-type timecourse recapitulates our previous findings. We find that activity-inducible regulatory elements increase in DNA break signal at 2 hours post-stimulation and return back down toward baseline by 10 hours post-stimulation, particularly in those samples in which the expression of activity-dependent genes is returning to baseline levels (Fig. 3f-h). We use Wilcoxon rank sum tests for BLISS signal (average of 8-10 replicates per timepoint) at 0 vs 2 vs 10 hours to demonstrate the statistical basis of these changes during stimulation (Fig. 3f-h; Extended Data Fig. 12d). We also observe that these dynamics are amplified at sites with high levels of NPAS4 ('NPAS4 Q4') (Fig. 3f,h, Extended Data Fig. 12). We now show the other regulatory elements which do not fall within our definition of activity-inducible elements do not show a statistically significant increase BLISS signal using Wilcoxon rank sum tests (Extended Data Fig. 10j).

ED Fig. 10f - Hard to see the colors. What are the criteria for defining MRE11 or NPAS4 peaks?

We have now updated the colors to make them more distinct and now show these data in Extended Data Fig. 13d. We apologize for the lack of clarity on calling peaks. All peaks were called using SEACR¹, which is optimized for CUT&RUN peak calling. Final peak sets include sites found in 4 of 5 NPAS4 replicates, 2 of 2 MRE11 replicates, and 3 of 4 RAD50 replicates. In addition to the information presented in the methods section, we now add a better description to relevant figure legends.

3i,j - MRE11 binding is reduced but not completely abolished by lack of NPAS4 or TIP60. Does this suggest there are additional, NuA4-independent mechanisms for recruiting DNA repair?

We believe that there are additional means by which MRE11 may be recruited to these activity-inducible sites. For example, MRE11 itself is known to have the ability to bind to broken DNA ends and so may be directly targeted to the genome. Moreover, MRE11 has been suggested to interact with components of RNA PolIII at highly expressed genes². We now mention these possibilities on lines (459-462) of the text.

Fig 3i - The Delta-CRE mice should not be labeled as "WT". Instead they could be labeled "control"

We have now changed the WT designation to Control or Δ Cre, where relevant. The new BLISS dataset for Figure 3 uses only wild-type mice, n=8 at 0 hr, n=8 at 2 hr, and n=10 at 10 hr.

Fig. 4 - The evidence for mutation accumulation in aging was intriguing but relatively thin compared with the rest of the paper. Mutations were analyzed in NPAS4 cKO mice

at only one time point, making it hard to directly assess aging.

Fig 4d,e - What is the absolute (not normalized) rate of mutations?

*We now provide non-normalized values and additional data on the types of mutations in **Extended Data Fig. 17e**. However, given the nature of the assay, we believe it is important to normalize mutational frequency for a given locus across replicates. This is because not every site has the same baseline mutation rate and each independent mouse may have a slightly different set of amplicons that pass QC threshold (for example there is some variability in how well select sites were amplified in a given sample). Because different sites do have different mutation frequencies, we found it best to normalize by site to in order to more accurately compare across ages.*

Fig 4e - "Each point represents one site sampled from one mouse." How many mice are shown here? The analysis should take into account this nested data structure, rather than treating all the data points as independent.

*We now provide the non-normalized mutation rates in young mice in the supplement (**Extended Data Fig. 17e**). The dataset quantifying mutation rates across wild-type mice of different ages contains data from 4 individual mice at 3 months, 4 mice at 12 months, and 3 mice at 23-27 months. Because of the nature of this assay, we believe our analysis is the most appropriate way to group mutation rates for multiple sites across multiple mice. Each primer has a UMI, so each individual UMI family represents a sampling from a single allele template. Each UMI family that passes QC thresholds could therefore be considered a biological replicate sampling that genomic location from one cell from one mouse. Conceptually, this is comparable to grouping of samples in a single-cell RNA-seq dataset.*

Because the distribution of sites where we have sufficient sequencing depth for a site of interest can vary from mouse to mouse, it can be difficult to get significance when we separate out individual mice. Therefore, we chose to analyze all samples together for statistical purposes but do indicate in the legend of Figure 4 that we have performed these experiments using multiple biological replicates at each timepoint.

*In the course of our revision process, we sought to strengthen our datasets examining changes to mutational accumulation in NPAS4 KO mice. We generated an additional dataset from FACS-isolated NeuN+ neurons isolated from 11 additional NPAS4 KO mice and 8 additional WT mice examining mutations at an expanded set of NPAS4 binding sites. Given our interest in DNA DSB repair, we sought to specifically examine insertion and deletion events, which are the main signature of DNA DSB failure. However, while we observed a consistent trend towards increased insertion and deletion events across our multiple datasets, these trends never reached significance, as these events were extremely rare (**Rebuttal Fig. 2**). Deletion of NPAS4 is expected to both reduce activity-dependent gene transcription and thereby decrease the chances of generating transcription-coupled damage and mutations at the relevant gene's regulatory elements. However, at the same time (due to the absence of NuA4 at these*

same elements) there is also likely a decrease in DNA repair which may contribute to an increase in DNA damage and accumulating mutations. Because of these opposing effects on DNA damage, it is difficult to predict what the cumulative effect of deleting NPAS4 would be on DNA damage, and this may account for the relatively modest effects that the absence of NPAS4 has on the accumulation of mutations. We anticipate that capturing these relatively modest changes may be especially difficult without generating much more expansive datasets. Given the scope of the current manuscript, we believe it is most prudent in our revised manuscript to remove our analysis of the DNA mutations in NPAS4 KO mice. Rather, we believe gaining clarity on this issue will require additional, complementary validating techniques to identify insertion/deletions and more expansive datasets that were not feasible to generate in our revision timeline.

Line 2145 missing a word: "we ranked all regulatory in our dataset"

We have now amended this in the text.

REFEREE #3

Overall, this manuscript contains extremely interesting data that represent an enormous amount of excellent work, involving an impressive arsenal of complex technologies. The results on the NPAS4 interaction with the NuA4/TIP60 chromatin modifier complex in neurons are consistent and in all regards convincing. The DNA repair connection, on the other hand, raises a number of questions of experimental and conceptual nature.

Major comments:

I should start by complimenting the authors for the high scientific quality of their study, including the use of proper controls in all of the experiments. Striking examples are:

1. The approach to generate NPAS4-FLAG-HA and Arnt2-FLAG-HA knock-in transgenic mouse lines using Crispr-Cas9 technology, in which FLAG and HA epitope tags are fused in frame to the 3' end of the endogenous gene for the isolation of NPAS4 protein complex directly from the brain is very powerful as it ensures physiological expression levels and normal regulation. The subsequent interaction between NPAS4 and all known subunits of the NuA4/TIP60 chromatin modifier complex and one new subunit is absolutely stunning (Fig. 1 and Ext. Data Fig. 1).

We thank the reviewer for these compliments and are pleased that the biochemical complex identification is considered robust.

2. Another elegant step is the nested PCR strategy employed in the library preparation to amplify viral transcripts in order to identify individual infected cells and discriminate them from non-infected cells, which strongly improves the power of the analysis (Fig. 2a,b, Ext. Data Fig. 5e).

We again thank the reviewer for appreciating this experimental design.

3. A third nice example of scientific rigor is the generation of a mutant form of NPAS4 that cannot interact with NuA4 (Ext. Data Fig. 1f), used in luciferase reporter assays (Ext. Data Fig. 7d,e). This is indeed a very powerful experimental approach. Also, the stereotactic injections of AAV transgenic CRE-marker constructs and the subsequent analyses, including single-nucleus RNA-sequencing and current recordings of neural circuits as in Fig. 2 are truly impressive.

*We thank the reviewer. We have now updated the reference to the western blot identifying the truncation mutant as **Extended Data Fig. 2c**.*

Altogether the paper presents very solid and conclusive evidence that NPAS4 and NuA4 can function together as a discrete protein complex in neurons upon stimulation. For this part of the manuscript there are no comments, only compliments.

The DNA repair connection

As reported before (and cited in the paper) the NuA4 chromatin modifier complex on its own exerts two functions: control of gene expression and helping in coordinating DSB repair, although this latter activity appears to this referee still rather ill-defined in current literature.

We appreciate that the current literature remains unclear regarding all the DNA damage processes the NuA4 complex contributes to, particularly in neurons where DNA repair processes in vivo remain to be well characterized. However, we have cited the limited number studies from yeast to mammalian cells that suggest a role for select subunits of this NuA4 complex in DNA DSB repair (see text lines 174-176).

Obviously, the DNA repair machinery has to operate in the context of chromatin and repair transactions involve all known chromatin modifications. Therefore, it is a priori not very surprising that for virtually all DNA repair systems, including DSB repair pathways, many connections with chromatin-modifying protein complexes have emerged in the past two decades. However, these are always transient and hence dynamic in nature, which is intrinsic to the requirements of repair of damage that is stochastic, random and can occur at any time at any location in the genome. I presume that this is the same in the case of the NPAS4/NuA4 complex and the repair proteins investigated here and that some of the specific questions raised below relate to this phenomenon. Presumably, in this light the link between genuine DNA damage induction, intrinsically dynamic DNA repair proteins and the NPAS4/Nu4/TIP60 chromatin modifier complex has to be considered.

Also, it has been amply demonstrated (as cited in the manuscript) that transcription is linked to induction of DSBs, which are usually transient as well because persistent DSBs are very potent activators of cell cycle checkpoints and the DNA damage response, and e.g., lead to SCARS, which are associated with cellular senescence. Presumably, part of these 'DSBs' might simply be intermediates of normal DNA metabolism that are captured in the middle of a DNA transaction at the moment of cell lysis, which usually involves detergents. Examples could be topoisomerase-DNA intermediates involved in transcription-related unwinding reactions. They would have been sealed moments later. A small fraction of the transcription-related DSBs are undoubtedly toxic. However, this raises the question how to discriminate the artificial breaks of DNA metabolism intermediates from the long-lived, toxic DSBs and how to quantify them in a reliable manner?

We now attempt to be more specific in the text and make it clear that we think the complex is important for DNA break repair that occurs downstream of inducible transcription. One idea is that regions that undergo these types of consistent DNA breaks, including those generated by topoisomerase enzymes, may in rare cases result in long-lived DSBs if the process goes awry. To really distinguish long-lived DSBs from these transient intermediates would likely require looking via live imaging at broken DNA ends themselves over a timecourse of stimulation. By mapping sites of

reproducible DNA breaks in the brain, our manuscript has identified genomic locations that would be prime candidates for such future studies.

Given the inherent limitations of measuring DNA break dynamics in vivo, we felt it would be crucial to use multiple, independent measurements of DNA DSBs to understand their localization and their dynamics following neuronal stimulation. We have attempted to capture DNA DSB signaling by a variety of methods including: a) γ H2AX ChIP-seq b) recruitment of the DSB repair enzymes MRE11 and RAD50 in an activity- and NPAS4-dependent manner c) direct measures of DNA breaks by BLISS-seq using multiple replicates and pairing this data with RNA-seq to better understand the relationship between DSBs and transcriptional induction and d) using an alternative DNA DSB mapping technique, END-seq to corroborate BLISS findings.

*Because technical damage that occurs during the course of the experiment should be distributed evenly across all replicates, the consistency of our results across many replicates in the new BLISS timecourse in WT mice (n=8 at 0hr, n=8 at 2hr, n=10 at 10hr, see **Extended Data Fig. 12**) argues that the differences we observe in BLISS signal across timepoints reflect changing levels of DSBs at these genomic locations following neuronal stimulation rather than simply artificial breaks introduced by sample processing.*

To incorporate additional methods to measure DSBs, we now also provide an additional experiment using a secondary DSB-mapping method, END-Seq, as a complementary method to observe DSBs downstream of neuronal activity in cultured neurons. END-seq avoids chemical fixation by embedding unfixed cells in agarose before in situ adapter ligation, thereby avoiding artificial breaks that can accompany chemical fixation. Our findings with END-seq corroborate our in vivo data providing support that there is an accumulation of DSB signal at NPAS4-bound sites in an activity-dependent manner.

Specific comments:

1. To assess the effect of NPAS4 and TIP60 on the formation and repair of DNA breaks at NPAS4- or TIP60-bound sites and across the neuronal genome, Cre-expressing or (inactive) Δ Cre-expressing AAVs were injected in the brain of NPAS4^{fl/fl} or Tip60^{fl/fl} mice and nuclei isolated at 0, 2, or 10 hr post-stimulation for BLISS-seq. The use of a recombination-deficient version of Cre (Δ Cre-GFP) is an excellent control. However, one potential complication that might interfere with some of the assays in this study is that CRE will induce DSBs by itself, not only in flox-sequences but also off-target, at other sites in the genome as amply demonstrated (e.g., see Loonstra A, et al.2001. Growth inhibition and DNA damage induced by Cre recombinase in mammalian cells. Proc. Natl Acad. Sci. USA 98:9209-14. doi: 10.1073/pnas.161269798, PMID: 11481484). In wt cells the effects may be limited, as these lesions generally can be quite efficiently repaired but in case of DSB-repair mutants, as in this study, this may exert significant negative effects including activation of DNA damage signaling and increased DSBs. This may explain at least in part the generally higher number of DSBs in Fig. 3f and in Ext. Data Fig. 9h-j after KA induction and even

before KA induction in the mutant which have active Cre, versus wt, which have the inactive Δ Cre. It would also explain the elevated number of genome-wide DSBs at all time points with and without KA activation in floxed mutant brain only with CRE expression and in NPAS4 mutant conditions, but not in wt, as displayed in Fig. 4f. and for the Tip60-conditional inactivation by Cre presented in Ext. Data Fig. 14a. This is relevant for the discussion on p. 13 (lines 291 and further), as well as the discussion on p. 18 (lines 401 etc.), which is speculative and to me not very plausible. On the other hand, this would argue that NPAS4 and Tip60 mutations affect (Cre-induced) DSB repair.

*The reviewer raises an important point regarding the use of CRE and DSBs. To further test whether the increase in DSBs observed in NPAS4 cKOs is primarily driven by expression of CRE, we performed BLISS-seq in germline NPAS4 KO vs WT mice at 10 hours post-stimulation. However, the signal-to-noise in this dataset was not as high quality as our other BLISS-seq datasets and therefore we are hesitant to include this data in the full manuscript. We did find, however, elevated BLISS signal in the global KO at sites with high levels of NPAS4 binding (Q4). This result aligns with our other data suggesting increased DNA breaks upon loss of NPAS4, amplified at sites bound by the NPAS4:NuA4 complex (**Rebuttal Fig. 3**).*

We also attempted to control for the effects of CRE expression on DNA breaks by also introducing CRE into wild-type mice, where we do not see a substantial difference between the Cre- and Δ Cre-treated nuclei in total cuts across the genome. That said, we acknowledge the reviewer's point that, in wild-type cells, CRE-mediated breaks might be efficiently resolved, whereas in repair mutants CRE-mediated breaks may persist. Even so, this global result is of interest because it newly implicates NPAS4 broadly in DNA break repair. However, we note that we do observe amplified effects on DNA break accumulation in NPAS4 mutants at the sites where the complex is binding and furthermore observe recruitment of DSB repair factors to sites with high levels of NPAS4. Taken together, these results lead us to conclude that there are specific local effects on DSBs mediated by these factors. We now acknowledge the interesting possibility that CRE may also be sensitizing our mutants in the text.

Are mutant cells sensitive to DSB-inducing agents such as ionizing radiation, bleomycin or topoisomerase inhibitors?

The reviewer proposes an interesting experiment to address whether NPAS4 can be regarded as a repair factor due to increased sensitization to DSB-inducing agents upon loss of NPAS4. This is a compelling experiment that we would like to do in the future. Namely, these experiments would have to be coupled to inducing neuronal activity, as we have found that simply treating neurons with DNA damaging agents is not sufficient to induce NPAS4 in the absence of neuronal depolarization (**Rebuttal Fig. 4**). We believe these experiments would be best conducted in vivo, where we have generated more data examining features of DNA damage in response to neuronal activity and have a firmer understanding of NPAS4: NuA4's assembly, binding sites, and role in circuit function than in immature cultured neurons. Performing this experiment in vivo, however, is more challenging and we believe lies outside the scope of the current manuscript.

Figure 4: Treatment of Cultured Neurons with Damage-Inducing Agents

a. Western blot of cultured mouse neurons treated with 55 mM KCl to induce membrane depolarization and calcium influx and/or with various DNA-damaging agents, including etoposide (10 μ M), H2O2 (100 μ M), and Paraquat (25 μ M). NPAS4 was induced at the protein level when KCl was added with damage agents, suggesting the NPAS4's role in repair works selectively in depolarized neurons.

2. MRE11 is a suitable marker for DSBs in vivo, and the co-localization and presumed interaction of MRE11 and NPAS4/NuA4 complex reported here is sound, but it raises a number of questions. The nature of this interaction is still unclear and could well be indirect, transient and non-selective. Affinity of MRE11 with chromatin modifiers and accessible genomic regions and with ongoing DNA transactions would explain “that MRE11 is often present at NPAS4-bound elements even in unstimulated neurons” (lines 318-319), but also at other sites in the genome as observed in this study. Slight differences in affinity to specific binding sites (e.g., chromatin modifications) could explain why e.g., MRE11 is found more at NPAS4 sites than at FOS sites in the genome as found here. Photobleaching experiments of fluorescently-tagged MRE11 and other partners would be required to determine whether the interactions are dynamic, based on short-lived affinities.

To get better insight it would be informative to determine the presence of other components of the MRN complex: NBS1 and/or Rad50. Even more critical: DSB are very genotoxic, can frequently cause chromosomal aberrations, but are generally not particularly mutagenic (i.e., causing base substitutions), because of their low abundance, compared to much more abundant oxidative lesions and abasic sites due to spontaneous hydrolysis. MRE11 is important for DSB repair. This raises the question what about the link with other DNA repair systems such as base excision, nucleotide excision and transcription-coupled repair, which are important for preserving long-term neurofunction? Were any of these proteins or repair pathways enriched in the Mass spec analysis for NPAS4 interactors in Fig.1d that identified all components of the

*As requested by the reviewer, we have re-analyzed our mass spec data from immunoprecipitation of NPAS4 from the brain and HEK-293Ts and do not find any reproducible interactions with other repair system machinery. That said, we have analyzed recent genome-wide datasets capturing synthesis-dependent repair in iPSC-derived human neurons. We find enrichment of this repair signature at NPAS4-bound sites (**Extended Data Fig. 12f**), which could reflect either DSB or SSB repair, and raises the possibility that NPAS4 is engaged in additional repair pathways beyond DSBs. This is an exciting future direction.*

*As also requested by the reviewer, we conducted CUT&RUN for another member of the MRN complex, RAD50, and find the same activity-dependent recruitment of this factor preferentially to NPAS4-bound sites (**Fig. 4a,b; Extended Data Fig. 13f**). We believe this result provides important corroboration of our data suggesting that NPAS4 contributes to regulated DNA DSB pair at NPAS4:NuA4-bound sites downstream of neuronal activity, in part by facilitating the recruitment of specific repair enzymes. It remains possible that MRE11 and its associated complex might play important underappreciated roles in transcriptional induction in neurons, which is a topic outside the scope of this study but nonetheless interesting.*

We appreciate that the nature of the MRE11 recruitment is still unclear and could be either direct, through a biochemical interaction, which has been reported for NuA4

component TRRAP³, or indirect, resulting from modifications to chromatin that retain MRE11. We do acknowledge that there are other mechanisms by which MRE11 might be targeted to neuronal genomes, but remain confident that perturbation of NPAS4 impairs the activity-dependent recruitment of this factor to select sites in chromatin. We also note that the recruitment of MRE11 to NPAS4-bound sites is not driven by differences in histone acetylation and chromatin accessibility at these sites since sites with equivalent levels of ATAC-seq and H3K27ac but no NPAS4 have lower levels of MRE11 and RAD50 than NPAS4-bound sites in our CUT&RUN datasets (see **Extended Data Fig. 14a,b**). It is also relevant that we find a greater degree of overlap between sites with a peak of NPAS4 (but not a peak of FOS) and MRE11/RAD50 than sites with a peak of FOS (but not NPAS4) (**Rebuttal Fig. 5**).

3. Mutation analysis (Fig. 4): Questions that emerge: how do absolute (not relative) mutation frequencies of the different classes of mutations (base substitutions, small deletions/insertions, large deletions, translocations, etc.) compare with mutation frequencies known in aging for brain and other organs/tissues? Are they in the same order of magnitude? What type of mutation signatures are observed?

See comments to Referee#2 (copied here)

We now provide non-normalized values and additional data on the types of single nucleotide variants in the supplement (Extended Data Fig. 17e). In our assay, we quantify the total number of base changes in a PCR-amplified site (amplicon) divided by the total number of bases assessed. In the case of specific types of mutations, for example C > T (G > A), we divide by the total number of C or G bases assessed in our amplicon. This quantified rate may not be directly comparable to other published measurements, particularly those that quantify mutations on a genome-wide scale or in much larger genomic windows. Indeed, few studies have looked specifically at mutation rates in mouse neurons at select active regulatory elements. That said, other studies,

primarily in human brains, have reported base changes with a range of $(10^{-5}$ to $10^{-8})^{4-7}$, compared to our range of $(10^{-5} - 10^{-6})$. As all of our sites are accessible regulatory elements that have been suggested to have higher rates of mutation than less accessible regions^{5,8}, our rates might be expected to be somewhat higher. We do not, however, think that a direct comparison can be made.

Moreover, given the nature of the assay, we believe it is most informative to normalize mutational frequency for a given locus across replicates. This is because not every site has the same baseline mutation rate and each independent mouse may have a slightly different set of amplicons that pass QC threshold (for example there is some variability in how well select sites were amplified in a given sample). Because different sites do have different mutation frequencies, we found it best to normalize by site to in order to more accurately compare across samples.

In the course of our revision process, we sought to strengthen our datasets examining changes to mutational accumulation in NPAS4 KO mice. We generated an additional dataset from FACS-isolated NeuN+ neurons isolated from 11 additional NPAS4 KO mice and 8 additional WT mice examining mutations at an expanded set of NPAS4 binding sites. Given our interest in DNA DSB repair, we sought to specifically examine insertion and deletion events, which are the main signature of DNA DSB failure. However, while we observed a consistent trend towards increased insertion and deletion events across our multiple datasets, these trends never reached significance, as these events were extremely rare. We believe that capturing these events in mice, whose lifespan is relatively short is difficult. Deletion of NPAS4 is expected to reduce transcriptional activity (thereby decreasing the chances of generating transcription-coupled damage and decreasing the chance that mutations occur). However, deletion of *Npas4* will also lead to a loss of NuA4 at the *Npas4* sites and thus any damage that does occur at these sites is not likely to be repaired and this could tip the balance towards increased damage. Given these potentially opposing effects of *Npas4* loss on DNA damage, we expect that it will be difficult to detect significant changes in DNA damage due to *Npas4* deletion without much more expansive datasets. Given the scope of the current manuscript, we believe it is most prudent to remove our analysis of the DNA mutations in NPAS4 KO mice. Rather, we believe gaining clarity on this issue will require additional, complementary validating techniques and more expansive datasets that were not feasible to generate in our revision timeline. If the reviewer feels that this is essential to the publication of our manuscript we could delve deeper into this matter. Please see **Rebuttal Figure 2**.

The NPAS4-KO mice show a quite unusual survival curve in Fig. 4h, with equal mortality rates over their entire lifespan. What is the explanation and what is the main cause of death? How is neural functioning affected in physiology and behavior?

The proximate cause of mortality in the NPAS4 knockout mice remains unclear. Previous published studies have suggested that *Npas4* deletion may lead to changes in glucose metabolism⁹. However, we have established that these mice show no substantial difference in weight or glucose metabolism compared to wild-type littermates across different ages (**Rebuttal Fig. 6**). Our most likely explanation is compounding deficits in inhibitory control, which has been linked to decreased lifespan in other organisms^{10,11} Unfortunately, due to COVID-19 restrictions we were forced to collect tissue from our aged NPAS4 WT vs total body KO mouse cohorts prior to completion of behavioral and physiological characterization. However, we have now completed a lifespan study on a small cohort of conditional *CaMKIIa-Cre; Npas4^{fl/fl}* mice where NPAS4 is knocked out in a subset of excitatory neurons in the forebrain. These mice also show a decreased lifespan compared to littermate controls (**Extended Data 18**), which further argues that these mice have an abbreviated lifespan due to neuronal defects.

[This has been redacted]

[This has been redacted]

4. Conceptually, the authors argue that the NuA4/TIP60 chromatin modifier complex in neurons may function as a guardian of genome integrity through activity-dependent DNA repair at promoters and enhancers. However, this would only apply to MRE11-dependent DSB repair processes. Moreover, this concept would account only for part of the genome (determined to be 1766 activity-regulated genes) as other promoters and enhancers of e.g., constitutively expressed genes, which generally are important for cell viability, would not enjoy the same protection and hence might limit neuronal lifespan.

Although we provide evidence that NPAS4:NuA4 recruits MRE11 repair machinery to sites of activity-dependent DSBs in the neuronal genome, we acknowledge that other repair pathways and other repair factors independent of MRE11 may contribute to DNA repair at sites undergoing activity-induced DSBs. Defining the full repertoire of mechanisms of DNA repair downstream of NPAS4:NuA4 in stimulated neurons will be an active area of future investigation. We now state this explicitly in the revised manuscript on lines 628-639.

We agree with the reviewer that this extra layer of protection against persistent DNA damage seems to be enhanced at sites bound by the complex. Other sites must therefore rely on more canonical DNA repair coupled to constitutive transcription or as yet undetermined mechanisms. Our study provides evidence of a context-specific repair mechanism that is synchronized with synaptic activity and is targeted within each cell-type to key activity-inducible regulatory elements such as the promoters of activity-dependent genes. Across their lifespan, neurons are repeatedly stimulated and undergo acutely elevated damage at these sites and thus neurons might need additional layers of DNA damage repair at these sites that are linked to synaptic activity. While other sites that do not have these protective mechanisms likely limit neuronal function, these activity-inducible sites are particularly critical for long-term adaptation to environmental cues, including aspects of learning and memory. Damage to these genomic locations over time, for example in the case of impaired NPAS4:NuA4 function, may therefore underlies aspects of decreased plasticity and adaptability in the brain over the course of an organism's lifespan.

Minor comments

1. Ext. Data Fig. 1d: The MW marker for the TUJ1 loading control is indicated to be 15 kDa, whereas it is 50 kDa.

We have now corrected this error.

2. It is unclear why the reader is referred to Ext. Data Fig. 1f in the context of the NPAS4 mutant which has lost the interaction with NuA4 (line 197), as this panel only shows the wt NPAS4. Moreover, it is clearer to readers to refer to this mutant in the main text as “a truncation mutant of NPAS4”, instead of “a mutant form of NAPS4”, to discern it from a point mutant (which probably would have been as a better separation-of-function mutant).

*We apologize for error and now refer to mutants as “truncated forms of NPAS4 that cannot interact with NuA4” (line 237). We now correctly site **Extended Data Fig. 2c** as the figure which demonstrates that truncation mutant interaction with NuA4.*

3. Fig. 3e: Colors to distinguish the line for promoters and for introns could be made more distinct.

Having generated a new WT timecourse of BLISS-seq data, we have now removed this specific panel.

4. Fig. 4a: The authors “aged cohorts of NPAS4 knockout (KO) mice and their wild-type littermates and collected hippocampal tissue from young (3 month), wild-type mice, middle-aged (12-17 month) wild-type and NPAS4 KO mice, and old (23-27 month) wild-type mice raised under standard housing conditions” (line 344). Why was only one age (12 months) used for the NPAS4 KO mouse cohort? To distinguish developmental from progressive degenerative symptoms associated with aging it would have been informative to have at least one interim cohort at young age.

In the revised manuscript we have toned down the emphasis on the findings about somatic mutations. We have retained the data showing an age-dependent accumulation of somatic mutations across a series of wild-type mice increasing in age. We believe this data is the strongest evidence for the conclusion we support – that sites bound by the NPAS4:NuA4 complex are relatively protected from aging-associated genome instability. As discussed above, we now provide additional details on the non-normalized mutation rates and the contributions of individual mice to each dataset in supplementary figures and legends. As discussed above, deletion of NPAS4 is expected to both reduce transcriptional activity (thereby decreasing the chances of generating transcription-coupled damage and generating mutations) and decrease repair (leading to an increase in mutations), thereby capturing reliable changes may be especially difficult without generating much more expansive datasets. Given the scope of the current manuscript, we believe it is most prudent to remove our analysis of the DNA mutations in NPAS4 KO mice. Rather, we believe gaining clarity on this issue will require additional complementary validation techniques and more expansive datasets that were not feasible to generate in our revision timeline.

Overall, I am impressed by the large amount data and high quality of the work. The part on the NPAS4 link with NuA4 chromatin modifier complex is sound. The repair connection raises a number of questions and will need significant strengthening.

We now strengthen our arguments by:

- 1) *Adding a new BLISS timecourse with many replicates in wild-type mice, to measure DSBs directly in the mouse brain in vivo for the first time. We pair these new datasets with RNA-seq to better understand the relationship between DSB signal and transcription. These data are reproducible across many replicates and expand upon our earlier findings. We now show that NPAS4-bound sites undergo*

- increased DNA breaks at 2 hours post-stimulation and a return to baseline by 10 hours post-stimulation. These data more clearly demonstrate the relationship between transcriptional induction, binding of NPAS4:NuA4, and DNA DSBs.*
- 2) Using a complementary technique to BLISS, END-seq, to show NPAS4-bound sites undergo activity-dependent increases in DSB signal. We show greater enrichment of DSB signal at NPAS4-bound sites relative to sites bound by another activity-dependent transcription factor, FOS, suggesting NPAS4 is more specifically targeting DSB sites in the genome.*
 - 3) Performing an analysis of previously published maps of DNA repair hotspots (SAR-seq) in iPSC-derived human neurons at NPAS4-bound sites relative to FOS-bound sites. This analysis shows elevated signatures of DNA repair at NPAS4-bound sites and aligns with our BLISS-seq and END-seq data.*
 - 4) Performing CUT&RUN for another DSB repair factor, RAD50, in stimulated neurons in vivo. In addition to our original results with the DSB repair protein MRE11, we now show that RAD50 is also present at NPAS4-binding sites and its binding is highly inducible upon neuronal activation.*
 - 5) Demonstrating that acute deletion of NPAS4 results in decreased recruitment of both MRE11 and EP400 to NPAS4 target sites. Both factors are known to be critical for DSB repair.*
 - 6) Demonstrating that deletion of NPAS4 results in an increase in DNA breaks at NPAS4 target sites as well as an increase in DNA breaks across the genome. We also observe this global increase in DNA DSBs upon acute removal of NuA4 subunit TIP60.*
 - 7) Performing an improved analysis of mutation signatures at NPAS4-bound sites in both young and aged mice. This new analysis demonstrates that NPAS4-bound sites are relatively protected from the accumulation of age-associated mutations.*

REFEREE #4

In the manuscript from Pollina et al, the authors generate a mouse model to investigate the NPAS4 transcription factor in neurons. They find that NPAS4 interacts with the NuA4 chromatin modifying complex and that this complex mediates transcription of an overlapping set of target genes and functions in the same subset of cell types. In addition, they propose that NPAS4 works together with NuA4 to mediate rapid repair of DNA double strand breaks (DSBs) in the vicinity of NPAS4 binding, and that this confers protection to genome integrity. In Figure 4, the authors show differences in mutation numbers between regions with NPAS4 binding compared with no binding, and between NPAS4 KO and WT, suggesting that NPAS4 plays a protective function in these genome locations during aging.

The data on the relationship between NPAS4 and NuA4, showing a physical interaction and evidence of common transcriptional activities is robust, and the mouse work is impressive. The major issue with this manuscript is that the data presented (primarily in Figure 3 and supporting supplementary data) do not support the conclusions around double strand breaks and their repair. Specifically:

1. Without further validation or clarification, it's not clear whether the BLISS data can be reliably compared across samples in a quantitative manner. In the paper where this technique was developed, this limitation is highlighted (Bouwman et al Nat Protocols 2020), and the data here show considerable noise and apparent inconsistencies (for example, why are there so many more DSBs at non-induced sites in Figure e?).

We appreciate the reviewer's concern about comparison of BLISS signal across samples. We felt it was critical to perform these experiments in vivo alongside genetic manipulation of the NPAS4:NuA4 complex, which required using low numbers of isolated nuclei. We elected to use BLISS-seq because it was most amenable to in vivo experiments addressing these specific questions. For example, other techniques such as END-seq require larger number of cells as input and require efficient dissociation of cells in suspension, which is challenging to perform from dissected brain tissue without inducing DNA damage during the procedure. BLISS has been shown in a number of other studies in non-neuronal cells to faithfully mark DNA breaks at highly expressed genes and genes with high levels of paused RNA Pol II^{13,14}. Therefore, it is perhaps not surprising that we observe DSBs at sites that are highly transcribed but not inducible following neuronal activity.

To address the reviewer's comments about BLISS comparison across samples, we have undertaken the following experiments.

a) A timecourse of DNA break induction measured by BLISS-seq in using many replicates (n=8-10 mice) for each timepoint. While we appreciate the BLISS performed on low numbers of freshly dissected cells may be subject to some noise, we attempted to address this concern by performing many replicates to better identify reproducible changes. In addition, we collected parallel RNA to better compare transcriptional induction levels with BLISS and to exclude samples that do not show robust stimulation (which can confound reproducibility).

b) A new experiment measuring DNA DSBs via a second assay (END-seq) to provide corroboration of activity-inducible DSBs at NPAS4-bound sites.

*Both using BLISS-Seq and END-Seq, we observe an increase in DNA break signal at two hours post KA stimulation at NPAS4-bound regulatory elements. Indeed, we find the magnitude of these changes is maximal at sites with high levels of NPAS4 binding (**Fig. 3f,h; Extended Data Fig. 12d,e**). In the new BLISS timecourse data, we note that the average signal returns towards baseline at 10 hours, especially in samples where the levels of activity-inducible genes are returning to baseline. We believe these data reflect physiologically relevant changes in the level of DSBs at these sites alongside their transcriptional induction kinetics. We show this induction and return towards baseline is consistent across each of many replicates (**see Extended Data Fig. 12e**). Moreover, we have now identified reproducible peaks of enriched BLISS signal in 2 hour stimulated neurons that are consistent in at least 5 of our 8 replicates (**Extended Data 10h**).*

Moreover, the recent paper from Nussenzweig and colleagues (Nature 2021) mapped sites of repair in post-mitotic neurons and found no evidence of DSBs (albeit in a very different system) using a similar end mapping approach, raising the possibility that many of the sites mapped here are not physiologically relevant.

We thank the reviewer for raising an important question about how to reconcile our findings with recent work suggesting single-strand break repair pathways are active at many genomic regulatory elements in iPSC-derived human neurons. We appreciate that differences in the in vivo mouse vs human iPSC derived neuronal cultures may account for some of these differences. We note that the BLISS assay may be more sensitive than END-seq and other end mapping approaches, particularly for detecting events that do not occur in every cell at the exact same location.

We also point out that Wu et al. did not specifically manipulate neuronal activity, while we focus on NPAS4's role in facilitating repair of activity-inducible DSBs. The human neuronal cultures used in the Nussenzweig study may not be sufficiently mature for proper induction of neuronal activity, as studies from our own lab have found that many days of culturing and specific conditions are required to achieve the requisite maturity levels for properly coordinated activity-dependent gene expression.

*As the reviewer suggested, we analyzed the SAR-seq data¹⁵. Although this assay is not limited to detecting DSBs, it should detect any synthesis-dependent repair that may occur in the course of repairing the induced DSBs we detect at activity-inducible sites in the genome by NHEJ or other mechanisms. Our analysis of the SAR-seq data reveals enrichment of the repair signal at NPAS4-bound sites, and a further enrichment of this signal at NPAS4-bound sites relative to FOS-bound sites (included in the revised manuscript as **Extended Data Fig. 12f**).*

At a minimum, all BLISS data should be presented (all time points, all samples) with information about the number of unique reads in each sample and which of these intersect with locations where gH2AX was enriched and/or regions of transcriptional activity.

*We thank the reviewer for these helpful suggestions. We now plot the average \pm SEM on aggregate plots (**Fig. 3f, Extended Data Fig. 10g, Fig. 11d-h, Fig. 12d**) to show the variability of independent replicates. As the reviewer requested, we now provide the data for each of our independent replicates at activity-dependent regulatory elements (see **Extended Data Fig. 12e**). We provide the numbers of reads and unique cut sites for all replicates in **Extended Data Table 1**. The consistency of the changes in BLISS signal across each of our replicates argues that this assay can capture consistent DNA breaks that are present in the majority of the cells at a given timepoint. We also show a significant correlation between the log₂ count of γ H2AX and BLISS in stimulated neurons (**Extended Data Figure 10k**). We note that this correlation would not be predicted to line up perfectly for each given genomic location due to the localization of γ H2AX to gene bodies (as can be seen **Fig. 3b, Extended Data Fig. 10b** and reported elsewhere) and the localization of BLISS primarily to gene promoters^{3,14}.*

*Regarding the relationship between BLISS signal and transcriptional activity, in our new BLISS timecourse we performed RNA-seq in parallel with the same samples. This allowed us to compare BLISS signal to transcriptional induction more directly. We find that BLISS signal tracks closely with the level of transcription (**Extended Data Fig. 10g**). Indeed, the mice in our 10 hour timepoint show variability in the level of transcriptional induction, which is common at this timepoint following kainate stimulation. We find that those samples with sustained high levels of activity-inducible genes also have higher levels of BLISS signal, while the samples in which activity-inducible genes are returning to basal levels also show BLISS signal returning to basal levels. We believe these findings provide evidence that the high levels of transcription from these sites evoked by neuronal activity coincide with elevated levels of DSBs at these sites.*

It would also be important to see how these sites compare with the patterns of mapped repair sites in Reid et al Science 2021 and the Nussenzweig Nature paper.

See comments above regarding a new analysis of SAR-seq data, copied here:

*As the reviewer suggested, we analyzed the SAR-seq data¹⁵. We thank the reviewer for this excellent suggestion. Although this assay is not limited to detecting DSBs, it should detect any synthesis-dependent repair that may occur in the course of repairing the induced DSBs we detect at activity-inducible sites in the genome by NHEJ or other mechanisms. Our analysis of the SAR-seq data reveals enrichment of the repair signal at NPAS4-bound sites, and a further enrichment of this signal at NPAS4-bound sites relative to FOS-bound sites (see **Extended Data Fig. 12f**).*

2. Related to this, the authors use multiple different approaches to categorize and compare genomic sites that make it incredibly difficult to assess the data, and not all results (samples or time points) are shown in each analysis.

For example, in Figure 3c, it appears that genes that are actively transcribed in a NPAS-dependent manner are plotted, but in the rest of the figure, it is a different set of sites (defined by ATAC-seq and H327Ac data).

Please see response to reviewer #2, pasted below.

We apologize for the brevity of our explanations in the text and figure legends. Throughout the text, we have now made an effort to carefully explain how select genomic locations were chosen and our rationale for performing an analysis in a particular way. We appreciate that assigning NPAS4 binding to select sets of genes can be nuanced, as many transcription factors bind many kilobases away from their target genes and can be involved in complex looping interactions in three-dimensional space. We no longer focus on activity-dependent genes but rather on the specific regulatory elements themselves.

Indeed, to enhance clarity of presentation, we now focus the majority of our analyses in Figures 3 and 4 (and associated Extended Data) on regulatory elements themselves, (defined as regions of ATAC-seq and/or H3K27ac). We now provide the definitions of these sites in a new figure, **Extended Data Fig. 9**. Furthermore, we subset these regulatory elements by high or low levels of NPAS4 binding (quartile 4 of IgG normalized NPAS4 CUT&RUN signal with a statistical peak of NPAS4 binding, versus quartile 1 of IgG normalized NPAS4 CUT&RUN signal with no statistical peak of NPAS4 binding). We believe this analysis provides a more direct assessment of how the NPAS4:NuA4 complex may influence damage and repair at its target sites. In addition to looking at all regulatory elements subset by NPAS4 binding quartiles, we further examine a subset of these sites that increase in ATAC and H3K27ac following neuronal stimulation (termed 'activity-inducible' elements; n=11,114 sites), and again subset by NPAS4 binding quartiles. We focus on these inducible elements to gain a better understanding of how DNA damage and repair are influenced at sites with dynamic changes to chromatin.

In addition, for certain comparisons, we also believe it is informative to focus on differences between NPAS4-bound sites and sites bound by another activity-inducible transcription factor, FOS (which does not interact with the NuA4 complex). In these comparisons, we consider NPAS4-bound sites that lack FOS or FOS sites that lack NPAS4 (see **Extended Data Fig. 11c,f,g-i**)

And in Figure 3f, the data from the KO from non-induced sites should be shown in the same way.

The Cre versus Δ Cre BLISS data has now been moved to **Fig. 4 and Extended Data Fig. 15**. We show data on BLISS counts in *Npas4^{fl/fl}* mice and wild-type mice (Cre vs Δ Cre) at both activity-inducible regulatory elements and all regulatory elements as boxplots.

If the model is that NuA4 functions to mediate repair at NPAS4-dependent transcriptionally active genes, then that should be how the data is divided for comparison in all analyses. The differences in gH2AX CHIP levels, DSB (BLISS) numbers, and MRE11 binding should then be shown according to the same classification at the same sites for the WT, NPAS4^{fl/fl} KO, and TIP60^{fl/fl} KO for all three time points.

We feel that looking directly at regulatory elements bound by the complex is the most proximal measure of the complex activity, as linking genes to specific regulatory elements (e.g. enhancers) can be imprecise.

We now consistently analyze the same two sets of sites for γ H2AX, BLISS-seq, MRE11 and RAD50 binding. This includes:

a) all regulatory elements (defined as the union of H3K27ac and ATAC-seq, **Extended Data Fig. 9**), subset by quartiles of NPAS4 binding

b) a subset of this larger group of sites that increase in ATAC and H3K27ac signal following neuronal stimulation (termed 'activity-inducible' elements, n=11,114) and also subset by NPAS4 binding quartiles. We focus on these inducible elements to gain a better understanding of how DNA damage and repair are influenced at sites with dynamic changes to chromatin resulting from elevated neuronal activity.

*Given the challenging nature of injecting animals to remove these proteins, and obtaining genome-wide datasets from limited numbers of nuclei freshly isolated from the brain, we were not able to perform all three assays (γH2AX, BLISS-Seq and MRE11) in all three genetic manipulations at three timepoints. Instead, we focus on manipulating NPAS4, which is the activity-dependent component of the complex and which has never before been implicated in repair. We performed the wildtype control experiment at a single timepoint, 2 hours post-stimulation, because this is the time when NPAS4 is maximally expressed. As we did not observe a difference between Cre- and ΔCre-treated nuclei in wild-type mice at this timepoint, as was observed in Npas4^{fl/fl} mice (**Extended Data Fig. 15**), we did not perform this control at all three timepoints.*

3. The NuA4/TIP0 complex has been implicated in DNA repair in human cells in a number of studies, and this work is not cited here. In particular, Tang et al NSMB 2013 and Jacquet et al Mol Cell 2016 showed that this complex is important for promoting the use of homologous recombination (HR) in the S and G2 phases of the cell cycle to repair DNA DSBs and prevents 53BP1 binding. This raises the question of what repair mechanism could be in use in these post-mitotic cells. There is data around use of RNA-templated recombination (see Meers, et al DNA Repair 2016), which could be relevant here. This should be discussed, and the relevant papers cited.

We apologize for this oversight in citing these important papers. We now include the human citations on line 267 of the text and address in the Discussion how NuA4 may be modulating repair in neurons, including the possibility that RNA may be part of the repair process^{16,17}, which is particularly interesting in post-mitotic cells (lines 628-639). In fact, NuA4 has been shown to be recruited to R-loops¹⁸ (DNA:RNA hybrids), raising the idea that these R-loops may contribute to repair in neurons. The idea of RNA-templated DNA repair in neurons is an exciting idea we plan to pursue in future studies.

The mutation data in Figure 4, while showing relatively modest effects, is compelling. Again, interesting to see how this compares with the repair sites mapped in the Science/Nature papers, which mention G4 structures at NPAS4 sites. However, it doesn't directly implicate failed DSB repair at these sites and could reflect altered SSB repair (or other pathways) instead.

We now acknowledge in the text that NPAS4:NuA4 may be involved in multiple forms of repair (lines 628-639). The reviewer suggests an important analysis to better understand the relationship between G-quadruplex structures, NPAS4:NuA4 binding, and DNA DSBs. We performed an analysis of the predicted G-quadruplex structures used in the Reid et al. paper^{19,20} and observed a 25-30% overlap of NPAS4-bound sites with observed G-quadruplex structures from human cells, compared to a 9-12% overlap

with FOS-bound sites (**Rebuttal Fig. 8**). A more thorough experiment to address this question would require knowledge of NPAS4- and FOS-bound sites in human neurons combined with a high-quality dataset of observed G-quadruplexes in human neurons, or genome-wide mapping of observed G-quadruplexes in the mouse brain. This will be an area of active investigation in the future.

[This has been redacted]

[This has been redacted]

[This has been redacted]

References

- 1 Meers, M. P., Tenenbaum, D. & Henikoff, S. Peak calling by Sparse Enrichment Analysis for CUT&RUN chromatin profiling. *Epigenetics Chromatin* **12**, 42, doi:10.1186/s13072-019-0287-4 (2019).
- 2 Salifou, K. *et al.* Chromatin-associated MRN complex protects highly transcribing genes from genomic instability. *Sci Adv* **7**, doi:10.1126/sciadv.abb2947 (2021).
- 3 Robert, F. *et al.* The transcriptional histone acetyltransferase cofactor TRRAP associates with the MRN repair complex and plays a role in DNA double-strand break repair. *Mol Cell Biol* **26**, 402-412, doi:10.1128/MCB.26.2.402-412.2006 (2006).
- 4 Bae, T. *et al.* Different mutational rates and mechanisms in human cells at pregastrulation and neurogenesis. *Science* **359**, 550-555, doi:10.1126/science.aan8690 (2018).
- 5 Lodato, M. A. *et al.* Somatic mutation in single human neurons tracks developmental and transcriptional history. *Science* **350**, 94-98, doi:10.1126/science.aab1785 (2015).
- 6 Milholland, B. *et al.* Differences between germline and somatic mutation rates in humans and mice. *Nat Commun* **8**, 15183, doi:10.1038/ncomms15183 (2017).
- 7 Tomkova, M., McClellan, M., Kriaucionis, S. & Schuster-Boeckler, B. 5-hydroxymethylcytosine marks regions with reduced mutation frequency in human DNA. *Elife* **5**, doi:10.7554/eLife.17082 (2016).
- 8 Lodato, M. A. *et al.* Aging and neurodegeneration are associated with increased mutations in single human neurons. *Science* **359**, 555-559, doi:10.1126/science.aao4426 (2018).
- 9 Sabatini, P. V. *et al.* Neuronal PAS Domain Protein 4 Suppression of Oxygen Sensing Optimizes Metabolism during Excitation of Neuroendocrine Cells. *Cell Rep* **22**, 163-174, doi:10.1016/j.celrep.2017.12.033 (2018).
- 10 Zullo, J. M. *et al.* Regulation of lifespan by neural excitation and REST. *Nature* **574**, 359-364, doi:10.1038/s41586-019-1647-8 (2019).
- 11 Evason, K., Huang, C., Yamben, I., Covey, D. F. & Kornfeld, K. Anticonvulsant medications extend worm life-span. *Science* **307**, 258-262, doi:10.1126/science.1105299 (2005).
- 12 Wiltschko, A. B. *et al.* Revealing the structure of pharmacobehavioral space through motion sequencing. *Nat Neurosci* **23**, 1433-1443, doi:10.1038/s41593-020-00706-3 (2020).
- 13 Dellino, G. I. *et al.* Release of paused RNA polymerase II at specific loci favors DNA double-strand-break formation and promotes cancer translocations. *Nat Genet* **51**, 1011-1023, doi:10.1038/s41588-019-0421-z (2019).
- 14 Gothe, H. J. *et al.* Spatial Chromosome Folding and Active Transcription Drive DNA Fragility and Formation of Oncogenic MLL Translocations. *Mol Cell* **75**, 267-283 e212, doi:10.1016/j.molcel.2019.05.015 (2019).
- 15 Wu, W. *et al.* Neuronal enhancers are hotspots for DNA single-strand break repair. *Nature*, doi:10.1038/s41586-021-03468-5 (2021).
- 16 Shen, Y. *et al.* RNA-driven genetic changes in bacteria and in human cells. *Mutat Res* **717**, 91-98, doi:10.1016/j.mrfmmm.2011.03.016 (2011).

- 17 Keskin, H. *et al.* Transcript-RNA-templated DNA recombination and repair. *Nature* **515**, 436-439, doi:10.1038/nature13682 (2014).
- 18 Fazio, T. G. Regulation of chromatin structure and cell fate by R-loops. *Transcription* **7**, 121-126, doi:10.1080/21541264.2016.1198298 (2016).
- 19 Reid, D. A. *et al.* Incorporation of a nucleoside analog maps genome repair sites in postmitotic human neurons. *Science* **372**, 91-94, doi:10.1126/science.abb9032 (2021).
- 20 Chambers, V. S. *et al.* High-throughput sequencing of DNA G-quadruplex structures in the human genome. *Nat Biotechnol* **33**, 877-881, doi:10.1038/nbt.3295 (2015).

Reviewer Reports on the First Revision:

Referees' comments:

Referee #2 (Remarks to the Author):

The authors have addressed all of my concerns and critiques from the first submission. The revised manuscript more clearly explains the number of replicates used for each experiment and clarifies many of the analysis procedures. The paper is compelling, timely and important, and I look forward to its publication.

Referee #3 (Remarks to the Author):

In revision the authors have carried out a vast amount of work in an attempt to validate or extend specific findings and conclusions in response to the detailed comments by all four referees.

Remarks made by several referees such as on clarity, reproducibility, statistics or on alternative interpretation of experimental findings (e.g., the synaptic phenotype of the conditional NPAS4 mutation) have been overall addressed in an adequate manner. This has significantly improved the manuscript.

My main conceptual and technical concern dealt with the connection between NPAS4 neuron-specific gene activation and the new phenomenon of focused repair of activity-dependent regions in the genome, via interaction with NPAS4. In response and revision, the authors have added additional evidence or modified their interpretation. I will address the various aspects one-by-one:

1. I am happy with the remark on additional manners by which MRE11 may be recruited as mentioned in lines 459-463 of the revised manuscript. This is in line with the generally dynamic nature of DNA repair pathways, which is a widely accepted concept within the DNA repair field. Additionally, the remark by the authors that the nature of the MRE11 recruitment is still unclear and could be either direct, or indirect, as well as the acknowledgment that other repair pathways may contribute to DNA repair at sites undergoing activity-induced DSBs (lines 628-639) are appreciated.
2. In response to comments by referee 2 and 3, the authors have decided to remove the analysis of the mutations in NPAS4 KO mice and toning down the emphasis on the findings on somatic mutations. Also, these modifications constitute improvements, as these results were difficult to compare with others and raised questions. It is a pity that determining insertion and deletion events, which are the main consequence of DSB repair failure, only yielded a trend towards increased insertions and deletions across multiple datasets, but that this trend failed to reach significance, because the events appear extremely rare. I do not advise further delving into this aspect as inactivation of NPAS4 will have multiple consequences which might turn out to be difficult to disentangle and the effect on DSB-mediated insertions and deletions might be subtle, as it most likely is not a dominant pathway and there are alternative processes.

3. A major issue is the occurrence of DSBs in an activity- and NPAS4-dependent manner. The authors in their rebuttal indicate that they have made several attempts using different strategies to get a better and accurate quantitation of these DSBs. Although some of the methods used did not work (such as γ H2AX ChIP-seq and recruitment of the DSB repair enzymes MRE11 and RAD50 in an activity- and NPAS4-dependent manner), they have extended the number of samples for direct measures of DNA breaks using BLISS-seq to improve the statistical power and complemented this with the END-seq method. Overall, these additions make the important conclusion on the presence of elevated DSBs at NPAS4 sites clearly stronger. Concerning the possibility that induction of CRE may also contribute to the generation of DSBs, the authors now acknowledge this option in the text, demonstrating that they also considered this possibility.

4. The connection of the DSB with repair. The additional analysis of another member of the MRN complex, Rad50, using CUT&RUN (presented in Fig. 4a,b; Extended Data Fig. 13f) makes the finding that this complex preferentially binds at NPAS4-bound sites in an activity-dependent manner much more convincing. This referee is happy with this result. On the other hand, it is a pity that treating neurons with DNA-damaging agents is not sufficient to induce NPAS4 in the absence of neuronal depolarization (Rebuttal Fig. 4), and that probing sensitivity to various genotoxic compounds requires experiments in mice rather than cultured neuronal lines. Unfortunately, this prohibits a better characterization of the DNA repair phenotype of NPAS4 deficiency.

5. A main worry of this referee was that “part of these ‘DSBs’ might simply be intermediates of normal DNA metabolism that are captured in the middle of a DNA transaction at the moment of cell lysis”. Examples could be topoisomerase-DNA intermediates involved in transcription-related unwinding reactions. Alternatively, SSBs can be intermediates of frequently occurring base excision repair events. Such breaks would have been sealed moments later, when cells would not have been lysed. Additionally, when genes are activated after being transcriptional inactive for a longer period of time there will be accumulated DNA lesions in the transcribed strand that will stall elongating RNA polII complexes in the first round of transcription. These need to be repaired by transcription-coupled repair, delaying transcriptional progress. This may favor R-loop formation, which would occur particularly in activated genes, like those induced by NPAS4. Resolving these R-loops and stalled transcription complexes may well involve generation of DSBs, requiring MRN and DSB repair, facilitated by recruitment by NPAS4. This would be consistent with the acute nature of the response and the notion that the NuA4 complex has been reported to bind to DNA:RNA hybrids (R-loops) in non-neuronal cells (ref 70, cited in the manuscript). Moreover, as noted by the authors in their rebuttal to comment 1 of referee 4: “BLISS has been shown in a number of other studies in non-neuronal cells to faithfully mark DNA breaks at highly expressed genes and genes with high levels of paused RNA Pol II (refs 13,14)”. Additionally, it would be in line with the remarks by the author in revision: “... the enrichment of synthesis-dependent DNA repair signal (SAR-seq), which captures multiple forms of repair, at NPAS4 binding sites (Extended Data Fig. 12f), raises the possibility that the NPAS4:NuA4 complex may engage with additional repair pathways beyond DSB machinery” (lines 634-637). This is a possible scenario that the authors might entertain in their manuscript.

6. Finally, the cause of death of NPAS4 KO mice with a short lifespan is, unfortunately, still enigmatic. The survival curve of the *Npas4* *Camk2a*-Cre cKO mice in extended figure 18, shown as an alternative, is rather unusual as a large fraction of the conditional KO animals dies early, but a significant minority appears to have a close to wt survival (Extended figure 18C), rendering the difference barely significant ($p=0,049$). Is there an explanation for this diverse lifespan? [REDACTED]

[This has been redacted]

[This has been redacted]

I have carefully gone through all the comments, rebuttal and this extensive manuscript. The revised manuscript has been significantly improved after the comments of the referees. Apart from the remark above (see point 5) I believe the impressive findings reported are now convincing, including a connection with DNA repair.

Jan H.J. Hoeijmakers

Referee #4 (Remarks to the Author):

The authors have very thoroughly and thoughtfully addressed the concerns raised with the previous manuscript. The inclusion of additional data as well as improvements to the clarity of the analysis and data presentation have greatly improved the manuscript. I am supportive of acceptance.

Author Rebuttals to First Revision:

Referee #2 (Remarks to the Author):

The authors have addressed all of my concerns and critiques from the first submission. The revised manuscript more clearly explains the number of replicates used for each experiment and clarifies many of the analysis procedures. The paper is compelling, timely and important, and I look forward to its publication.

Referee #3 (Remarks to the Author):

In revision the authors have carried out a vast amount of work in an attempt to validate or extend specific findings and conclusions in response to the detailed comments by all four referees.

Remarks made by several referees such as on clarity, reproducibility, statistics or on alternative interpretation of experimental findings (e.g., the synaptic phenotype of the conditional NPAS4 mutation) have been overall addressed in an adequate manner. This has significantly improved the manuscript.

My main conceptual and technical concern dealt with the connection between NPAS4 neuron-specific gene activation and the new phenomenon of focused repair of activity-dependent regions in the genome, via interaction with NPAS4. In response and revision, the authors have added additional evidence or modified their interpretation. I will address the various aspects one-by-one:

1. I am happy with the remark on additional manners by which MRE11 may be recruited as mentioned in lines 459-463 of the revised manuscript. This is in line with the generally dynamic nature of DNA repair pathways, which is a widely accepted concept within the DNA repair field. Additionally, the remark by the authors that the nature of the MRE11 recruitment is still unclear and could be either direct, or indirect, as well as the acknowledgment that other repair pathways may contribute to DNA repair at sites undergoing activity-induced DSBs (lines 628-639) are appreciated.
2. In response to comments by referee 2 and 3, the authors have decided to remove the analysis of the mutations in NPAS4 KO mice and toning down the emphasis on the findings on somatic mutations. Also, these modifications constitute improvements, as these results were difficult to compare with others and raised questions. It is a pity that determining insertion and deletion events, which are the main consequence of DSB repair failure, only yielded a trend towards increased insertions and deletions across multiple datasets, but that this trend failed to reach significance, because the events appear extremely rare. I do not advise further delving into this aspect as inactivation of NPAS4 will have multiple consequences which might turn out to be difficult to disentangle and the effect on DSB-mediated insertions and deletions might be subtle, as it most likely is not a dominant pathway and there are alternative processes.
3. A major issue is the occurrence of DSBs in an activity- and NPAS4-dependent manner. The authors in their rebuttal indicate that they have made several attempts

using different strategies to get a better and accurate quantitation of these DSBs. Although some of the methods used did not work (such as γ H2AX ChIP-seq and recruitment of the DSB repair enzymes MRE11 and RAD50 in an activity- and NPAS4-dependent manner), they have extended the number of samples for direct measures of DNA breaks using BLISS-seq to improve the statistical power and complemented this with the END-seq method. Overall, these additions make the important conclusion on the presence of elevated DSBs at NPAS4 sites clearly stronger. Concerning the possibility that induction of CRE may also contribute to the generation of DSBs, the authors now acknowledge this option in the text, demonstrating that they also considered this possibility.

4. The connection of the DSB with repair. The additional analysis of another member of the MRN complex, Rad50, using CUT&RUN (presented in Fig. 4a,b; Extended Data Fig. 13f) makes the finding that this complex preferentially binds at NPAS4-bound sites in an activity-dependent manner much more convincing. This referee is happy with this result. On the other hand, it is a pity that treating neurons with DNA-damaging agents is not sufficient to induce NPAS4 in the absence of neuronal depolarization (Rebuttal Fig. 4), and that probing sensitivity to various genotoxic compounds requires experiments in mice rather than cultured neuronal lines. Unfortunately, this prohibits a better characterization of the DNA repair phenotype of NPAS4 deficiency.

5. A main worry of this referee was that “part of these ‘DSBs’ might simply be intermediates of normal DNA metabolism that are captured in the middle of a DNA transaction at the moment of cell lysis”. Examples could be topoisomerase-DNA intermediates involved in transcription-related unwinding reactions. Alternatively, SSBs can be intermediates of frequently occurring base excision repair events. Such breaks would have been sealed moments later, when cells would not have been lysed. Additionally, when genes are activated after being transcriptional inactive for a longer period of time there will be accumulated DNA lesions in the transcribed strand that will stall elongating RNA polII complexes in the first round of transcription. These need to be repaired by transcription-coupled repair, delaying transcriptional progress. This may favor R-loop formation, which would occur particularly in activated genes, like those induced by NPAS4. Resolving these R-loops and stalled transcription complexes may well involve generation of DSBs, requiring MRN and DSB repair, facilitated by recruitment by NPAS4. This would be consistent with the acute nature of the response and the notion that the NuA4 complex has been reported to bind to DNA:RNA hybrids (R-loops) in non-neuronal cells (ref 70, cited in the manuscript). Moreover, as noted by the authors in their rebuttal to comment 1 of referee 4: “BLISS has been shown in a number of other studies in non-neuronal cells to faithfully mark DNA breaks at highly expressed genes and genes with high levels of paused RNA Pol II (refs 13,14)”. Additionally, it would be in line with the remarks by the author in revision: “... the enrichment of synthesis-dependent DNA repair signal (SAR-seq), which captures multiple forms of repair, at NPAS4 binding sites (Extended Data Fig. 12f), raises the possibility that the NPAS4:NuA4 complex may engage with additional repair pathways beyond DSB machinery” (lines 634-637). This is a possible scenario that the authors might entertain in their manuscript.

6. Finally, the cause of death of NPAS4 KO mice with a short lifespan is, unfortunately, still enigmatic. The survival curve of the *Npas4* *Camk2a*-Cre cKO mice in extended figure 18, shown as an alternative, is rather unusual as a large fraction of the conditional KO animals dies early, but a significant minority appears to have a

close to wt survival (Extended figure 18C), rendering the difference barely significant ($p=0,049$). Is there an explanation for this diverse lifespan?

[This has been redacted]

I have carefully gone through all the comments, rebuttal and this extensive manuscript. The revised manuscript has been significantly improved after the comments of the referees. Apart from the remark above (see point 5) I believe the impressive findings reported are now convincing, including a connection with DNA repair.

Jan H.J. Hoeijmakers

We thank the reviewer for his thorough analysis and believe his contributions have strengthened our manuscript. Regarding the points raised in comment 5, the reviewer has asked that we clarify in the manuscript that BLISS and END-seq capture both long-lived double-strand breaks and more transient breaks.

Furthermore, the reviewer asks that we mention in the manuscript that there are several possible mechanisms that could give rise to DSBs at NPAS4-bound sites. One possibility is that these breaks arise from pre-positioned topoisomerase enzymes that are post-translationally modified downstream of neuronal activity. As the reviewer suggests, it is also possible these breaks form as a result of resolving "R-loops and stalled transcription complexes" that occur with rapid transcriptional induction of previously quiescent regulatory elements. We will address these points with several sentences added to the manuscript as indicated below.

The final point of the reviewer is that there may be additional forms of DNA repair engaged at NPAS4 sites. We have acknowledged these possibilities in the revised manuscript with the following addition to the Discussion section:

"The mechanisms leading to both the formation and repair of DSBs at NPAS4-bound sites remain to be fully elucidated. As previously suggested⁴, these breaks may arise from pre-bound topoisomerase enzymes that are post-translationally modified downstream of neuronal activity. In addition, DSBs may form in the process of resolving DNA:RNA hybrids (R-loops) or releasing stalled transcription complexes that occur with rapid induction of previously quiescent regulatory elements. Notably, NuA4 has been reported to bind R-loops, which suggests that R-loop formation may contribute to both DNA damage and repair at activity-dependent regulatory elements. Intriguing work outside the nervous system has suggested that RNA itself could facilitate repair of these transcribed regions by serving as a template in place of a sister chromatid in post-mitotic cells. Although it is likely that canonical NHEJ pathways mediate much of the DSB repair in activated neurons, it is possible that NPAS4:NuA4 may engage multiple repair pathways. Future studies that probe the precise mechanisms neurons employ to repair activity-induced damage, including those mediated by NPAS4:NuA4, will be a critical area of investigation."

Referee #4 (Remarks to the Author):

The authors have very thoroughly and thoughtfully addressed the concerns raised with the previous manuscript. The inclusion of additional data as well as improvements to the clarity of the analysis and data presentation have greatly improved the manuscript. I am supportive of acceptance.

References

- 1 Meers, M. P., Tenenbaum, D. & Henikoff, S. Peak calling by Sparse Enrichment Analysis for CUT&RUN chromatin profiling. *Epigenetics Chromatin* **12**, 42, doi:10.1186/s13072-019-0287-4 (2019).
- 2 Salifou, K. *et al.* Chromatin-associated MRN complex protects highly transcribing genes from genomic instability. *Sci Adv* **7**, doi:10.1126/sciadv.abb2947 (2021).
- 3 Robert, F. *et al.* The transcriptional histone acetyltransferase cofactor TRRAP associates with the MRN repair complex and plays a role in DNA double-strand break repair. *Mol Cell Biol* **26**, 402-412, doi:10.1128/MCB.26.2.402-412.2006 (2006).
- 4 Bae, T. *et al.* Different mutational rates and mechanisms in human cells at pregastrulation and neurogenesis. *Science* **359**, 550-555, doi:10.1126/science.aan8690 (2018).
- 5 Lodato, M. A. *et al.* Somatic mutation in single human neurons tracks developmental and transcriptional history. *Science* **350**, 94-98, doi:10.1126/science.aab1785 (2015).
- 6 Milholland, B. *et al.* Differences between germline and somatic mutation rates in humans and mice. *Nat Commun* **8**, 15183, doi:10.1038/ncomms15183 (2017).
- 7 Tomkova, M., McClellan, M., Kriaucionis, S. & Schuster-Boeckler, B. 5-hydroxymethylcytosine marks regions with reduced mutation frequency in human DNA. *Elife* **5**, doi:10.7554/eLife.17082 (2016).
- 8 Lodato, M. A. *et al.* Aging and neurodegeneration are associated with increased mutations in single human neurons. *Science* **359**, 555-559, doi:10.1126/science.aao4426 (2018).
- 9 Sabatini, P. V. *et al.* Neuronal PAS Domain Protein 4 Suppression of Oxygen Sensing Optimizes Metabolism during Excitation of Neuroendocrine Cells. *Cell Rep* **22**, 163-174, doi:10.1016/j.celrep.2017.12.033 (2018).
- 10 Zullo, J. M. *et al.* Regulation of lifespan by neural excitation and REST. *Nature* **574**, 359-364, doi:10.1038/s41586-019-1647-8 (2019).
- 11 Evason, K., Huang, C., Yamben, I., Covey, D. F. & Kornfeld, K. Anticonvulsant medications extend worm life-span. *Science* **307**, 258-262, doi:10.1126/science.1105299 (2005).
- 12 Wiltchko, A. B. *et al.* Revealing the structure of pharmacobehavioral space through motion sequencing. *Nat Neurosci* **23**, 1433-1443, doi:10.1038/s41593-020-00706-3 (2020).
- 13 Dellino, G. I. *et al.* Release of paused RNA polymerase II at specific loci favors DNA double-strand-break formation and promotes cancer translocations. *Nat Genet* **51**, 1011-1023, doi:10.1038/s41588-019-0421-z (2019).

- 14 Gothe, H. J. *et al.* Spatial Chromosome Folding and Active Transcription Drive DNA Fragility and Formation of Oncogenic MLL Translocations. *Mol Cell* **75**, 267-283 e212, doi:10.1016/j.molcel.2019.05.015 (2019).
- 15 Wu, W. *et al.* Neuronal enhancers are hotspots for DNA single-strand break repair. *Nature*, doi:10.1038/s41586-021-03468-5 (2021).
- 16 Shen, Y. *et al.* RNA-driven genetic changes in bacteria and in human cells. *Mutat Res* **717**, 91-98, doi:10.1016/j.mrfmmm.2011.03.016 (2011).
- 17 Keskin, H. *et al.* Transcript-RNA-templated DNA recombination and repair. *Nature* **515**, 436-439, doi:10.1038/nature13682 (2014).
- 18 Fazio, T. G. Regulation of chromatin structure and cell fate by R-loops. *Transcription* **7**, 121-126, doi:10.1080/21541264.2016.1198298 (2016).
- 19 Reid, D. A. *et al.* Incorporation of a nucleoside analog maps genome repair sites in postmitotic human neurons. *Science* **372**, 91-94, doi:10.1126/science.abb9032 (2021).
- 20 Chambers, V. S. *et al.* High-throughput sequencing of DNA G-quadruplex structures in the human genome. *Nat Biotechnol* **33**, 877-881, doi:10.1038/nbt.3295 (2015).